# A Survey of Multi-agent Reinforcement Learning from Game Theoretical Perspective

## Abstract

Following the remarkable success of the AlphaGO series, 2019 was a booming year that witnessed significant advances in multi-agent reinforcement learning (MARL) techniques. MARL corresponds to the learning problem in a multi-agent system in which multiple agents learn simultaneously. It is an interdisciplinary domain with a long history that includes game theory, machine learning, stochastic control, psychology, and optimisation. Although MARL has achieved considerable empirical success in solving real-world games, there is a lack of a self-contained overview in the literature that elaborates the game theoretical foundations of modern MARL methods and summarises the recent advances. In fact, the majority of existing surveys are outdated and do not fully cover the recent developments since 2010. In this work, we provide a monograph on MARL that covers both the fundamentals and the latest developments in the research frontier.

This survey is separated into two parts. From §1 to §4, we present the self-contained fundamental knowledge of MARL, including problem formulations, basic solutions, and existing challenges. Specifically, we present the MARL formulations through two representative frameworks, namely, stochastic games and extensive-form games, along with different variations of games that can be addressed. The goal of this part is to enable the readers, even those with minimal related background, to grasp the key ideas in MARL research. From §5 to §9, we present an overview of recent developments of MARL algorithms. Starting from new taxonomies for MARL methods, we conduct a survey of previous survey papers. In later sections, we highlight several modern topics in MARL research, including Q-function factorisation, multi-agent soft learning, networked multi-agent MDP, stochastic potential games, zero-sum continuous games, online MDP, turn-based stochastic games, policy space response oracle, approximation methods in general-sum games, and mean-field type learning in games with infinite agents. Within each topic, we select both the most fundamental and cutting-edge algorithms.

The goal of our survey is to provide a self-contained assessment of the current state-of-the-art MARL techniques from a game theoretical perspective. We expect this work to serve as a stepping stone for both new researchers who are about to enter this fast-growing domain and existing domain experts who want to obtain a panoramic view and identify new directions based on recent advances.

# 1   Introduction

Machine learning can be considered as the process of converting data into knowledge (Shalev-Shwartz & Ben-David, 2014). The input of a learning algorithm is training data (for example, images containing cats), and the output is some knowledge (for example, rules about how to detect cats in an image). This knowledge is usually represented as a computer program that can perform certain task(s) (for example, an automatic cat detector). In the past decade, considerable progress has been made by means of a special kind of machine learning technique: deep learning (LeCun et al., 2015). One of the critical embodiments of deep learning is different kinds of deep neural networks (DNNs) (Schmidhuber, 2015) that can find disentangled representations (Bengio, 2009) in high-dimensional data, which allows the software to train itself to perform new tasks rather than merely relying on the programmer for designing hand-crafted rules. An uncountable number of breakthroughs in real-world AI applications have been achieved through the usage of DNNs, with the domains of computer vision (Krizhevsky et al., 2012) and natural language processing (Brown et al., 2020; Devlin et al., 2018) being the greatest beneficiaries.

In addition to feature recognition from existing data, modern AI applications often require computer programs to make decisions based on acquired knowledge (see Figure 1). To illustrate the key components of decision making, let us consider the real-world example of controlling a car to drive safely through an intersection. At each time step, a robot car can move by steering, accelerating and braking. The goal is to safely exit the intersection and reach the destination (with possible decisions of going straight or turning left/right into another lane). Therefore, in addition to being able to detect objects, such as traffic lights, lane markings, and other cars (by converting data to knowledge), we aim to find a steering policy that can control the car to make a sequence of manoeuvres to achieve the goal (making decisions based on the knowledge gained). In a decision-making setting such as this, two additional challenges arise:

1. First, during the decision-making process, at each time step, the robot car should consider not only the immediate value of its current action but also the consequences of its current action in the future. For example, in the case of driving through an intersection, it would be detrimental to have a policy that chooses to steer in a "safe" direction at the beginning of the process if it would eventually lead to a car crash later on.

2. Second, to make each decision correctly and safely, the car must also consider other cars' behaviour and act accordingly. Human drivers, for example, often predict in advance other cars' movements and then take strategic moves in response (like giving way to an oncoming car or accelerating to merge into another lane).

The need for an adaptive decision-making framework, together with the complexity of addressing multiple interacting learners, has led to the development of multi-agent RL. Multi-agent RL tackles the sequential decision-making problem of having multiple intelligent agents that operate in a shared stochastic environment, each of which targets to maximise its long-term reward through interacting with the environment and other agents. Multi-agent RL is built on the knowledge of both multi-agent systems (MAS) and RL. In the next section, we provide a brief overview of (single-agent) RL and the research developments in recent decades.

## 1.1   A Short History of RL

RL is a sub-field of machine learning, where agents learn how to behave optimally based on a trial-and-error procedure during their interaction with the environment. Unlike supervised learning, which takes labelled data as the input (for example, an image labelled with cats), RL is goal-oriented: it constructs a learning model that learns to achieve the optimal long-term goal by improvement through trial and error, with the learner having no labelled data to obtain knowledge from. The word "reinforcement" refers to the learning mechanism since the actions that lead to satisfactory outcomes are reinforced in the learner's set of behaviours.

Historically, the RL mechanism was originally developed based on studying cats' behaviour in a puzzle box (Thorndike, 1898). Minsky (1954) first proposed the computational model of RL in his Ph.D. thesis and

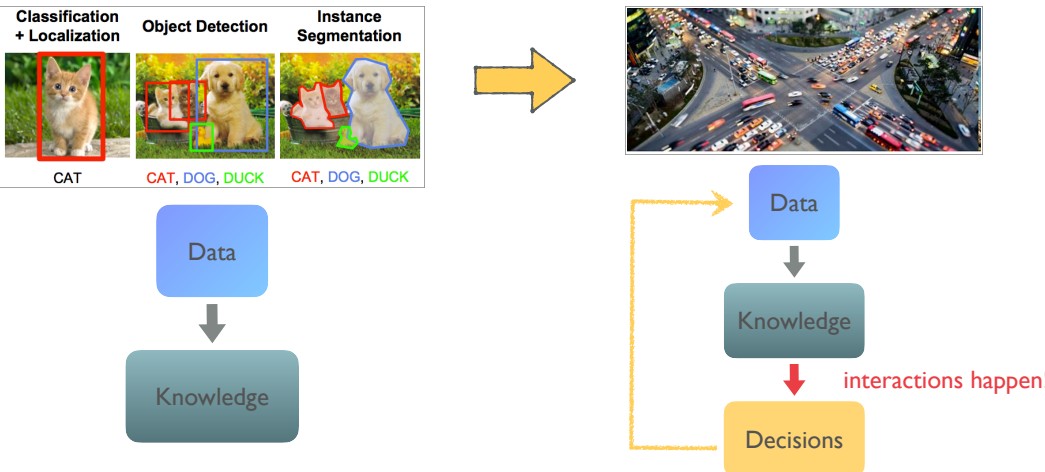

**Figure 1:** Modern AI applications are being transformed from pure feature recognition (for example, detecting a cat in an image) to decision making (driving through a traffic intersection safely), where interaction among multiple agents inevitably occurs. As a result, each agent has to behave strategically. Furthermore, the problem becomes more challenging because current decisions influence future outcomes.

named his resulting analog machine the *stochastic neural-analog reinforcement calculator*. Several years later, he first suggested the connection between dynamic programming (Bellman, 1952) and RL (Minsky, 1961). In 1972, Klopf (1972) integrated the trial-and-error learning process with the finding of *temporal difference (TD)* learning from psychology. TD learning quickly became indispensable in scaling RL for larger systems. On the basis of dynamic programming and TD learning, Watkins & Dayan (1992) laid the foundations for present day RL using the Markov decision process (MDP) and proposing the famous Q-learning method as the solver. As a dynamic programming method, the original Q-learning process inherits Bellman's "curse of dimensionality" (Bellman, 1952), which strongly limits its applications when the number of state variables is large. To overcome such a bottleneck, Bertsekas & Tsitsiklis (1996) proposed approximate dynamic programming methods based on neural networks. More recently, Mnih et al. (2015) from DeepMind made a significant breakthrough by introducing the deep Q-learning (DQN) architecture, which leverages the representation power of DNNs for approximate dynamic programming methods. DQN has demonstrated human-level performance on 49 Atari games. Since then, deep RL techniques have become common in machine learning/AI and have attracted considerable attention from the research community.

RL originates from an understanding of animal behaviour where animals use trial-and-error to reinforce beneficial behaviours, which they then perform more frequently. During its development, computational RL incorporated ideas such as optimal control theory and other findings from psychology that help mimic the way humans make decisions to maximise the long-term profit of decision-making tasks. As a result, RL methods can naturally be used to train a computer program (an agent) to a performance level comparable to that of a human on certain tasks. The earliest success of RL methods against human players can be traced back to the game of backgammon (Tesauro, 1995). More recently, the advancement of applying RL to solve sequential decision-making problems was marked by the remarkable success of the AlphaGo series (Silver et al., 2016; 2017; 2018), a self-taught RL agent that beats top professional players of the game GO, a game whose search space ($10^{761}$ possible games) is even greater than the number of atoms in the universe[1].

In fact, the majority of successful RL applications, such as those for the game GO[2], robotic control (Kober et al., 2013), and autonomous driving (Shalev-Shwartz et al., 2016), naturally involve the participation of multiple AI agents, which probe into the realm of MARL. As we would expect, the significant progress

---

[1]There are an estimated $10^{82}$ atoms in the universe. If one had one trillion computers, each processing one trillion states per second for one trillion years, one could only reach $10^{43}$ states.

[2]Arguably, AlphaGo can also be treated as a multi-agent technique if we consider the opponent in self-play as another agent.

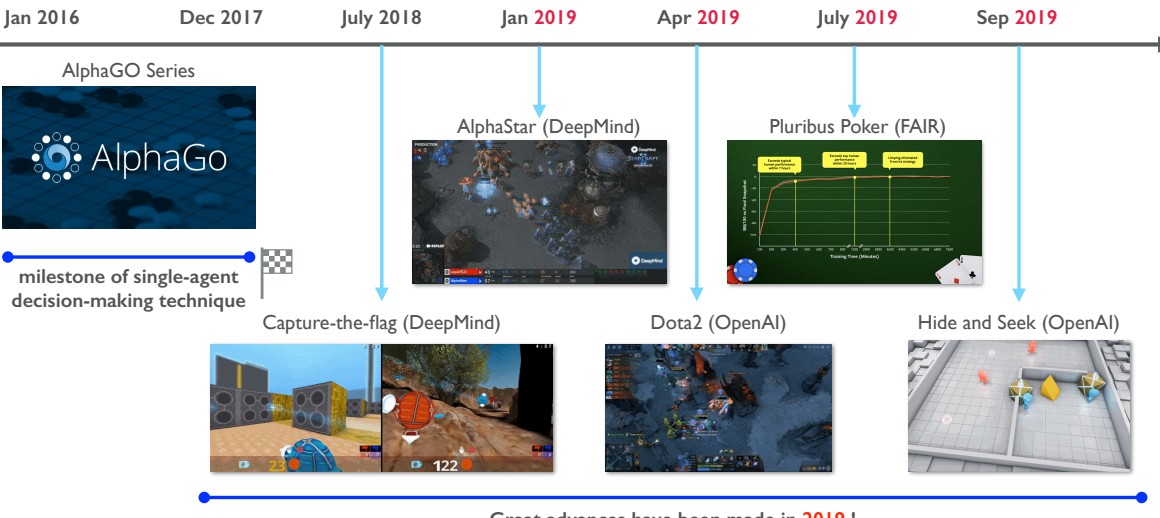

**Figure 2:** The success of the AlphaGo series marks the maturation of the single-agent decision-making process. The year 2019 was a booming year for MARL techniques; remarkable progress was achieved in solving immensely challenging multi-player real-strategy video games and multi-player incomplete-information poker games.

achieved by single-agent RL methods – marked by the 2016 success in GO – foreshadowed the breakthroughs of multi-agent RL techniques in the following years.

## 1.2   2019: A Booming Year for MARL

2019 was a booming year for MARL development as a series of breakthroughs were made in immensely challenging multi-agent tasks that people used to believe were impossible to solve via AI. Nevertheless, the progress made in the field of MARL, though remarkable, has been overshadowed to some extent by the prior success of AlphaGo (Chalmers, 2020). It is possible that the AlphaGo series (Silver et al., 2016; 2017; 2018) has largely fulfilled people's expectations for the effectiveness of RL methods, such that there is a lack of interest in further advancements in the field. The ripples caused by the progress of MARL were relatively mild among the research community. In this section, we highlight several pieces of work that we believe are important and could profoundly impact the future development of MARL techniques.

One popular test-bed of MARL is StarCraft II (Vinyals et al., 2017), a multi-player real-time strategy computer game that has its own professional league. In this game, each player has only limited information about the game state, and the dimension of the search space is orders of magnitude larger than that of GO ($10^{26}$ possible choices for every move). The design of effective RL methods for StarCraft II was once believed to be a long-term challenge for AI (Vinyals et al., 2017). However, a breakthrough was accomplished by AlphaStar in 2019 (Vinyals et al., 2019b), which has exhibited grandmaster-level skills by ranking above 99.8% of human players.

Another prominent video game-based test-bed for MARL is Dota2, a zero-sum game played by two teams, each composed of five players. From each agent's perspective, in addition to the difficulty of incomplete information (similar to StarCraft II), Dota2 is more challenging, in the sense that both cooperation among team members and competition against the opponents must be considered. The OpenAI Five AI system (Pachocki et al., 2018) demonstrated superhuman performance in Dota2 by defeating world champions in a public e-sports competition.

In addition to StarCraft II and Dota2, Jaderberg et al. (2019) and Baker et al. (2019a) showed human-level performance in capture-the-flag and hide-and-seek games, respectively. Although the games themselves are less sophisticated than either StarCraft II or Dota2, it is still non-trivial for AI agents to master their tactics,

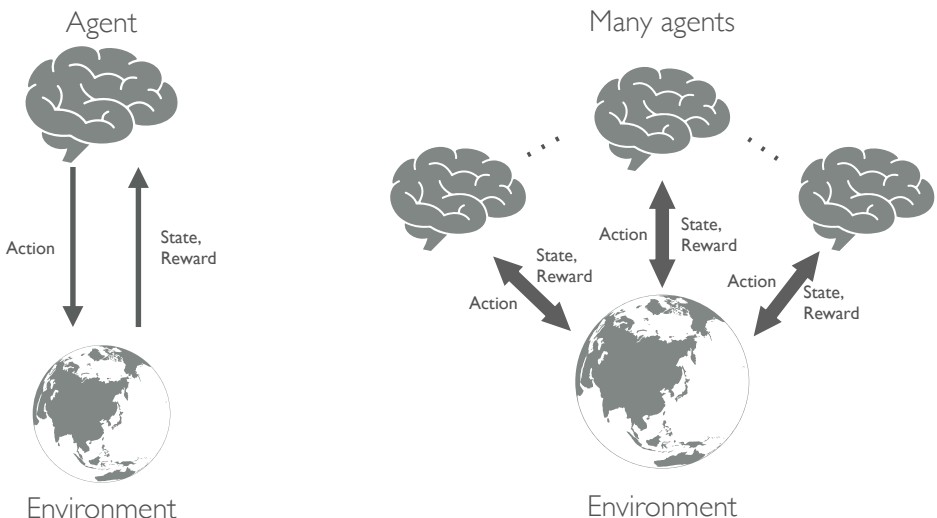

**Figure 3:** Diagram of a single-agent MDP (left) and a multi-agent MDP (right).

so the agents' impressive performance again demonstrates the efficacy of MARL. Interestingly, both authors reported emergent behaviours induced by their proposed MARL methods that humans can understand and are grounded in physical theory.

One last remarkable achievement of MARL worth mentioning is its application to the poker game Texas hold' em, which is a multi-player extensive-form game with incomplete information accessible to the player. Heads-up (namely, two player) no-limit hold'em has more than $6 \times 10^{161}$ information states. Only recently have ground-breaking achievements in the game been made, thanks to MARL. Two independent programs, *DeepStack* (Moravčík et al., 2017) and *Libratus* (Brown & Sandholm, 2018), are able to beat professional human players. Even more recently, Libratus was upgraded to Pluribus (Brown & Sandholm, 2019) and showed remarkable performance by winning over one million dollars from five elite human professionals in a no-limit setting.

For a deeper understanding of RL and MARL, mathematical notation and deconstruction of the concepts are needed. In the next section, we provide mathematical formulations for these concepts, starting from single-agent RL and progressing to multi-agent RL methods.

## 2 Single-Agent RL

Through trial and error, an RL agent attempts to find the optimal policy to maximise its long-term reward. This process is formulated by Markov Decision Processes.

### 2.1 Problem Formulation: Markov Decision Process

**Definition 1 (Markov Decision Process)** *An MDP can be described by a tuple of key elements* $\langle \mathbb{S}, \mathbb{A}, P, R, \gamma \rangle$.

- $\mathbb{S}$*: the set of environmental states.*

- $\mathbb{A}$*: the set of agent's possible actions.*

- $P : \mathbb{S} \times \mathbb{A} \to \Delta(\mathbb{S})$*: for each time step* $t \in \mathbb{N}$*, given agent's action* $a \in \mathbb{A}$*, the transition probability from a state* $s \in \mathbb{S}$ *to the state in the next time step* $s' \in \mathbb{S}$*.*

- $R : \mathbb{S} \times \mathbb{A} \times \mathbb{S} \to \mathbb{R}$*: the reward function that returns a scalar value to the agent for a transition from* $s$ *to* $s'$ *as a result of action* $a$*. The rewards have absolute values uniformly bounded by* $R_{max}$*.*

- $\gamma \in [0, 1]$ *is the discount factor that represents the value of time.*

At each time step $t$, the environment has a state $s_t$. The learning agent observes this state[3] and executes an action $a_t$. The action makes the environment transition into the next state $s_{t+1} \sim P(\cdot|s_t, a_t)$, and the new environment returns an immediate reward $R(s_t, a_t, s_{t+1})$ to the agent. The reward function can be also written as $R : \mathbb{S} \times \mathbb{A} \to \mathbb{R}$, which is interchangeable with $R : \mathbb{S} \times \mathbb{A} \times \mathbb{S} \to \mathbb{R}$ (see Van Otterlo & Wiering (2012), page 10). The goal of the agent is to solve the MDP: to find the optimal policy that maximises the reward over time. Mathematically, one common objective is for the agent to find a Markovian (i.e., the input depends on only the current state) and stationary (i.e., function form is time-independent) policy function[4] $\pi : \mathbb{S} \to \Delta(\mathbb{A})$, with $\Delta(\cdot)$ denoting the probability simplex, which can guide it to take sequential actions such that the discounted cumulative reward is maximised:

$$\mathbb{E}_{s_{t+1} \sim P(\cdot|s_t, a_t)} \left[ \sum_{t \geq 0} \gamma^t R(s_t, a_t, s_{t+1}) \Big| a_t \sim \pi(\cdot \mid s_t), s_0 \right]. \tag{1}$$

Another common mathematical objective of an MDP is to maximise the time-average reward:

$$\lim_{T \to \infty} \mathbb{E}_{s_{t+1} \sim P(\cdot|s_t, a_t)} \left[ \frac{1}{T} \sum_{t=0}^{T-1} R(s_t, a_t, s_{t+1}) \Big| a_t \sim \pi(\cdot \mid s_t), s_0 \right], \tag{2}$$

which we do not consider in this work and refer to Mahadevan (1996) for a full analysis of the objective of time-average reward.

Based on the objective function of Eq. (1), under a given policy $\pi$, we can define the state-action function (namely, the Q-function, which determines the expected return from undertaking action $a$ in state $s$) and the value function (which determines the return associated with the policy in state $s$) as:

$$Q^\pi(s, a) = \mathbb{E}^\pi \left[ \sum_{t \geq 0} \gamma^t R(s_t, a_t, s_{t+1}) \Big| a_0 = a, s_0 = s \right], \forall s \in \mathbb{S}, a \in \mathbb{A} \tag{3}$$

$$V^\pi(s) = \mathbb{E}^\pi \left[ \sum_{t \geq 0} \gamma^t R(s_t, a_t, s_{t+1}) \Big| s_0 = s \right], \forall s \in \mathbb{S} \tag{4}$$

where $\mathbb{E}^\pi$ is the expectation under the probability measure $\mathbb{P}^\pi$ over the set of infinitely long state-action trajectories $\tau = (s_0, a_0, s_1, a_1, ...)$ and where $\mathbb{P}^\pi$ is induced by state transition probability $P$, the policy $\pi$, the initial state $s$ and initial action $a$ (in the case of the Q-function). The connection between the Q-function and value function is $V^\pi(s) = \mathbb{E}_{a \sim \pi(\cdot|s)}[Q^\pi(s, a)]$ and $Q^\pi = \mathbb{E}_{s' \sim P(\cdot|s, a)}[R(s, a, s') + V^\pi(s')]$.

## 2.2 Justification of Reward Maximisation

The current model for RL, as given by Eq. (1), suggests that the expected value of a single reward function is sufficient for any problem we want our "intelligent agents" to solve. The justification for this idea is deeply rooted in the *von Neumann-Morgenstern (VNM) utility theory* (Von Neumann & Morgenstern, 2007). This theory essentially proves that an agent is *VNM-rational* if and only if there exists a real-valued utility (or, reward) function such that every preference of the agent is characterised by maximising the single expected reward. The VNM utility theorem is the basis for the well-known *expected utility theory* (Schoemaker, 2013), which essentially states that *rationality* can be modelled as maximising an expected value. Specifically, the VNM utility theorem provides both necessary and sufficient conditions under which the expected utility hypothesis holds. In other words, rationality is equivalent to VNM-rationality, and it is safe to assume an intelligent entity will always choose the action with the highest expected utility in any complex scenarios.

---

[3]The agent can only observe part of the full environment state. The partially observable setting is introduced in Definition 7 as a special case of Dec-PODMP.

[4]Such an optimal policy exists as long as the transition function and the reward function are both Markovian and stationary (Feinberg, 2010).

Admittedly, it was accepted long before that some of the assumptions on rationality could be violated by real decision-makers in practice (Gigerenzer & Selten, 2002). In fact, those conditions are rather taken as the "axioms" of rational decision making. In the case of the multi-objective MDP, we are still able to convert multiple objectives into a single-objective MDP with the help of a *scalarisation function* through a two-timescale process; we refer to Roijers et al. (2013) for more details.

## 2.3 Solving Markov Decision Processes

One commonly used notion in MDPs is the (discounted-normalised) occupancy measure $\mu^\pi(s, a)$, which uniquely corresponds to a given policy $\pi$ and vice versa (Syed et al., 2008, Theorem 2), defined by

$$
\begin{aligned}
\mu^\pi(s, a) &= \mathbb{E}_{s_t \sim P, a_t \sim \pi} \left[ (1 - \gamma) \sum_{t \geq 0} \gamma^t \mathbb{1}_{(s_t = s \wedge a_t = a)} \right] . \\
&= (1 - \gamma) \sum_{t \geq 0} \gamma^t \mathbb{P}^\pi(s_t = s, a_t = a),
\end{aligned} \tag{5}
$$

where $\mathbb{1}$ is an indicator function. Note that in Eq. (5), $P$ is the state transitional probability and $\mathbb{P}^\pi$ is the probability of specific state-action pairs when following stationary policy $\pi$. The physical meaning of $\mu^\pi(s, a)$ is that of a probability measure that counts the expected discounted number of visits to the individual admissible state-action pairs. Correspondingly, $\mu^\pi(s) = \sum_a \mu^\pi(s, a)$ is the discounted state visitation frequency, i.e., the stationary distribution of the Markov process induced by $\pi$. With the occupancy measure, we can write Eq. (4) as an inner product of $V^\pi(s) = \frac{1}{1-\gamma} \langle \mu^\pi(s, a), R(s, a) \rangle$. This implies that solving an MDP can be regarded as solving a linear program (LP) of $\max_\mu \langle \mu(s, a), R(s, a) \rangle$, and the optimal policy is then

$$
\pi^*(a|s) = \mu^*(s, a)/\mu^*(s) \tag{6}
$$

However, this method for solving the MDP remains at a textbook level, aiming to offer theoretical insights but lacking practically in the case of a large-scale LP with millions of variables (Papadimitriou & Tsitsiklis, 1987). When the state-action space of an MDP is continuous, LP formulation cannot help solve either.

In the context of optimal control (Bertsekas, 2005), dynamic-programming approaches, such as policy iteration and value iteration, can also be applied to solve for the optimal policy that maximises Eq. (3) & Eq. (4), but these approaches require knowledge of the exact form of the model: the transition function $P(\cdot|s, a)$, and the reward function $R(s, a, s')$ .

On the other hand, in the setting of RL, the agent learns the optimal policy by a trial-and-error process during its interaction with the environment rather than using prior knowledge of the model. The word "learning" essentially means that the agent turns its experience gained during the interaction into knowledge about the model of the environment. Based on the solution target, either the optimal policy or the optimal value function, RL algorithms can be categorised into two types: value-based methods and policy-based methods.

### 2.3.1 Value-Based Methods

For all MDPs with finite states and actions, there exists at least one deterministic stationary optimal policy (Szepesvári, 2010; Sutton & Barto, 1998). Value-based methods are introduced to find the optimal Q-function $Q^*$ that maximises Eq. (3). Correspondingly, the optimal policy can be derived from the Q-function by taking the greedy action of $\pi^* = \arg \max_a Q^*(s, a)$. The classic Q-learning algorithm (Watkins & Dayan, 1992) approximates $Q^*$ by $\hat{Q}$, and updates its value via temporal-difference learning (Sutton, 1988).

$$
\underbrace{\hat{Q}(s_t, a_t)}_{\text{new value}} \leftarrow \underbrace{\hat{Q}(s_t, a_t)}_{\text{old value}} + \underbrace{\alpha}_{\text{learning rate}} \cdot \Bigg( \overbrace{\underbrace{R_t + \gamma \cdot \max_{a \in \mathbb{A}} \hat{Q}(s_{t+1}, a)}_{\text{temporal difference target}} - \underbrace{\hat{Q}(s_t, a_t)}_{\text{old value}}}^{\text{temporal difference error}} \Bigg) \tag{7}
$$

Theoretically, given the Bellman optimality operator $\mathbf{H}^*$, defined by

$$(\mathbf{H}^*Q)(s,a) = \sum_{s'} P(s'|s,a) \left[ R(s,a,s') + \gamma \max_{b \in \mathbb{A}} Q(s,b) \right], \tag{8}$$

we know it is a contraction mapping and the optimal Q-function is the unique[5] fixed point, i.e., $\mathbf{H}^*(Q^*) = Q^*$. The Q-learning algorithm draws random samples of $(s,a,R,s')$ in Eq. (7) to approximate Eq. (8), but is still guaranteed to converge to the optimal Q-function (Szepesvári & Littman, 1999) under the assumptions that the state-action sets are discrete and finite and are visited an infinite number of times. Munos & Szepesvári (2008) extended the convergence result to a more realistic setting by deriving the high probability error bound for an infinite state space with a finite number of samples.

Recently, Mnih et al. (2015) applied neural networks as a function approximator for the Q-function in updating Eq. (7). Specifically, DQN optimises the following equation:

$$\min_{\theta} \mathbb{E}_{(s_t,a_t,R_t,s_{t+1}) \sim \mathcal{D}} \left[ \left( R_t + \gamma \max_{a \in \mathbb{A}} Q_{\theta^-}(s_{t+1},a) - Q_\theta(s_t,a_t) \right)^2 \right]. \tag{9}$$

The neural network parameters $\theta$ is fitted by drawing i.i.d. samples from the replay buffer $\mathcal{D}$ and then updating in a supervised learning fashion. $Q_{\theta^-}$ is a slowly updated target network that helps stabilise training. The convergence property and finite sample analysis of DQN have been studied by Yang et al. (2019b).

### 2.3.2 Policy-Based Methods

Policy-based methods are designed to directly search over the policy space to find the optimal policy $\pi^*$. One can parameterise the policy expression $\pi^* \approx \pi_\theta(\cdot|s)$ and update the parameter $\theta$ in the direction that maximises the cumulative reward $\theta \leftarrow \theta + \alpha \nabla_\theta V^{\pi_\theta}(s)$ to find the optimal policy. However, the gradient will depend on the unknown effects of policy changes on the state distribution. The famous policy gradient (PG) theorem (Sutton et al., 2000) derives an analytical solution that does not involve the state distribution, that is:

$$\nabla_\theta V^{\pi_\theta}(s) = \mathbb{E}_{s \sim \mu^{\pi_\theta}(\cdot), a \sim \pi_\theta(\cdot|s)} \left[ \nabla_\theta \log \pi_\theta(a|s) \cdot Q^{\pi_\theta}(s,a) \right] \tag{10}$$

where $\mu^{\pi_\theta}$ is the state occupancy measure under policy $\pi_\theta$ and $\nabla \log \pi_\theta(a|s)$ is the updating score of the policy. When the policy is deterministic and the action set is continuous, one obtains the deterministic policy gradient (DPG) theorem (Silver et al., 2014) as

$$\nabla_\theta V^{\pi_\theta}(s) = \mathbb{E}_{s \sim \mu^{\pi_\theta}(\cdot)} \left[ \nabla_\theta \pi_\theta(a|s) \cdot \nabla_a Q^{\pi_\theta}(s,a) \big|_{a=\pi_\theta(s)} \right]. \tag{11}$$

A classic implementation of the PG theorem is REINFORCE (Williams, 1992), which uses a sample return $R_t = \sum_{i=t}^{T} \gamma^{i-t} r_i$ to estimate $Q^{\pi_\theta}$. Alternatively, one can use a model of $Q_\omega$ (also called *critic*) to approximate the true $Q^{\pi_\theta}$ and update the parameter $\omega$ via TD learning. This approach gives rise to the famous actor-critic methods (Konda & Tsitsiklis, 2000; Peters & Schaal, 2008). Important variants of actor-critic methods include trust-region methods (Schulman et al., 2015; 2017), PG with optimal baselines (Weaver & Tao, 2001; Zhao et al., 2011), soft actor-critic methods (Haarnoja et al., 2018), and deep deterministic policy gradient (DDPG) methods (Lillicrap et al., 2015).

## 3 Multi-Agent RL

In the multi-agent scenario, much like in the single-agent scenario, each agent is still trying to solve the sequential decision-making problem through a trial-and-error procedure. The difference is that the evolution of the environmental state and the reward function that each agent receives is now determined by all agents' joint actions (see Figure 3). As a result, agents need to take into account and interact with not only the environment but also other learning agents. A decision-making process that involves multiple agents is usually modelled through a stochastic game (Shapley, 1953), also known as a Markov game (Littman, 1994).

---

[5]Note that although the optimal Q-function is unique, its corresponding optimal policies may have multiple candidates.

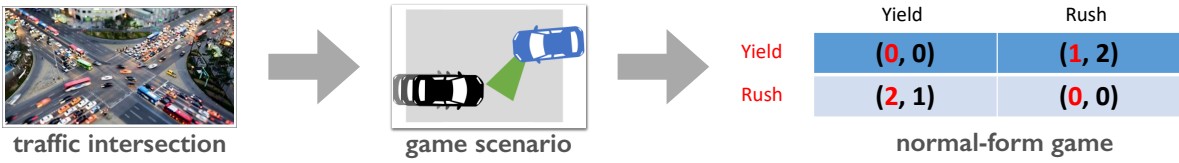

| | Yield | Rush |
|---|---|---|
| Yield | (**0**, 0) | (**1**, 2) |
| Rush | (**2**, 1) | (**0**, 0) |

traffic intersection · game scenario · normal-form game

**Figure 4:** A snapshot of stochastic time in the intersection example. The scenario is abstracted such that there are two cars, with each car taking one of two possible actions: to yield or to rush. The outcome of each joint action pair is represented by a normal-form game, with the reward value for the row player denoted in red and that for the column player denoted in black. The Nash equilibria (NE) of this game are (rush, yield) and (yield, rush). If both cars maximise their own reward selfishly without considering the others, they will end up in an accident.

### 3.1 Problem Formulation: Stochastic Game

**Definition 2 (Stochastic Game)** *A stochastic game can be regarded as a multi-player[6] extension to the MDP in Definition 1. Therefore, it is also defined by a set of key elements* $\langle N, \mathbb{S}, \{\mathbb{A}^i\}_{i \in \{1,...,N\}}, P, \{R^i\}_{i \in \{1,...,N\}}, \gamma \rangle$*.*

- *N: the number of agents, $N = 1$ degenerates to a single-agent MDP, $N \gg 2$ is referred as many-agent cases in this paper.*

- $\mathbb{S}$*: the set of environmental states shared by all agents.*

- $\mathbb{A}^i$*: the set of actions of agent i. We denote $\mathbf{A} := \mathbb{A}^1 \times \cdots \times \mathbb{A}^N$.*

- $P : \mathbb{S} \times \mathbf{A} \to \Delta(\mathbb{S})$*: for each time step $t \in \mathbb{N}$, given agents' joint actions $\mathbf{a} \in \mathbf{A}$, the transition probability from state $s \in \mathbb{S}$ to state $s' \in \mathbb{S}$ in the next time step.*

- $R^i : \mathbb{S} \times \mathbf{A} \times \mathbb{S} \to \mathbb{R}$*: the reward function that returns a scalar value to the $i-$th agent for a transition from $(s, \mathbf{a})$ to $s'$. The rewards have absolute values uniformly bounded by $R_{max}$.*

- $\gamma \in [0, 1]$ *is the discount factor that represents the value of time.*

We use the superscript of $(\cdot^i, \cdot^{-i})$ (for example, $\mathbf{a} = (a^i, a^{-i})$), when it is necessary to distinguish between agent $i$ and all other $N - 1$ opponents.

Ultimately, the stochastic game (SG) acts as a framework that allows simultaneous moves from agents in a decision-making scenario[7]. The game can be described sequentially, as follows: At each time step $t$, the environment has a state $s_t$, and given $s_t$, each agent executes its action $a_t^i$, simultaneously with all other agents. The joint action from all agents makes the environment transition into the next state $s_{t+1} \sim P(\cdot|s_t, \mathbf{a}_t)$; then, the environment determines an immediate reward $R^i(s_t, \mathbf{a}_t, s_{t+1})$ for each agent. As seen in the single-agent MDP scenario, the goal of each agent $i$ is to solve the SG. In other words, each agent aims to find a behavioural policy (or, a mixed strategy[8] in game theory terminology (Osborne & Rubinstein, 1994)), that is, $\pi^i \in \Pi^i : \mathbb{S} \to \Delta(\mathbb{A}^i)$ that can guide the agent to take sequential actions such that the discounted cumulative reward[9] in Eq. (12) is maximised. Here, $\Delta(\cdot)$ is the probability simplex on a set. In

---

[6]Player is a common word used in game theory; agent is more commonly used in machine learning. We do not discriminate between their usages in this work. The same holds for strategy vs policy and utility/payoff vs reward. Each pair refers to the game theory usage vs machine learning usage.

[7]Extensive-form games allow agents to take sequential moves; the full description can be found in (Shoham & Leyton-Brown, 2008, Chapter 5).

[8]A behavioural policy refers to a function map from the history $(s_0, a_0^i, s_1, a_1^i, ..., s_{t-1})$ to an action. The policy is typically assumed to be Markovian such that it depends on only the current state $s_t$ rather than the entire history. A mixed strategy refers to a randomisation over pure strategies (for example, the actions). In SGs, the behavioural policy and mixed policy are exactly the same. In extensive-form games, they are different, but if the agent retains the history of previous actions and states (has perfect recall), each behavioural strategy has a realisation-equivalent mixed strategy, and vice versa (Kuhn, 1950a).

[9]Similar to single-agent MDP, we can adopt the objective of time-average rewards.

game theory, $\pi^i$ is also called a pure strategy (vs a mixed strategy) if $\Delta(\cdot)$ is replaced by a Dirac measure.

$$V^{\pi^i,\pi^{-i}}(s) = \mathbb{E}_{s_{t+1}\sim P(\cdot|s_t,a_t),a^{-i}\sim\pi^{-i}(\cdot|s_t)} \left[ \sum_{t\geq 0} \gamma^t R_t^i (s_t, \boldsymbol{a}_t, s_{t+1}) \Big| a_t^i \sim \pi^i (\cdot \mid s_t), s_0 \right]. \tag{12}$$

Comparison of Eq. (12) with Eq. (4) indicates that the optimal policy of each agent is influenced by not only its own policy but also the policies of the other agents in the game. This scenario leads to fundamental differences in the *solution concept* between single-agent RL and multi-agent RL.

## 3.2   Solving Stochastic Games

An SG can be considered as a sequence of normal-form games, which are games that can be represented in a matrix. Take the original intersection scenario as an example (see Figure 4). A snapshot of the SG at time $t$ (stage game) can be represented as a normal-form game in a matrix format. The rows correspond to the action set $\mathbb{A}^1$ for agent 1, and the columns correspond to the action set $\mathbb{A}^2$ for agent 2. The values of the matrix are the rewards given for each of the joint action pairs. In this scenario, if both agents care only about maximising their own possible reward with no consideration of other agents (the solution concept in a single-agent RL problem) and choose the action to rush, they will reach the outcome of crashing into each other. Clearly, this state is unsafe and is thus sub-optimal for each agent, despite the fact that the possible reward was the highest for each agent when rushing. Therefore, to solve an SG and truly maximise the cumulative reward, each agent must take strategic actions with consideration of others when determining their policies.

Unfortunately, in contrast to MDPs, which have polynomial time-solvable linear-programming formulations, solving SGs usually involves applying Newton's method for solving nonlinear programs. However, there are two special cases of two-player general-sum discounted-reward SGs that can still be written as LPs (Shoham & Leyton-Brown, 2008, Chapter 6.2)[10]. They are as follows:

- *single-controller SG*: the transition dynamics are determined by a single player, i.e., $P(\cdot|\boldsymbol{a},s) = P(\cdot|a^i,s)$ if the i-th index in the vector $\boldsymbol{a}$ is $\boldsymbol{a}[i] = a^i, \forall s \in \mathbb{S}, \forall \boldsymbol{a} \in \mathbb{A}$.

- *separable reward state independent transition (SR-SIT) SG*: the states and the actions have independent effects on the reward function and the transition function depends on only the joint actions, i.e., $\exists \alpha : \mathbb{S} \to \mathbb{R}, \beta : \mathbb{A} \to \mathbb{R}$ such that these two conditions hold: 1) $R^i(s, \boldsymbol{a}) = \alpha(s) + \beta(\boldsymbol{a}), \forall i \in \{1, ..., N\}, \forall s \in \mathbb{S}, \forall \boldsymbol{a} \in \mathbb{A}$, and 2) $P(\cdot|s', \boldsymbol{a}) = P(\cdot|s, \boldsymbol{a}), \forall \boldsymbol{a} \in \mathbb{A}, \forall s, s' \in \mathbb{S}$.

### 3.2.1   Value-Based MARL Methods

The single-agent Q-learning update in Eq. (7) still holds in the multi-agent case. In the $t$-th iteration, for each agent $i$, given the transition data $\left\{(s_t, \boldsymbol{a}_t, R^i, s_{t+1})\right\}_{t\geq 0}$ sampled from the replay buffer, it updates only the value of $Q(s_t, \boldsymbol{a}_t)$ and keeps the other entries of the Q-function unchanged. Specifically, we have

$$Q^i(s_t, \boldsymbol{a}_t) \leftarrow Q^i(s_t, \boldsymbol{a}_t) + \alpha \cdot \left( R^i + \gamma \cdot \mathbf{eval}^i \left( \left\{Q^i(s_{t+1}, \cdot)\right\}_{i\in\{1,...,N\}} \right) - Q^i(s_t, \boldsymbol{a}_t) \right). \tag{13}$$

Compared to Eq. (7), the max operator is changed to $\mathbf{eval}^i\left(\{Q^i(s_{t+1}, \cdot)\}_{i\in\{1,...,N\}}\right)$ in Eq. (13) to reflect the fact that each agent can no longer consider only itself but must **eval**uate the situation of the stage game at time step $t+1$ by considering all agents' interests, as represented by the set of their Q-functions. Then, the optimal policy can be **solved** by $\mathbf{solve}^i\left(\{Q^i(s_{t+1}, \cdot)\}_{i\in\{1,...,N\}}\right) = \pi^{i,*}$. Therefore, we can further write the evaluation operator as

$$\mathbf{eval}^i \left( \left\{Q^i(s_{t+1}, \cdot)\right\}_{i\in\{1,...,N\}} \right) = V^i \left( s_{t+1}, \left\{ \mathbf{solve}^i \left( \{Q^i(s_{t+1}, \cdot)\}_{i\in\{1,...,N\}} \right) \right\}_{i\in\{1,...,N\}} \right). \tag{14}$$

---

[10]According to Filar & Vrieze (2012) [Section 3.5], single-controller SG is solvable in polynomial time only under zero-sum cases rather than general-sum cases, which contradicts the result in Shoham & Leyton-Brown (2008) [Chapter 6.2], and we believe Shoham & Leyton-Brown (2008) made a typo.

In summary, **solve**$^i$ returns agent $i'$s part of the optimal policy at some equilibrium point (not necessarily corresponding to its largest possible reward), and **eval**$^i$ gives agent $i$'s expected long-term reward under this equilibrium, assuming all other agents agree to play the same equilibrium.

### 3.2.2 Policy-Based MARL Methods

The value-based approach suffers from the curse of dimensionality due to the combinatorial nature of multi-agent systems (for further discussion, see Section 4.1). This characteristic necessitates the development of policy-based algorithms with function approximations. Specifically, each agent learns its own optimal policy $\pi^i_{\theta^i} : \mathbb{S} \to \Delta(\mathbb{A}^i)$ by updating the parameter $\theta^i$ of, for example, a neural network. Let $\theta = (\theta^i)_{i \in \{1,...,N\}}$ represent the collection of policy parameters for all agents, and let $\boldsymbol{\pi}_\theta := \prod_{i \in \{1,...,N\}} \pi^i_{\theta^i}(a^i|s)$ be the joint policy. To optimise the parameter $\theta^i$, the policy gradient theorem in Section 2.3.2 can be extended to the multi-agent context. Given agent $i$'s objective function $J^i(\theta) = \mathbb{E}_{s \sim P, \boldsymbol{a} \sim \boldsymbol{\pi}_\theta} \left[ \sum_{t \geq 0} \gamma_t R^i_t \right]$, we have:

$$\nabla_{\theta^i} J^i(\theta) = \mathbb{E}_{s \sim \mu^{\boldsymbol{\pi}_\theta}(\cdot), \boldsymbol{a} \sim \boldsymbol{\pi}_\theta(\cdot|s)} \Big[ \nabla_{\theta^i} \log \pi_{\theta^i}(a^i|s) \cdot Q^{i, \boldsymbol{\pi}_\theta}(s, \boldsymbol{a}) \Big]. \tag{15}$$

Considering a continuous action set with a deterministic policy, we have the multi-agent deterministic policy gradient (MADDPG) (Lowe et al., 2017) written as

$$\nabla_{\theta^i} J^i(\theta) = \mathbb{E}_{s \sim \mu^{\boldsymbol{\pi}_\theta}(\cdot)} \Big[ \nabla_{\theta^i} \log \pi_{\theta^i}(a^i|s) \cdot \nabla_{a_i} Q^{i, \boldsymbol{\pi}_\theta}(s, \boldsymbol{a}) \big|_{\boldsymbol{a} = \boldsymbol{\pi}_\theta(s)} \Big]. \tag{16}$$

Note that in both Eqs. (15) & (16), the expectation over the joint policy $\boldsymbol{\pi}_\theta$ implies that other agents' policies must be observed; this is often a strong assumption for many real-world applications.

### 3.2.3 Solution Concept of the Nash Equilibrium

Game theory plays an essential role in multi-agent learning by offering so-called *solution concepts* that describe the outcomes of a game by showing which strategies will finally be adopted by players. Many types of solution concepts exist for MARL (see Section 4.2), among which the most famous is probably the Nash equilibrium (NE) in non-cooperative game theory (Nash, 1951). The word "non-cooperative" does not mean agents cannot collaborate or have to fight against each other all the time, it merely means that each agent maximises its own reward independently and that agents cannot group into coalitions to make collective decisions.

In a normal-form game, the NE characterises an equilibrium point of the joint strategy profile $(\pi^{1,*}, ..., \pi^{N,*})$, where each agent acts according to their **best response** to the others. The best response produces the optimal outcome for the player once all other players' strategies have been considered. Player $i$'s best response[11] to $\pi^{-i}$ is a set of policies in which the following condition is satisfied.

$$\pi^{i,*} \in \mathbf{Br}(\pi^{-i}) := \Big\{ \arg \max_{\hat{\pi} \in \Delta(\mathbb{A}^i)} \mathbb{E}_{\hat{\pi}^i, \pi^{-i}} \big[ R^i(a^i, a^{-i}) \big] \Big\}. \tag{17}$$

NE states that if all players are perfectly rational, none of them will have a motivation to deviate from their best response $\pi^{i,*}$ given others are playing $\pi^{-i,*}$. Note that NE is defined in terms of the best response, which relies on relative reward values, suggesting that the exact values of rewards are not required for identifying NE. In fact, NE is invariant under positive affine transformations of a players' reward functions. By applying Brouwer's fixed point theorem, Nash (1951) proved that a mixed-strategy NE always exists for any games with a finite set of actions. In the example of driving through an intersection in Figure 4, the NE are $(yield, rush)$ and $(rush, yield)$.

For a SG, one commonly used equilibrium is a stronger version of the NE, called the Markov perfect NE (Maskin & Tirole, 2001), which is defined by:

---

[11]Best responses may not be unique; if a mixed-strategy best response exists, there must be at least one best response that is also a pure strategy.

**Definition 3 (Nash Equilibrium for Stochastic Game)** *A Markovian strategy profile $\boldsymbol{\pi}^* = (\pi^{i,*}, \pi^{-i,*})$ is a Markov perfect NE of a SG defined in Definition 2 if the following condition holds*

$$V^{\pi^{i,*},\pi^{-i,*}}(s) \geq V^{\pi^i,\pi^{-i,*}}(s), \quad \forall s \in \mathbb{S}, \forall \pi^i \in \Pi^i, \forall i \in \{1,...,N\}. \tag{18}$$

"Markovian" means the Nash policies are measurable with respect to a particular partition of possible histories (usually referring to the last state). The word "perfect" means that the equilibrium is also subgame-perfect (Selten, 1965) regardless of the starting state. Considering the sequential nature of SGs, these assumptions are necessary, while still maintaining generality. Hereafter, the Markov perfect NE will be referred to as NE.

A mixed-strategy NE[12] always exists for both discounted and average-reward[13] SGs (Filar & Vrieze, 2012), though they may not be unique. In fact, checking for uniqueness is $NP$-hard (Conitzer & Sandholm, 2002). With the NE as the solution concept of optimality, we can re-write Eq. (14) as:

$$\mathbf{eval}^i_{\mathrm{Nash}}\Big(\big\{Q^i(s_{t+1},\cdot)\big\}_{i\in\{1,...,N\}}\Big) = V^i\Big(s_{t+1}, \Big\{\mathbf{Nash}^i\big(\{Q^i(s_{t+1},\cdot)\}_{i\in\{1,...,N\}}\big)\Big\}_{i\in\{1,...,N\}}\Big). \tag{19}$$

In the above equation, $\mathbf{Nash}^i(\cdot) = \pi^{i,*}$ computes the NE of agent $i$'s strategy, and $V^i\big(s, \{\mathbf{Nash^i}\}_{i\in\{1,...,N\}}\big)$ is the expected payoff for agent $i$ from state $s$ onwards under this equilibrium. Eq. (19) and Eq. (13) form the learning steps of Nash Q-learning (Hu et al., 1998). This process essentially leads to the outcome of a learnt set of optimal policies that reach NE for every single-stage game encountered. In the case when NE is not unique, Nash-Q adopts hand-crafted rules for equilibrium selection (e.g., all players choose the first NE). Furthermore, similar to normal Q-learning, the Nash-Q operator defined in Eq. (20) is also proved to be a contraction mapping, and the stochastic updating rule provably converges to the NE for all states when the NE is unique:

$$(\mathbf{H}^{\mathrm{Nash}}Q)(s,a) = \sum_{s'} P(s'|s,a)\Big[R(s,a,s') + \gamma \cdot \mathbf{eval}^i_{\mathrm{Nash}}\Big(\{Q^i(s_{t+1},\cdot)\}_{i\in\{1,...,N\}}\Big)\Big]. \tag{20}$$

The process of finding a NE in a two-player general-sum game can be formulated as a linear complementarity problem (LCP), which can then be solved using the *Lemke-Howson* algorithm (Shapley, 1974). However, the exact solution for games with more than three players is unknown. In fact, the process of finding the NE is computationally demanding. Even in the case of two-player games, the complexity of solving the NE is $PPAD$-hard (polynomial parity arguments on directed graphs) (Daskalakis et al., 2009; Chen & Deng, 2006); therefore, in the worst-case scenario, the solution could take time that is exponential in relation to the game size. This complexity[14] prohibits any brute force or exhaustive search solutions unless $P = NP$ (see Figure 5). As we would expect, the NE is much more difficult to solve for general SGs, where determining whether a pure-strategy NE exists is $PSPACE$-hard. Even if the SG has a finite-time horizon, the calculation remains $NP$-hard (Conitzer & Sandholm, 2008). When it comes to approximation methods to $\epsilon$-NE, the best known polynomially computable algorithm can achieve $\epsilon = 0.3393$ on bimatrix games (Tsaknakis & Spirakis, 2007); its approach is to turn the problem of finding NE into an optimisation problem that searches for a stationary point.

### 3.2.4 Special Types of Stochastic Games

To summarise the solutions to SGs, one can think of the "master" equation

$$\textbf{Normal-form game solver} + \textbf{MDP solver} = \textbf{Stochastic game solver,}$$

---

[12]Note that this is different from a single-agent MDP, where a single, "pure" strategy optimal policy always exists. A simple example is the rock-paper-scissors game, where none of the pure strategies is the NE and the only NE is to mix between the three equally.

[13]Average-reward SGs entail more subtleties because the limit of Eq. (2) in the multi-agent setting may be a cycle and thus not exist. Instead, NE are proved to exist on a special class of irreducible SGs, where every stage game can be reached regardless of the adopted policy.

[14]The class of $NP$-complete is not suitable to describe the complexity of solving the NE because the NE is proven to always exist (Nash, 1951), while a typical $NP$-complete problem – the travelling salesman problem (TSP), for example – searches for the solution to the question: "Given a distance matrix and a budget B, find a tour that is cheaper than B, or report that none exists (Daskalakis et al., 2009)."

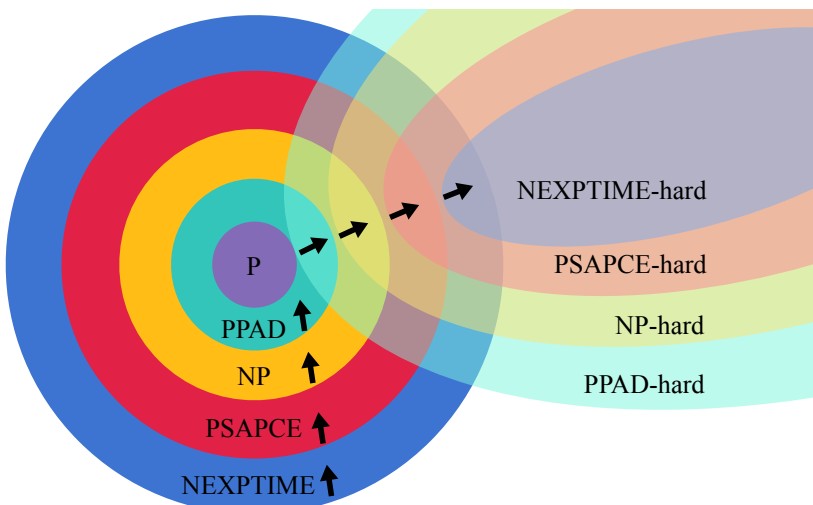

**Figure 5:** The landscape of different complexity classes. Relevant examples are 1) solving the NE in a two-player zero-sum game, $P$-complete (Neumann, 1928), 2) solving the NE in a general-sum game, $PPAD$-hard (Daskalakis et al., 2009), 3) checking the uniqueness of the NE, $NP$-hard (Conitzer & Sandholm, 2002), 4) checking whether a pure-strategy NE exists in a stochastic game, $PSPACE$-hard (Conitzer & Sandholm, 2008), and 5) solving Dec-POMDP, $NEXPTIME$-hard (Bernstein et al., 2002).

which was first summarised by Bowling & Veloso (2000) (in Table 4). The first term refers to solving an equilibrium (NE) for the stage game encountered at every time step. It assumes the transition and reward function is known. The second term refers to applying a RL technique (such as Q-learning) to model the temporal structure in the sequential decision-making process. It assumes to only receive observations of the transition and reward function. The combination of the two gives a solution to SGs, where agents reach a certain type of equilibrium at each and every time step during the game.

Since solving general SGs with NE as the solution concept for the normal-form game is computationally challenging, researchers instead aim to study special types of SGs that have tractable solution concepts. In this section, we provide a brief summary of these special types of games.

**Definition 4 (Special Types of Stochastic Games)** *Given the general form of SG in Definition 2, we have the following special cases:*

- ***normal-form game/repeated game**: $|S| = 1$, see the example in Figure 4. These games have only a single state. Though not theoretically grounded, it is practically easier to solve a small-scale SG.*

- ***identical-interest setting***[15]*: agents share the same learning objective, which we denote as* R*. Since all agents are treated independently, each agent can safely choose the action that maximises its own reward. As a result, single-agent RL algorithms can be applied safely, and a decentralised method developed. Several types of SGs fall into this category.*

  - ***team games/fully cooperative games/multi-agent MDP (MMDP)**: agents are assumed to be homogeneous and interchangeable, so importantly, they share the same reward function*[16]*,* R $= R^1 = R^2 = \cdots = R^N$.

---

[15]In some of the literature on this topic, identical-interest games are equivalent to team games. Here, we refer to this type of game as a more general class of games that involve a shared objective function that all agents collectively optimise, although their individual reward functions can still be different.

[16]In some of the literature on this topic (for example, Wang & Sandholm (2003)), agents are assumed to receive the same expected reward in a team game, which means in the presence of noise, different agents may receive different reward values at a particular moment.

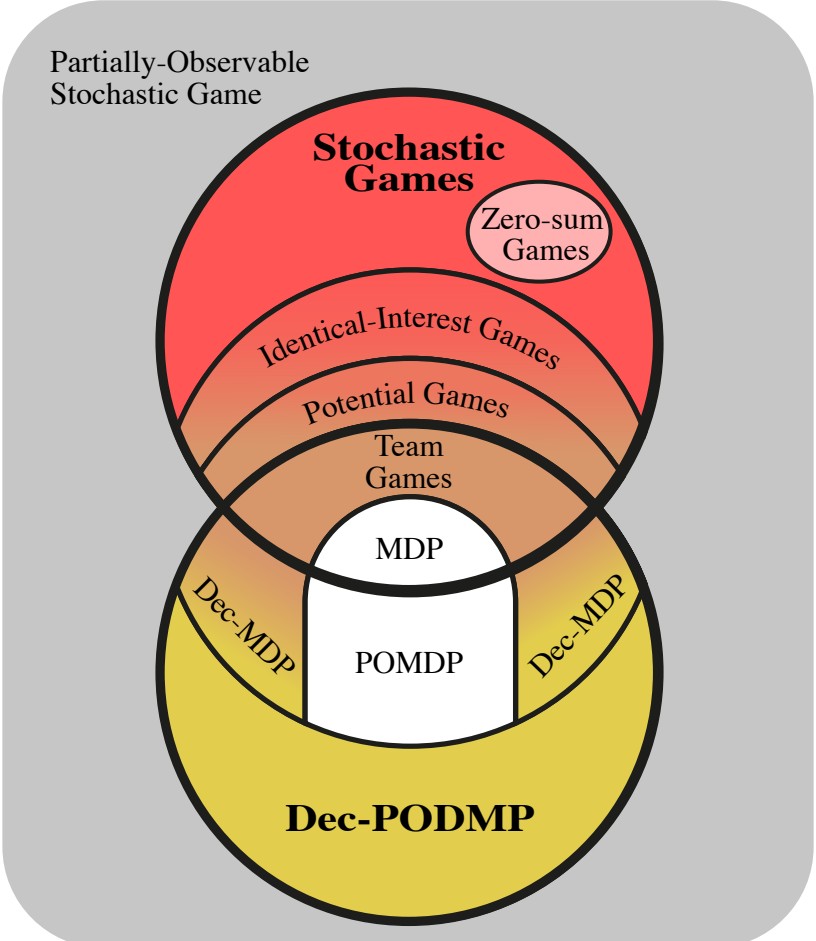

**Figure 6:** Venn diagram of different types of games in the context of POSGs. The intersection of SG and Dec-POMDP is the team game. In the upper-half SG, we have MDP ⊂ team games ⊂ potential games ⊂ identical-interest games ⊂ SGs, and zero-sum games ⊂ SGs. In the bottom-half Dec-POMDP, we have MDP ⊂ team games ⊂ Dec-MDP ⊂ Dec-POMDPs, and MDP ⊂ POMDP ⊂ Dec-POMDP. We refer to Sections (3.2.4 & 3.2.5) for detailed definitions of these games.

- - ***team-average reward games/networked multi-agent MDP (M-MDP):*** *agents can have different reward functions, but they share the same objective,* $\mathsf{R} = \frac{1}{N} \sum_{i=1}^{N} R^i$.
  - ***stochastic potential games:*** *agents can have different reward functions, but their mutual interests are described by a shared potential function* $\mathsf{R} = \phi$, *defined as* $\phi : \mathbb{S} \times \mathbf{A} \to \mathbb{R}$ *such that* $\forall (a^i, a^{-i}), (b^i, a^{-i}) \in \mathbf{A}, \forall i \in \{1, ..., N\}, \forall s \in \mathbb{S}$ *and the following equation holds:*

$$R^i\left(s, \left(a^i, a^{-i}\right)\right) - R^i\left(s, \left(b^i, a^{-i}\right)\right) = \phi\left(s, \left(a^i, a^{-i}\right)\right) - \phi\left(s, \left(b^i, a^{-i}\right)\right). \tag{21}$$

*Games of this type are guaranteed to have a pure-strategy NE (Mguni, 2020). Moreover, potential games degenerate to team games if one chooses the reward function to be a potential function.*

- ***zero-sum setting:*** *agents share opposite interests and act competitively, and each agent optimises against the worst-case scenario. The NE in a zero-sum setting can be solved using a linear program (LP) in polynomial time because of the minimax theorem developed by Neumann (1928). The idea of min-max values is also related to robustness in machine learning. We can subdivide the zero-sum setting as follows:*

- **two-player constant-sum games**: $R^1(s,a,s') + R^2(s,a,s') = c, \forall(s,a,s')$, where $c$ is a constant and usually $c = 0$. For cases when $c \neq 0$, one can always subtract the constant $c$ for all payoff entries to make the game zero-sum.
- **two-team competitive games**: two teams compete against each other, with team sizes $N_1$ and $N_2$. Their reward functions are:

$$\{R^{1,1}, ..., R^{1,N_1}, R^{2,1}, ..., R^{2,N_2}\}.$$

Team members within a team share the same objective of either

$$\mathsf{R}^1 = \sum_{i \in \{1,...,N_1\}} R^{1,i}/N_1,$$

or

$$\mathsf{R}^2 = \sum_{j \in \{1,...,N_2\}} R^{2,j}/N_2,$$

and $\mathsf{R}^1 + \mathsf{R}^2 = 0$.

- **harmonic games**: Any normal-form game can be decomposed into a potential game plus a harmonic game (Candogan et al., 2011). A harmonic game (for example, rock-paper-scissors) can be regarded as a general class of zero-sum games with a harmonic property. Let $\forall \boldsymbol{p} \in \mathbf{A}$ be a joint pure-strategy profile, and let $\mathbf{A}^{[-i]} = \{\boldsymbol{q} \in \mathbf{A} : \boldsymbol{q}^i \neq \boldsymbol{p}^i, \boldsymbol{q}^{-i} = \boldsymbol{p}^{-i}\}$ be the set of strategies that differ from $\boldsymbol{p}$ on agent $i$; then, the harmonic property is:

$$\sum_{i \in \{1,...,N\}} \sum_{\boldsymbol{q} \in \mathbf{A}^{[-i]}} \left(R^i(\boldsymbol{p}) - R^i(\boldsymbol{q})\right) = 0, \quad \forall \boldsymbol{p} \in \mathbf{A}.$$

- **linear-quadratic (LQ) setting**: the transition model follows linear dynamics, and the reward function is quadratic with respect to the states and actions. Compared to a black-box reward function, LQ games offer a simple setting. For example, actor-critic methods are known to facilitate convergence to the NE of zero-sum LQ games (Al-Tamimi et al., 2007). Again, the LQ setting can be subdivided as follows:

  - **two-player zero-sum LQ games**: $Q \in \mathbb{R}^{|\mathbb{S}|}, U^1 \in \mathbb{R}^{|\mathbb{A}^1|}$ and $W^2 \in \mathbb{R}^{|\mathbb{A}^2|}$ are the known cost matrices for the state and action spaces, respectively, while the matrices $A \in \mathbb{R}^{|\mathbb{S}| \times |\mathbb{S}|}, B \in \mathbb{R}^{|\mathbb{S}| \times |\mathbb{A}^1|}, C \in \mathbb{R}^{|\mathbb{S}| \times |\mathbb{A}^2|}$ are usually unknown to the agent:

$$s_{t+1} = As_t + Ba_t^1 + Ca_t^2, \quad s_0 \sim P_0,$$

$$R^1(a_t^1, a_t^2) = -R^2(a_t^1, a_t^2) = -\mathbb{E}_{s_0 \sim P_0}\left[\sum_{t \geq 0} s_t^T Q s_t + a_t^{1^T} U^1 a_t^1 - a_t^{2^T} W^2 a_t^2\right]. \tag{22}$$

  - **multi-player general-sum LQ games**: the difference with respect to a two-player game is that the summation of the agents' rewards does not necessarily equal zero:

$$s_{t+1} = As_t + B\boldsymbol{a}_t, \quad s_0 \sim P_0,$$

$$R^i(\boldsymbol{a}) = -\mathbb{E}_{s_0 \sim P_0}\left[\sum_{t \geq 0} s_t^T Q^i s_t + a_t^{i^T} U^i a_t^i\right]. \tag{23}$$

### 3.2.5 Partially Observable Settings

A partially observable stochastic game (POSG) assumes that agents have no access to the exact environmental state but only an observation of the true state through an observation function. Formally, this scenario is defined by:

**Definition 5 (partially-observable stochastic games)** *A POSG is defined by the set* $\langle N, \mathbb{S}, \{\mathbb{A}^i\}_{i \in \{1,...,N\}}, P, \{R^i\}_{i \in \{1,...,N\}}, \gamma, \underbrace{\{\mathbb{O}^i\}_{i \in \{1,...,N\}}, O}_{\text{newly added}}\rangle$. *In addition to the SG defined in Definition 2, POSGs add the following terms:*

- $\mathbb{O}^i$: *an observation set for each agent i. The joint observation set is defined as* $\mathbb{O} := \mathbb{O}^1 \times \cdots \times \mathbb{O}^N$.

- $O : S \times \mathbb{A} \rightarrow \Delta(\mathbb{O})$: *an observation function* $O(\boldsymbol{o}|\boldsymbol{a}, s')$ *denotes the probability of observing* $\boldsymbol{o} \in \mathbb{O}$ *given the action* $\boldsymbol{a} \in \mathbb{A}$, *and the new state* $s' \in \mathbb{S}$ *from the environment transition.*

*Each agent's policy now changes to* $\pi^i \in \Pi^i : \mathbb{O} \rightarrow \Delta(\mathbb{A}^i)$.

Although the added partial-observability constraint is common in practice for many real-world applications, theoretically it exacerbates the difficulty of solving SGs. Even in the simplest setting of a two-player fully cooperative finite-horizon game, solving a POSG is $NEXP$-hard (see Figure 5), which means it requires super-exponential time to solve in the worst-case scenario (Bernstein et al., 2002). However, the benefits of studying games in the partially observable setting come from the algorithmic advantages. Centralised-training-with-decentralised-execution methods (Oliehoek et al., 2016; Lowe et al., 2017; Foerster et al., 2017a; Rashid et al., 2018; Yang et al., 2020) have achieved many empirical successes, and together with DNNs, they hold great promise.

A POSG is one of the most general classes of games. An important subclass of POSGs is decentralised partially observable MDP (Dec-POMDP), where all agents share the same reward. Formally, this scenario is defined as follows:

**Definition 6 (Dec-POMDP)** *A Dec-POMDP is a special type of POSG defined in Definition 5 with* $R^1 = R^2 = \cdots = R^N$.

Dec-POMDPs are related to single-agent MDPs through the partial observability condition, and they are also related to stochastic team games through the assumption of identical rewards. In other words, versions of both single-agent MDPs and stochastic team games are particular types of Dec-POMDPs (see Figure 6).

**Definition 7 (Special types of Dec-POMDPs)** *The following games are special types of Dec-POMDPs.*

- ***partially observable MDP (POMDP):*** *there is only one agent of interest,* $N = 1$. *This scenario is equivalent to a single-agent MDP in Definition 1 with a partial-observability constraint.*

- ***decentralised MDP (Dec-MDP):*** *the agents in a Dec-MDP have joint full observability. That is, if all agents share their observations, they can recover the state of the Dec-MDP unanimously. Mathematically, we have* $\forall \boldsymbol{o} \in \mathbb{O}, \exists s \in \mathbb{S}$ *such that* $\mathbb{P}(S_t = s | \mathbb{O}_t = \boldsymbol{o}) = 1$.

- ***fully cooperative stochastic games:*** *assuming each agent has full observability,* $\forall i = \{1,...,N\}, \forall o^i \in O^i, \exists s \in \mathbb{S}$ *such that* $\mathbb{P}(S_t = s | \mathbb{O}_t = o^i) = 1$. *The fully-cooperative SG from Definition 4 is a type of Dec-POMDP.*

I conclude Section 3 by presenting the relationships between the many different types of POSGs through a Venn diagram in Figure 6.

### 3.3 Problem Formulation: Extensive-Form Game

An SG assumes that a game is represented as a large table in each stage where the rows and columns of the table correspond to the actions of the two players[17]. Based on the big table, SGs model the situations in which agents act simultaneously and then receive their rewards. Nonetheless, for many real-world games, players take actions alternately. Poker is one class of games in which who plays first has a critical role

---

[17]A multi-player game is represented as a high-dimensional tensor in an SG.

in the players' decision-making process. Games with alternating actions are naturally described by an extensive-form game (EFG) (Osborne & Rubinstein, 1994; Von Neumann & Morgenstern, 1945) through a tree structure. Recently, Kovařík et al. (2019) has made a significant contribution in unifying the framework of EFGs and the framework of POSGs.

Figure 7 shows the game tree of two-player Kuhn poker (Kuhn, 1950b). In Kuhn poker, the dealer has three cards, a King, Queen, and Jack (King>Queen>Jack), each player is dealt one card (the orange nodes in Figure 7), and the third card is put aside unseen. The game then develops as follows.

- Player one acts first; he/she can *check* or *bet.*

- If player one *checks*, then player two decides to *check* or *bet.*

- If player two *checks*, then the higher card wins 1\$ from the other player.

- If player two *bets*, then player one can *fold* or *call.*

- If player one *folds*, then player two wins 1\$ from player one.

- If player one *calls*, then the higher card wins 2\$ from the other player.

- If player one *bets*, then player two decides to *fold* or *call.*

- If player two *folds*, then player one wins 1\$ from player two.

- If player two calls, then the higher card wins 2\$ from the other player.

An important feature of EFGs is that they can handle imperfect information for multi-player decision making. In the example of Kuhn poker, the players do not know which card the opponent holds. However, unlike Dec-POMDP, which also models imperfect information in the SG setting but is intractable to solve, EFG, represented in an equivalent sequence form, can be solved by an LP in polynomial time in terms of game states (Koller & Megiddo, 1992). In the next section, we first introduce EFG and then consider the sequence form of EFG.

**Definition 8 (Extensive-form Game)** *An (imperfect-information) EFG can be described by a tuple of key elements* $\langle N, \mathbb{A}, \mathbb{H}, \mathbb{T}, \{R^i\}_{i \in \{1,...,N\}}, \chi, \rho, P, \{\mathbb{S}^i\}_{i \in \{1,...,N\}} \rangle$.

- *$N$: the number of players. Some EFGs involve a special player called "chance", which has a fixed stochastic policy that represents the randomness of the environment. For example, the chance player in Kuhn poker is the dealer, who distributes cards to the players at the beginning.*

- *$\mathbb{A}$: the (finite) set of all agents' possible actions.*

- *$\mathbb{H}$: the (finite) set of non-terminal choice nodes.*

- *$\mathbb{T}$: the (finite) set of terminal choice nodes, disjoint from $\mathbb{H}$.*

- *$\chi : \mathbb{H} \to 2^{|\mathbb{A}|}$ is the action function that assigns a set of valid actions to each choice node.*

- *$\rho : \mathbb{H} \to \{1, ..., N\}$ is the player indicating function that assigns, to each non-terminal node, a player who is due to choose an action at that node.*

- *$P : \mathbb{H} \times \mathbb{A} \to \mathbb{H} \cup \mathbb{T}$ is the transition function that maps a choice node and an action to a new choice/terminal node such that $\forall h_1, h_2 \in \mathbb{H}$ and $\forall a_1, a_2 \in \mathbb{A}$, if $P(h_1, a_1) = P(h_2, a_2)$, then $h_1 = h_2$ and $a_1 = a_2$.*

- *$R^i : \mathbb{T} \to \mathbb{R}$ is a real-valued reward function for player $i$ on the terminal node. Kuhn poker is a zero-sum game since $R^1 + R^2 = 0$.*

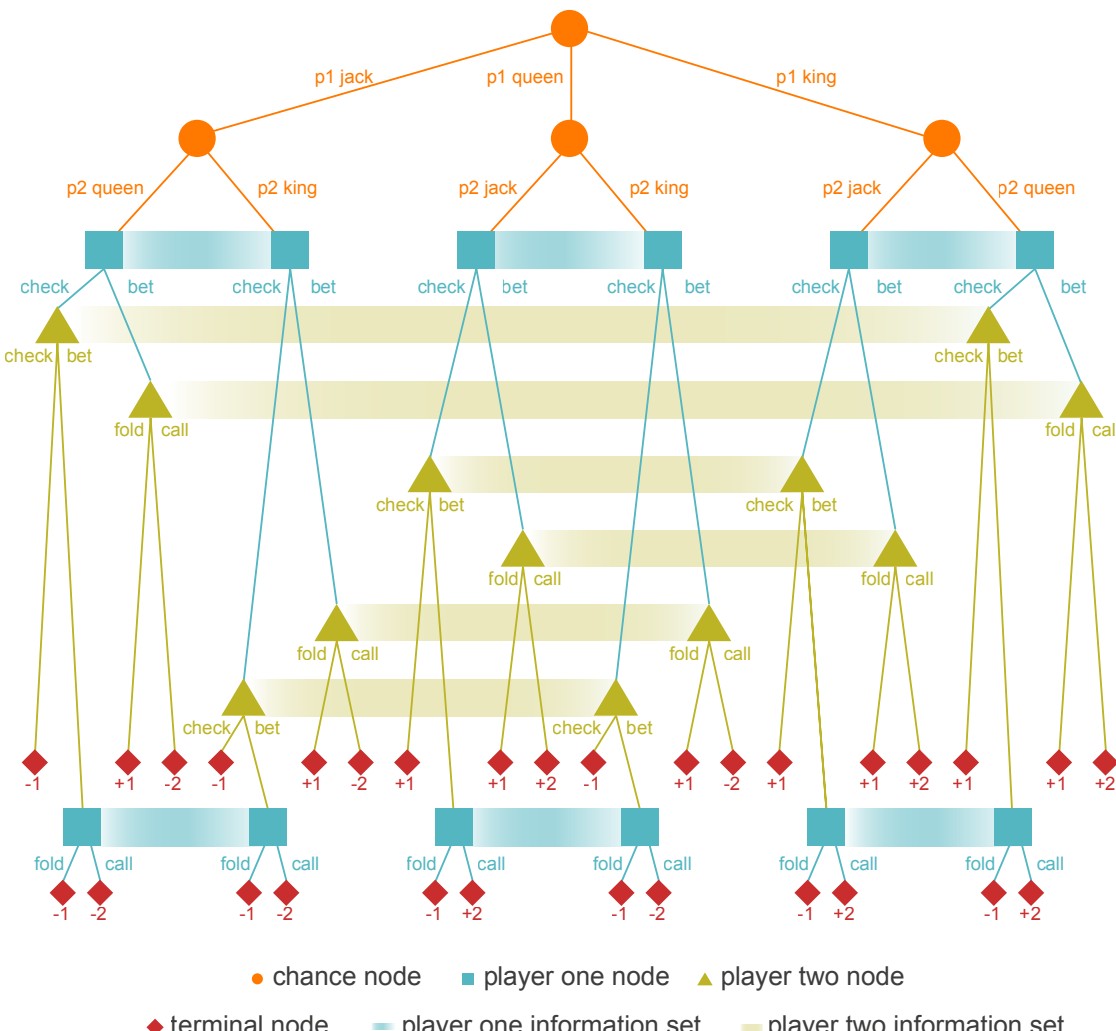

**Figure 7:** Game tree of two-player Kuhn poker. Each node (i.e., circles, squares and rectangles) represents the choice of one player, each edge represents a possible action, and the leaves (i.e., diamond) represent final outcomes over which each player has a reward function (only player one's reward is shown in the graph since Kuhn poker is a zero-sum game). Each player can observe only their own card; for example, when player one holds a Jack, it cannot tell whether player two is holding a Queen or a King, so the choice nodes of player one in each of the two scenarios stay within the same information set.

- $\mathbb{S}^i$: a set of equivalence classes/partitions $\mathbb{S}^i = (S_1^i, ..., S_{k^i}^i)$ for agent $i$ on $\{h \in \mathbb{H} : \rho(h) = i\}$ with the property that $\forall j \in \{1, ..., k^i\}, \forall h, h' \in S_j^i$, we have $\chi(h) = \chi(h')$ and $\rho(h) = \rho(h')$. The set $S_j^i$ is also called an **information state**. The physical meaning of the information state is that the choice nodes of an information state are indistinguishable. In other words, the set of valid actions and agent identities for the choice nodes within an information state are the same; one can thus use $\chi(S_j^i), \rho(S_j^i)$ to denote $\chi(h), \rho(h), \forall h \in S_j^i$.

Inclusion of the information sets in EFG helps to model the imperfect-information cases in which players have only partial or no knowledge about their opponents. In the case of Kuhn poker, each player can only observe their own card. For example, when player one holds a Jack,it cannot tell whether player two is holding a Queen or a King, so the choice nodes of player one under each of the two scenarios (Queen or King) stay within the same information set. Perfect-information EFGs (e.g., GO or chess) are a special case where the information set is a singleton, i.e., $|S_j^i| = 1, \forall j$, so a choice node can be equated to the unique

history that leads to it. Imperfect-information EFGs (e.g., Kuhn poker or Texas hold'em) are those in which there exists $i, j$ such that $|S_j^i| \geq 1$, so the information state can represent more than one possible history. However, with the assumption of perfect recall (described later), the history that leads to an information state is still unique.

### 3.3.1 Normal-Form Representation

A (simultaneous-move) NFG can be equivalently transformed into an imperfect-information EFG[18] (Shoham & Leyton-Brown, 2008) [Chapter 5]. Specifically, since the choices of actions by other agents are unknown to the central agent, this could potentially leads to different histories (triggered by other agents) that can be aggregated into one information state for the central agent.

On the other direction, an imperfect-information EFG can also be transformed into an equivalent NFG in which the pure strategies of each agent $i$ are defined by the Cartesian product $\prod_{S_j^i \in \mathbb{S}^i} \chi(S_j^i)$, which is a complete specification[19] of which action to take at every information state of that agent. In the Kuhn poker example, one pure strategy for player one can be check-bet-check-fold-call-fold; altogether, player one has $2^6 = 64$ pure strategies, corresponding to $3 \times 2^3 = 24$ pure strategies for the chance node and $2^6 = 64$ pure strategies for player two. The mixed strategy of each player is then a distribution over all its pure strategies. In this way, the NE in NFG in Eq. (17) can still be applied to the EFG, and the NE of an EFG can be solved in two steps: first, convert the EFG into an NFG; second, solve the NE of the induced NFG by means of the Lemke-Howson algorithm (Shapley, 1974). If one further restricts the action space to be state-dependent and adopts the discounted accumulated reward at the terminal node, then the EFG recovers to an SG. While the NE of an EFG can be solved through its equivalent normal form, the computational benefit can be achieved by dealing with the extensive form directly; this motivates the adoption of the sequence-form representation of EFGs.

### 3.3.2 Sequence-Form Representation

Solving EFGs via the NFG representation, though universal, is inefficient because the size of the induced NFG is exponential in the number of information states. In addition, the NFG representation does not consider the temporal structure of games. One way to address these problems is to operate on the sequence form of the EFG, also known as the realisation-plan representation, the size of which is only linear in the number of game states and is thus exponentially smaller than that of the NFG. Importantly, this approach enables polynomial-time solutions to EFGs (Koller & Megiddo, 1992).

In the sequence form of EFGs, the main focus shifts from mixed strategies to *behavioural strategies* in which, rather than randomising over complete pure strategies, the agents randomise independently at each information state $S^i \in \mathbb{S}^i$, i.e., $\pi^i : \mathbb{S}^i \to \Delta(\chi(S^i))$. With the help of behavioural strategies, the key insight of the sequence form is that rather than building a player's strategy around the notion of pure strategies that can be exponentially many, one can build the strategy based on the paths in the game tree from the root to each node.

In general, the expressive power of behavioural strategy and mixed strategy are non-comparable. However, if the game has *perfect recall*, which intuitively[20] means that each agent remembers all his historical moves in different information states precisely, then the behavioural strategy and mixed strategy are somehow equivalent. Specifically, suppose all choice nodes in an information state share the same history that led to them (otherwise the agent can distinguish between the choice nodes). In that case, the well-known Kuhn's

---

[18]Note that this transformation is not unique, but they share the same equilibria as the original game. Moreover, this transformation from NFG to EFG does not hold for perfect-information EFGs.

[19]One subtlety of the pure strategy is that it designates a decision at each choice node, regardless of whether it is possible to reach that node given the other choice nodes.

[20]More formally, on the path from the root node to a decision node $h \in S_t^i$ of player $i$, list in chronological order which information sets of $i$ were encountered, i.e., $S_t^i \in \mathbb{S}^i$, and what action player $i$ took at that information set, i.e., $a_t^i \in \chi(S_t^i)$. If one calls this list of $(S_0^i, a_0^i, ..., S_{t-1}^i, a_{t-1}^i, S_t^i)$ the *experience* of player $i$ in reaching node $h \in S_t^i$, then the game has perfect recall if and only if, for all players, any nodes in the same information set have the same experience. In other words, there exists one and only one experience that leads to each information state and the decision nodes in that information state; because of this, all perfect-information EFGs are games of perfect recall.

theorem (Kuhn, 1950a) guarantees that the expressive power of behavioural strategies and that of mixed strategies coincides in the sense that they induce the same probability on outcomes for games of perfect recall. As a result, the set of NE does not change if one considers only behavioural strategies. In fact, the sequence-form representation is primarily useful for describing imperfect-information EFGs of perfect recall, written as:

**Definition 9 (Sequence-form Representation)** *The sequence-form representation of an imperfect-information EFG, defined in Definition 8, of perfect recall is described by* $(N, \boldsymbol{\Sigma}, \{G^i\}_{i \in \{1,...,N\}}, \{\pi^i\}_{i \in \{1,...,N\}}, \{\mu^{\pi^i}\}_{i \in \{1,...,N\}}, \{C^i\}_{i \in \{1,...,N\}})$ *where*

- *$N$: the number of agents, including the chance node, if any, denoted by $c$.*

- *$\boldsymbol{\Sigma} = \prod_{i=1}^N \Sigma^i$: where $\Sigma^i$ is the set of sequences available to agent $i$. A sequence of actions of player $i$, $\sigma^i \in \Sigma^i$, defined by a choice node $h \in \mathbb{H} \cup \mathbb{T}$, is the ordered set of player $i$'s actions that has been taken from the root to node $h$. Let $\varnothing$ be the sequence that corresponds to the root node.*

  *Note that other players' actions are not part of agent $i$'s sequence. In the example of Kuhn poker, $\Sigma^c = \{\varnothing,$ Jack, Queen, King, Jack-Queen, Jack-King, Queen-Jack, Queen-King, King-Jack, King-Queen$\}$, $\Sigma^1 = \{\varnothing,$ check, bet, check-fold, check-bet$\}$, and $\Sigma^2 = \{\varnothing,$ check, bet, fold, call$\}$.*

- *$\pi^i: \mathbb{S}^i \to \Delta\big(\chi(S^i)\big)$ is the behavioural policy that assigns a probability of taking a valid action $a^i \in \chi(S^i)$ at an information state $S^i \in \mathbb{S}^i$. This policy randomises independently over different information states. In the example of Kuhn poker, each player has six information states; their behavioural strategy is therefore a list of six independent probability distributions.*

- *$\mu^{\pi^i}: \Sigma^i \to [0,1]$ is the realisation plan that provides the realisation probability, i.e., $\mu^{\pi^i}(\sigma^i) = \prod_{c \in \sigma^i} \pi^i(c)$, that a sequence $\sigma^i \in \Sigma^i$ would arise under a given behavioural policy $\pi^i$ of player $i$. In the Kuhn poker case, the realisation probability that player one chooses the sequence of check and then fold is $\mu^{\pi^1}(check\text{-}fold) = \pi^1(check) \times \pi^1(fold)$.*

  *Based on the realisation plan, one can recover the underlying behavioural strategy[21] (an idea similar to Eq. (6)). To do so, we need three additional pieces of notation. Let $\mathsf{Seq}: \mathbb{S}^i \to \Sigma_i$ return the sequence $\sigma^i \in \Sigma^i$ that leads to a given information state $S^i \in \mathbb{S}^i$. Since the game assumes perfect recall, $\mathsf{Seq}(S^i)$ is known to be unique. Let $\sigma^i a^i$ denote a sequence that consists of the sequence $\sigma^i$ followed by the single action $a^i$. Since there are many possible actions $a^i$ to choose, let $\mathsf{Ext}: \Sigma^i \to 2^{\Sigma^i}$ denote the set of all possible sequences that extend the given sequence by taking one additional action. It is trivial to see that sequences that include a terminal node cannot be extended, i.e., $\mathsf{Ext}(T) = \emptyset$. Finally, we can write the behavioural policy $\pi^i$ for an information state $S^i$ as*

$$\pi^i\big(a^i \in \chi(S^i)\big) = \frac{\mu^{\pi^i}\big(\mathsf{Seq}(S^i)a^i\big)}{\mu^{\pi^i}\big(\mathsf{Seq}(S^i)\big)}, \quad \forall S^i \in \mathbb{S}^i, \ \forall\big(\mathsf{Seq}(S^i)a^i\big) \in \mathsf{Ext}\big(\mathsf{Seq}(S^i)\big). \tag{24}$$

- *$G^i: \boldsymbol{\Sigma} \to \mathbb{R}$ is the reward function for agent $i$ given by $G^i(\boldsymbol{\sigma}) = R^i(T)$ if a terminal node $T \in \mathbb{T}$ is reached when each player plays their part of the sequence in $\boldsymbol{\sigma} \in \boldsymbol{\Sigma}$, and $G^i(\boldsymbol{\sigma}) = 0$ if non-terminal nodes are reached.*

  *Note that since each payoff that corresponds to a terminal node is stored only once in the sequence-form representation (due to the perfect recall, each terminal node has only one sequence that leads to it), compared to the normal-form representation, which is a Cartesian product over all information sets for each agent and is thus exponential in size, the sequence form is only linear in the size of the EFG. In the example of Kuhn poker, the normal-form representation is a tensor with $64 \times 64 \times 32$ elements, while in the sequence-form representation, since there are $30$ terminal nodes and each node has only one unique sequence leading to it, the payoff tensor has only $30$ elements (plus $\varnothing$ for each player).*

---

[21]Empirically, it is often the case that working on the realisation plan of a behavioural strategy is more computationally friendly than working on the behavioural strategy directly.

- $C^i$: is a set of linear constraints on the realisation probability of $\mu^{\pi^i}$. Under the notations of $\mathsf{Seq}$ and $\mathsf{Ext}$ defined in the bullet points of $\mu^{\pi^i}$, we know the realisation plan must meet the condition that

$$\mu^{\pi^i}(\varnothing) = 1, \quad \mu^{\pi^i}(\sigma^i) \geq 0, \quad \forall \sigma^i \in \Sigma^i$$

$$\mu^{\pi^i}\Big(\mathsf{Seq}(S^i)\Big) = \sum_{\sigma^i \in \mathsf{Ext}\big(\mathsf{Seq}(S^i)\big)} \mu^{\pi^i}(\sigma^i), \quad \forall S^i \in \mathbb{S}^i. \tag{25}$$

The first constraint requires that $\mu^{\pi^i}$ is a proper probability distribution. In addition, the second constraint in Eq. (25) indicates that in order for a realisation plan to be valid to recover a behavioural strategy, at each information state of agent $i$, the probability of reaching that information state must equal the summation of the realisation probabilities of all the extended sequences. In the example of Kuhn poker, we have $C^1$ for player one by $\mu^{\pi^1}(check) = \mu^{\pi^1}(check\text{-}fold) + \mu^{\pi^1}(check\text{-}call)$.

### 3.4 Solving Extensive-Form Games

In the sequence-form EFG, given a joint (behavioural) policy $\boldsymbol{\pi} = (\pi^1, ..., \pi^N)$, we can write the realisation probability of agents reaching a terminal node $T \in \mathbb{T}$, assuming the sequence that leads to the node $T$ is $\boldsymbol{\sigma}_T$, in which each player, including the chance player, follows its own path $\sigma_T^i$ as

$$\mu^{\boldsymbol{\pi}}(\boldsymbol{\sigma}_T) = \prod_{i \in \{1,...,N\}} \mu^{\pi^i}(\sigma_T^i). \tag{26}$$

The expected reward for agent $i$, which covers all possible terminal nodes following the joint policy $\boldsymbol{\pi}$, is thus given by Eq. (27).

$$R^i(\boldsymbol{\pi}) = \sum_{T \in \mathbb{T}} \mu^{\boldsymbol{\pi}}(\boldsymbol{\sigma}_T) \cdot G^i(\boldsymbol{\sigma}_T) = \sum_{T \in \mathbb{T}} \mu^{\boldsymbol{\pi}}(\boldsymbol{\sigma}_T) \cdot R^i(T). \tag{27}$$

If we denote the expected reward by $R^i(\boldsymbol{\pi})$ for simplicity, then the solution concept of NE for the EFG can be written as

$$R^i(\pi^{i,*}, \pi^{-i,*}) \geq R^i(\pi^i, \pi^{-i,*}), \quad \text{for any policy } \pi^i \text{ of agent } i \text{ and for all } i. \tag{28}$$

#### 3.4.1 Perfect-Information Games

Every finite perfect-information EFG has a pure-strategy NE (Zermelo & Borel, 1913). Since players take turns and every agent sees everything that has occurred thus far, it is unnecessary to introduce randomness or mixed strategies into the action selection. However, the NE can be too weak of a solution concept for the EFG. In contrast to that in NFGs, the NE in EFGs can represent *non-credible threats*, which represent the situation where the Nash strategy is not executed as claimed if agents truly reach that decision node. A refinement of the NE in the perfect-information EFG is a *subgame-perfect equilibrium* (SPE). The SPE rules out non-credible threats by picking only the NE that is the best response at every subgame of the original game.

The fundamental principle in solving the SPE is *backward induction*, which identifies the NE from the bottom-most subgame and assumes those NE will be played as considers increasingly large trees. Specifically, backward induction can be implemented through a depth-first search algorithm on the game tree, which requires time that is only linear in the size of the EFG. In contrast, finding NE in NFG is known to be $PPAD$-hard, let alone the NFG representation is exponential in the size of an EFG.

In the case of two-player zero-sum EFGs, backward induction needs to propagate only a single payoff from the terminal node to the root node in the game tree. Furthermore, due to the strictly opposing interests between players, one can further *prune* the backward induction process by recognising that certain subtrees will never be reached in NE, even without examining those subtree nodes[22], which leads to the well-known

---

[22]This occurs, for example, in the case that the worst case of one player in one subgame is better than the best case of that player in another subgame.

Alpha-Beta-Pruning algorithm (Shoham & Leyton-Brown, 2008, Chapter 5.1). For games with very deep game trees, such as Chess or GO, a common approach is to search only nodes up to certain depths and use an approximate value function to estimate those nodes' value without roll outing to the end (Silver et al., 2016).

Finally, backward induction can identify one NE in linear time; yet, it does not provide an effective way to find all NE. A theoretical result suggests that finding all NE in a two-player perfect-information EFG (not necessarily zero-sum) requires $\mathcal{O}(|\mathbb{T}|^3)$, which is still tractable (Shoham & Leyton-Brown, 2008, Theorem 5.1.6).

### 3.4.2   Imperfect-Information Games

By means of the sequence-form representation, one can write the solution of a two-player EFG as a LP. Given a fixed behavioural strategy of player two, in the form of realisation plan $\mu^{\pi^2}$, the best response for player one can be written as

$$\max_{\mu^{\pi^1}} \sum_{\sigma^1 \in \Sigma^1} \mu^{\pi^1}\left(\sigma^1\right) \left( \sum_{\sigma^2 \in \Sigma^2} g^1\left(\sigma^1, \sigma^2\right) \mu^{\pi^2}\left(\sigma^2\right) \right)$$

subject to the constraints in Eq. (25). In NE, player one and player two form a mutual best response. However, if we treat both $\mu^{\pi^1}$ and $\mu^{\pi^2}$ as variables, then the objective becomes nonlinear. The key to address this issue is to adopt the dual form of the LP (Koller & Megiddo, 1996), which is written as

$$
\begin{aligned}
\min \ & v_0 \\
\text{s.t. } & v_{\mathcal{I}(\sigma^1)} - \sum_{I' \in \mathcal{I}\left(\mathsf{Ext}(\sigma^1)\right)} v_{I'} \geq \sum_{\sigma^2 \in \Sigma^2} g^1\left(\sigma^1, \sigma^2\right) \mu^{\pi^2}\left(\sigma^2\right), \quad \forall \sigma^1 \in \Sigma^1 
\end{aligned}
\tag{29}
$$

where $\mathcal{I} : \Sigma^i \to \mathbb{S}^i$ is a mapping function that returns the information set[23] encountered when the final action in $\sigma^i$ was taken. With slight abuse of notation, we let $\mathcal{I}\left(\mathsf{Ext}(\sigma^1)\right)$[24] denote the set of final information states encountered in the set of the extension of $\sigma^i$. The variable $v_0$ represents, given $\mu^{\pi^2}$, player one's expected reward under its own realisation plan $\mu^{\pi^1}$, and $v_{I'}$ can be considered as the part of this expected utility in the subgame starting from information state $I'$. Note that the constraint needs to hold for every sequence of player one.

In the dual form of best response in Eq. (29), if one treats $\mu^{\pi^2}$ as an optimising variable rather than a constant, which means $\mu^{\pi^2}$ must meet the requirements in Eq. (25) to be a proper realisation plan, then the LP formulation for a two-player zero-sum EFG can be written as follows.

$$\min \ v_0 \tag{30}$$

$$\text{s.t. } v_{\mathcal{I}(\sigma^1)} - \sum_{I' \in \mathcal{I}\left(\mathsf{Ext}(\sigma^1)\right)} v_{I'} \geq \sum_{\sigma^2 \in \Sigma^2} g^1\left(\sigma^1, \sigma^2\right) \mu^{\pi^2}\left(\sigma^2\right), \quad \forall \sigma^1 \in \Sigma^1 \tag{31}$$

$$\mu^{\pi^2}\left(\varnothing\right) = 1, \quad \mu^{\pi^2}\left(\sigma^2\right) \geq 0, \quad \forall \sigma^2 \in \Sigma^2 \tag{32}$$

$$\mu^{\pi^2}\left(\mathsf{Seq}(S^2)\right) = \sum_{\sigma^2 \in \mathsf{Ext}\left(\mathsf{Seq}(S^2)\right)} \mu^{\pi^2}\left(\sigma^2\right), \quad \forall S^2 \in \mathbb{S}^2. \tag{33}$$

Player two's realisation plan is now selected to minimise player one's expected utility. Based on the minimax theorem (Von Neumann & Morgenstern, 1945), we know this process will lead to a NE. Notably, though the zero-sum EFG and zero-sum SG (see the formulation in Eq. (51)) both adopt the LP formulation to solve the NE and can be solved in polynomial time, the size of the representation for the game itself is very different. If one chooses first to transform the EFG into an NFG presentation and then solve it by LP, then the time complexity would in fact become exponential in the size of the original EFG.

---

[23]Recall that this information set is unique under the assumption of perfect recall.

[24]Recall that $\mathsf{Ext}(\sigma^1)$ is the set of all possible sequences that extend $\sigma^1$ one step ahead.

The solution to a two-player general-sum EFG can also be formulated using an approach similar to that used for the zero-sum EFG. The difference is that there will be no objective function such as Eq. (30) since in the general-sum context, one agent's reward can no longer be determined based on the other player's reward. The LP with only Eqs. (31 - 33) thus becomes a constraint satisfaction problem. Specifically, one would need to repeat Eqs. (31 - 33) twice to consider each player independently. One final subtlety required in solving the two-player general-sum EFG is that to ensure $v^1$ and $v^2$ are bounded[25], a *complementary slack condition* must be further imposed; we have $\forall \sigma^1 \in \Sigma^1$ (vice versa $\forall \sigma^2 \in \Sigma^2$ for player two):

$$\mu^{\pi^1}(\sigma^1) \left[ \left( v^1_{\mathcal{I}(\sigma^1)} - \sum_{I' \in \mathcal{I}\left(\mathsf{Ext}(\sigma^1)\right)} v^1_{I'} \right) - \left( \sum_{\sigma^2 \in \Sigma^2} g^1\left(\sigma^1, \sigma^2\right) \mu^{\pi^2}\left(\sigma^2\right) \right) \right] = 0. \tag{34}$$

The above condition indicates that for each player, either the sequence $\sigma^i$ is never played, i.e., $\mu^{\pi^i}(\sigma^i) = 0$, or all sequences that are played by that player with positive probability must induce the same expected payoff such that $v^i$ takes arbitrarily large values, thus being bounded. Eqs. (31 - 33), together with Eq. (34), turns the solution to the NE into an LCP problem that can be solved by the generalised Lemke-Howson method (Lemke & Howson, 1964). Although in the worst case, polynomial time complexity cannot be achieved, as can for zero-sum games, this approach is still exponentially faster than running the Lemke-Howson method to solve the NE in a normal-form representation.

For a perfect-information EFG, recall that the SPE is a more informative solution concept than NE. Extending SPE to the imperfect-information scenario is therefore valuable. However, such an extension is non-trivial because a well-defined notion of a subgame is lacking. However, for EFGs with perfect recall, the intuition of subgame perfection can be effectively extended to a new solution concept, named the sequential equilibrium (SE) (Kreps & Wilson, 1982), which is guaranteed to exist and coincides with the SPE if all players in the game have perfect information.

## 4 Grand Challenges of MARL

Compared to single-agent RL, multi-agent RL is a general framework that better matches the broad scope of real-world AI applications. However, due to the existence of multiple agents that learn simultaneously, MARL methods pose more theoretical challenges, in addition to those already present in single-agent RL. Compared to classic MARL settings where there are usually two agents, solving a many-agent RL problem is even more challenging. As a matter of fact, ① **the combinatorial complexity**, ② **the multi-dimensional learning objectives**, and ③ **the issue of non-stationarity** all result in the majority of MARL algorithms being capable of solving games with ④ **only two players**, in particular, two-player zero-sum games. In this section, I will elaborate each of the grand challenge in many-agent RL.

### 4.1 The Combinatorial Complexity

In the context of multi-agent learning, each agent has to consider the other opponents' actions when determining the best response; this characteristic is deeply rooted in each agent's reward function and for example is represented by the joint action $\boldsymbol{a}$ in their Q-function $Q^i(s, \boldsymbol{a})$ in Eq. (13). The size of the joint action space, $|\mathbb{A}|^N$, grows exponentially with the number of agents and thus largely constrains the scalability of MARL methods. Furthermore, the combinatorial complexity is worsened by the fact that solving a NE in game theory is $PPAD$-hard, even for two-player games. Therefore, for multi-player general-sum games (neither team games nor zero-sum games), it is non-trivial to find an applicable solution concept.

One common way to address this issue is by assuming specific factorised structures on action dependency such that the reward function or Q-function can be significantly simplified. For example, a graphical game assumes an agent's reward is affected by only its neighbouring agents, as defined by the graph from (Kearns, 2007). This assumption leads to a polynomial-time solution for the computation of a NE in specific tree graphs (Kearns et al., 2013), though the scope of applications is limited beyond this specific scenario.

---

[25]Since the constraints are linear, they remain satisfied when both $v^1$ and $v^2$ are increased by the same constant to any arbitrarily large values.

Recent progress has also been made toward leveraging particular neural network architectures for Q-function decomposition (Sunehag et al., 2018; Rashid et al., 2018; Yang et al., 2020). In addition to the fact that these methods work only for the team-game setting, the majority of them lack theoretical backing. There remain open questions to answer, such as understanding the representational power (the approximation error) of the factorised Q-functions in a multi-agent task and how factorisation itself can be learnt from scratch.

## 4.2 The Multi-Dimensional Learning Objectives

Compared to single-agent RL, where the only goal is to maximise the learning agent's long-term reward, the learning goals in MARL are naturally multi-dimensional, as the objective of all agents are not necessarily aligned by one metric. Bowling & Veloso (2001; 2002) proposed to classify the goals of the learning task into two types: **rationality** and **convergence**. Rationality ensures an agent takes the best possible response to the opponents when they are stationary, and convergence ensures the learning dynamics eventually lead to a stable policy against a given class of opponents. Reaching both rationality and convergence gives rise to reaching the NE.

In terms of rationality, the NE characterises a fixed point of a joint optimal strategy profile from which no agents would be motivated to deviate as long as they are all perfectly rational. However, in practice, an agent's rationality can easily be bound by either cognitive limitations and/or the tractability of the decision problem. In these scenarios, the rationality assumption can be relaxed to include other types of solution concepts, such as the recursive reasoning equilibrium, which results from modelling the reasoning process recursively among agents with finite levels of hierarchical thinking (for example, an agent may reason in the following way: I believe that you believe that I believe ...) (Wen et al., 2018; 2019); best response against a target type of opponent (Powers & Shoham, 2005b); the mean-field game equilibrium, which describes multi-agent interactions as a two-agent interaction between each agent itself and the population mean (Guo et al., 2019; Yang et al., 2018b;a); evolutionary stable strategies, which describe an equilibrium strategy based on its evolutionary advantage of resisting invasion by rare emerging mutant strategies (Maynard Smith, 1972; Tuyls & Nowé, 2005; Tuyls & Parsons, 2007; Bloembergen et al., 2015); Stackelberg equilibrium (Zhang et al., 2019a), which assumes specific sequential order when agents take decisions; and the robust equilibrium (also called the trembling-hand perfect equilibrium in game theory), which is stable against adversarial disturbance (Li et al., 2019b; Goodfellow et al., 2014b; Yabu et al., 2007).

In terms of convergence, although most MARL algorithms are contrived to converge to the NE, the majority either lack a rigorous convergence guarantee (Zhang et al., 2019b), potentially converge only under strong assumptions such as the existence of a unique NE (Littman, 2001b; Hu & Wellman, 2003), or are provably non-convergent in all cases (Mazumdar et al., 2019a). Zinkevich et al. (2006) identified the non-convergent behaviour of value-iteration methods in general-sum SGs and instead proposed an alternative solution concept to the NE – *cyclic equilibria* – that value-based methods converge to. The concept of no regret (also called the Hannan consistency in game theory (Hansen et al., 2003)), measures convergence by comparison against the best possible strategy in hindsight. This was also proposed as a new criterion to evaluate convergence in zero-sum self-plays (Bowling, 2005; Hart & Mas-Colell, 2001; Zinkevich et al., 2008). In two-player zero-sum games with a non-convex non-concave loss landscape (training GANs (Goodfellow et al., 2014a)), gradient-descent-ascent methods are found to reach a Stackelberg equilibrium (Lin et al., 2019; Fiez et al., 2019) or a local differential NE (Mazumdar et al., 2019b) rather than the general NE.

Finally, although the above solution concepts account for convergence, building a convergent objective for MARL methods with DNNs remains an uncharted area. This is partly because the global convergence result of a single-agent deep RL algorithm, for example, neural policy gradient methods (Wang et al., 2019; Liu et al., 2019) and neural TD learning algorithms (Cai et al., 2019b), has not been extensively studied yet.

## 4.3 The Non-Stationarity Issue

The most well-known challenge of multi-agent learning versus single-agent learning is probably the non-stationarity issue. Since multiple agents concurrently improve their policies according to their own interests, from each agent's perspective, the environmental dynamics become non-stationary and challenging to interpret when learning. This problem occurs because the agent itself cannot tell whether the state transition

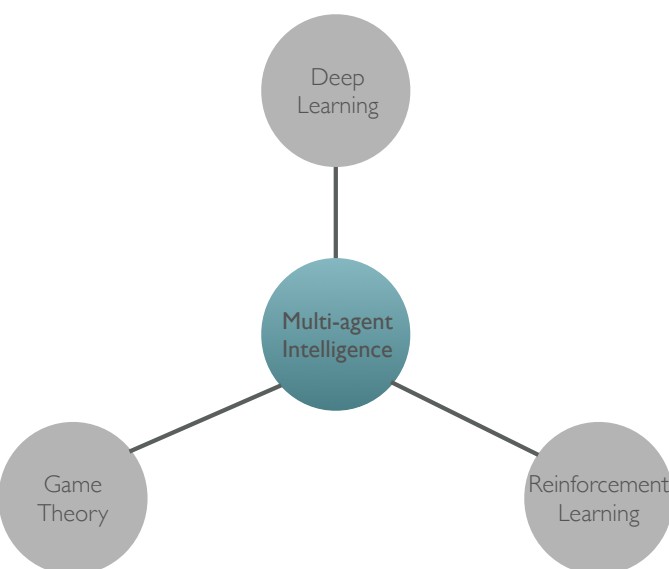

**Figure 8:** The scope of multi-agent intelligence, as described here, consists of three pillars. Deep learning serves as a powerful function approximation tool for the learning process. Game theory provides an effective approach to describe learning outcomes. RL offers a valid approach to describe agents' incentives in multi-agent systems.

– or the change in reward – is an actual outcome due to its own action or if it is due to its opponent's explorations. Although learning independently by completely ignoring the other agents can sometimes yield surprisingly powerful empirical performance (Papoudakis et al., 2020; Matignon et al., 2012), this approach essentially harms the stationarity assumption that supports the theoretical convergence guarantee of single-agent learning methods (Tan, 1993). As a result, the Markovian property of the environment is lost, and the state occupancy measure of the stationary policy in Eq. (5) no longer exists. For example, the convergence result of single-agent policy gradient methods in MARL is provably non-convergent in simple linear-quadratic games (Mazumdar et al., 2019b).

The non-stationarity issue can be further aggravated by TD learning, which occurs with the replay buffer that most deep RL methods currently adopt (Foerster et al., 2017b). In single-agent TD learning (see Eq. (9)), the agent bootstraps the current estimate of the TD error, saves it in the replay buffer, and samples the data in the replay buffer to update the value function. In the context of multi-agent learning, since the value function for one agent also depends on other agents' actions, the bootstrap process in TD learning also requires sampling other agents' actions, which leads to two problems. First, the sampled actions barely represent the full behaviour of other agents' underlying policies across different states. Second, an agent's policy can change during training, so the samples in the replay buffer can quickly become outdated. Therefore, the dynamics that yielded the data in the agent's replay buffer must be constantly updated to reflect the current dynamics in which it is learning. This process further exacerbates the non-stationarity issue.

In general, the non-stationarity issue forbids the reuse of the same mathematical tool for analysing single-agent algorithms in the multi-agent context. However, one exception exists: the identical-interest game in Definition 4. In such settings, each agent can safely perform selfishly without considering other agents' policies since the agent knows the other agents will also act in their own interest. The stationarity is thus maintained, so single-agent RL algorithms can still be applied.

## 4.4 The Scalability Issue when $N \gg 2$

Combinatorial complexity, multi-dimensional learning objectives, and the issue of non-stationarity all result in the majority of MARL algorithms being capable of solving games with only two players, in particular,

**Table 1:** Common assumptions on the level of local knowledge made by MARL algorithms.

| Levels | Assumptions |
|:---:|:---|
| 0 | Each agent observes the reward of his selected action. |
| 1 | Each agent observes the rewards of all possible actions. |
| 2 | Each agent observes others' selected actions. |
| 3 | Each agent observes others' reward values. |
| 4 | Each agent knows others' exact policies. |
| 5 | Each agent knows others' exact reward functions. |
| 6 | Each agent knows the equilibrium of the stage game. |

two-player zero-sum games (Zhang et al., 2019b). As a result, solutions to general-sum settings with more than two agents (for example, the many-agent problem) remain an open challenge. This challenge must be addressed from all three perspectives of multi-agent intelligence (see Figure (8)): game theory, which provides realistic and tractable solution concepts to describe learning outcomes of a many-agent system; RL algorithms, which offer provably convergent learning algorithms that can reach stable and rational equilibria in the sequential decision-making process; and finally deep learning techniques, which provide the learning algorithms expressive function approximators.

## 5 A Survey of MARL Surveys

In this section, I provide a non-comprehensive review of MARL algorithms. To begin, I introduce different taxonomies that can be applied to categorise prior approaches. Given multiple high-quality, comprehensive surveys on MARL methods already exist, a survey of those surveys is provided. Based on the proposed taxonomy, I review related MARL algorithms, covering works on identical interest games, zero-sum games, and games with an infinite number of players. This section is written to be selective, focusing on the algorithms that have theoretical guarantees and less focus on those with only empirical success or those that are purely driven by specific applications.

### 5.1 Taxonomy of MARL Algorithms

One significant difference between the taxonomy of single-agent RL algorithms and MARL algorithms is that in the single-agent setting, since the problem is unanimously defined, the taxonomy is driven mainly by the type of solution (Kaelbling et al., 1996; Li, 2017), for example, model-free vs model-based, on-policy vs off-policy, TD learning vs Monte-Carlo methods. By contrast, in the multi-agent setting, due to the existence of multiple learning objectives (see Section 4.2), the taxonomy is driven mainly by the type of problem rather than the solution. In fact, asking the right question for MARL algorithms is itself a research problem, which is referred to as the problem problem (Balduzzi et al., 2018b; Shoham et al., 2007).

**Based on Stage Games Types.** Since the solution concept varies considerably according to the game type, one principal component of the MARL taxonomy is the nature of stage games. A common division[26] includes team games (more generally, potential games), zero-sum games (more generally, harmonic games), and a mixed setting of the two games, namely, general-sum games. Other types of "exotic" games, such as potential games (Monderer & Shapley, 1996) and mean-field games (Lasry & Lions, 2007), that originate from non-game-theoretical research domains exist and have recently attracted tremendous attention. Based on the type of stage game, the taxonomy can be further enriched by how many times they are played. A repeated game is where one stage game is played repeatedly without considering the state transition. An SG is a sequence of stage games, which can be infinitely long, with the order of the games to play determined by the state-transition probability. Since solving a general-sum SG is at least $PSPACE$-hard (Conitzer &

---

[26]Such a division is complementary because any multi-player normal-form game can be decomposed into a potential game plus a harmonic game (Candogan et al., 2011) (also see Definition 4); in the two-player case, it corresponds to a team game plus a zero-sum game.

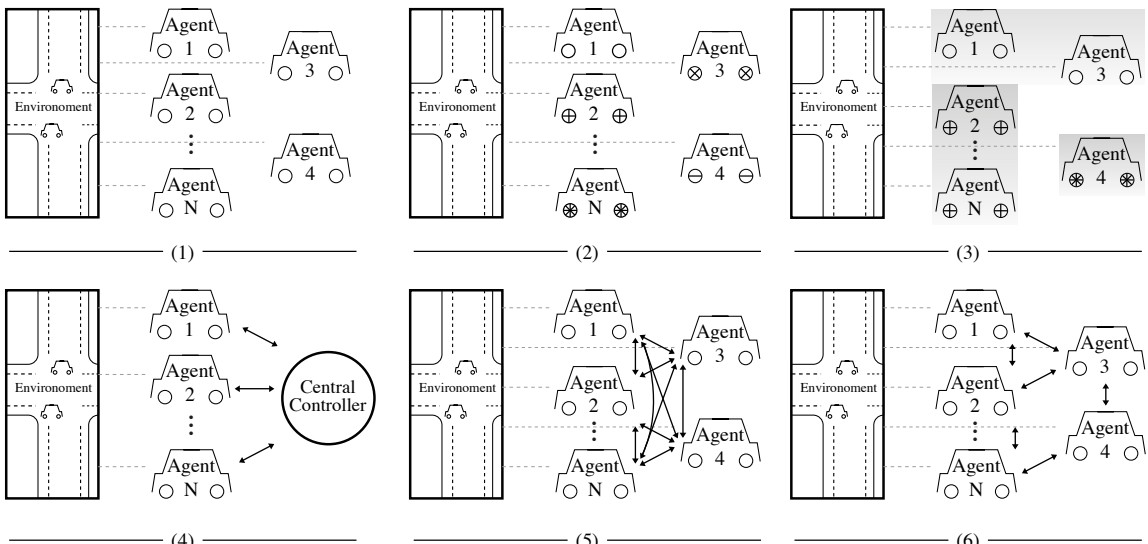

**Figure 9:** Common learning paradigms of MARL algorithms. (1) Independent learners with shared policy. (2) Independent learners with independent policies (i.e., denoted by the difference in wheels). (3) Independent learners with shared policy within a group. (4) One central controller controls all agents: agents can exchange information with any other agents at any time. (5) Centralised training with decentralised execution (CTDE): only during training, agents can exchange information with others; during execution, they act independently. (6) Decentralised training with networked agents: during training, agents can exchange information with their neighbours in the network; during execution, they act independently.

Sandholm, 2002), MARL algorithms usually have a clear boundary on what types of game they can solve. For general-sum games, there are few MARL algorithms that have a provable convergence guarantee without strong, even unrealistic, assumptions (e.g., the NE is unique) (Shoham et al., 2007; Zhang et al., 2019b).

**Based on Level of Local Knowledge.** The assumption on the level of local knowledge, i.e., what agents can and cannot know during training and execution time, is another major component to differentiate MARL algorithms. Having access to different levels of local knowledge leads to different local behaviours by agents and various levels of difficulty in developing theoretical analysis. I list the common assumptions that most MARL methods adopt in Table (1). The seven levels of assumptions are ranked based on how strong, or unrealistic, they are in general. The two extreme cases are that the agent can observe nothing apart from itself and that the agent knows the equilibrium point, i.e., the direct answer of the game. Among the multiple levels, the nuance between level 0 and level 1, which has been mainly investigated in the online learning literature, is referred to as the *bandit* setting vs *full-information* setting. In addition, knowledge of the agents' exact policy/reward function forms is a much stronger assumption than being able to observe their sampled actions/rewards. In fact, knowing the exact policy parameters of other agents in most cases are only possible in simulations. Furthermore, from an applicability perspective, observing other agents' rewards is also more unrealistic than observing their actions.

**Based on Learning Paradigms.** In addition to various levels of local knowledge, MARL algorithms can be classified based on the learning paradigm, as shown in Figure 9. For example, the 4th learning paradigm addresses multi-agent problems by building a single-agent controller, which takes the joint information from all agents as inputs and outputs the joint policies for all agents. In this paradigm, agents can exchange any information with any other opponent through the central controller. The information that can be exchanged depends on the assumptions about the level of local knowledge described in Table (1), e.g., private observations from each agent, the reward value, or policy parameters for each agent. The 5th learning paradigm allows agents to exchange information with other agents only during training; during

**Table 2:** Summary of the five agendas for multi-agent learning research Shoham et al. (2007).

| ID | Agenda | Description |
|----|--------|-------------|
| 1 | COMPUTATIONAL | To develop efficient methods that can compute solution concepts of the game. Examples: Berger (2007); Leyton-Brown & Tennenholtz (2005) |
| 2 | DESCRIPTIVE | To develop formal models of learning that agree with the behaviours of people/animals/organisations. Examples: Erev & Roth (1998); Camerer et al. (2002) |
| 3 | NORMATIVE | To determine which sets of learning rules are in equilibrium with each other. For example, we can ask if fictitious play and Q-learning can reach equilibrium with each other in a repeated prisoner's dilemma game. |
| 4 | PRESCRIPTIVE, COOPERATIVE | To develop distributed learning algorithms for team games. In this agenda, there is rarely a role for equilibrium analysis since the agents have no motivation to deviate from the prescribed algorithm. Examples: Claus & Boutilier (1998b) |
| 5 | PRESCRIPTIVE, NON-COOPERATIVE | To develop effective methods for obtaining a "high reward" in a given environment, for example, an environment with a selected class of opponents. Examples: Powers & Shoham (2005a;b) |

execution, each agent has to act in a decentralised manner, making decisions based on its own observations only. The 6th paradigm can be regarded as a particular case of Paradigm 5 in that agents are assumed to be interconnected via a (time-varying) network such that information can still spread across the whole network if agents communicate with their neighbours. The most general case is Paradigm 2, where agents are fully decentralised, with no information exchange of any kind allowed at any time, and each agent executes its own policy. Relaxation of Paradigm 2 yields the 1st and the 3rd paradigms, where the agents, although they cannot exchange information, share a single set of policy parameters, or, within a pre-defined group, share a single set of policy parameters.

**Based on Five AI Agendas.** In order for MARL researchers to be specific about the problem being addressed and the associated evaluation criteria, Shoham et al. (2007) identified five coherent agendas for MARL studies, each of which has a clear motivation and success criterion. Though proposed more than a decade ago, these five distinct goals are still useful in evaluating and categorising recent contributions. I, therefore, choose to incorporate them into the taxonomy of MARL algorithms.

## 5.2 A Survey of Surveys

A multi-agent system (MAS) is a generic concept that could refer to many different domains of research across different academic subjects; general overviews are given by Weiss (1999), Wooldridge (2009), and Shoham & Leyton-Brown (2008). Due to the many possible ways of categorising multi-agent (reinforcement) learning algorithms, it is impossible to have a single survey that includes all relevant works considering all directions of categorisations. In the past two decades, there has been no lack of survey papers that summarise the current progress of specific categories of multi-agent learning research. In fact, there are so many that these surveys themselves deserve a comprehensive review. Before proceeding to review MARL algorithms based on the proposed taxonomy in Section 5.1, in this section, I provide an overview of relevant surveys that study multi-agent systems from the machine learning, in particular, the RL, perspective.

One of the earliest studies that surveyed MASs in the context of machine learning/AI was published by Stone & Veloso (2000): the research works up to that time were summarised into four major scenarios considering

whether agents were homogeneous or heterogeneous and whether or not agents were allowed to communicate with each other. Shoham et al. (2007) considered the game theory and RL perspective and introspectively asked the question of "if multi-agent learning is the answer, what is the question?". Upon failing to find a single answer, Shoham et al. (2007) proposed the famous five AI agendas for future research work to address. Stone (2007) tried to answer Shoham's question by emphasising that MARL can be more broadly framed than through game theoretic terms, and he noted that how to apply the MARL technique remains an open question, rather than being an answer, in contrast to the suggestion of Shoham et al. (2007). The survey of Tuyls & Weiss (2012) also reflected on Stone's viewpoint; they believed that the entanglement of only RL and game theory is too narrow in its conceptual scope, and MARL should embrace other ideas, such as transfer learning (Taylor & Stone, 2009), swarm intelligence (Kennedy, 2006), and co-evolution (Tuyls & Parsons, 2007).

Panait & Luke (2005) investigated the cooperative MARL setting; instead of considering only reinforcement learners, they reviewed learning algorithms based on the division of *team learning* (i.e., applying a single learner to search for the optimal joint behaviour for the whole team) and *concurrent learning* (i.e., applying one learner per agent), which includes broader areas of evolutionary computation, complex systems, etc. Matignon et al. (2012) surveyed the solutions for fully-cooperative games only; in particular, they focused on evaluating independent RL solutions powered by Q-learning and its many variants. Jan't Hoen et al. (2005) conducted an overview with a similar scope; moreover, they extended the work to include fully competitive games in addition to fully cooperative games. Buşoniu et al. (2010), to the best of my knowledge, presented the first comprehensive survey on MARL techniques, covering both value iteration-based and policy search-based methods, together with their strengths and weaknesses. In their survey, they considered not only fully cooperative or competitive games but also the effectiveness of different algorithms in the general-sum setting. Nowé et al. (2012), in the 14th chapter, addressed the same topic as Buşoniu et al. (2010) but with a much narrower coverage of multi-agent RL algorithms.

Tuyls & Nowé (2005) and Bloembergen et al. (2015) both surveyed the dynamic models that have been derived for various MARL algorithms and revealed the deep connection between evolutionary game theory and MARL methods. We refer to Table 1 in Tuyls & Nowé (2005) for a summary of this connection.

Hernandez-Leal et al. (2017) provided a different perspective on the taxonomy of how existing MARL algorithms cope with the issue of non-stationarity induced by opponents. On the basis of the opponent and environment characteristics, they categorised the MARL algorithms according to the type of opponent modelling.

Da Silva & Costa (2019) introduced a new perspective of reviewing MARL algorithms based on how knowledge is reused, i.e., transfer learning. Specifically, they grouped the surveyed algorithms into *intra-agent* and *inter-agent* methods, which correspond to the reuse of knowledge from experience gathered from the agent itself and that acquired from other agents, respectively.

Most recently, deep MARL techniques have received considerable attention. Nguyen et al. (2020) surveyed how deep learning techniques were used to address the challenges in multi-agent learning, such as partial observability, continuous state and action spaces, and transfer learning. OroojlooyJadid & Hajinezhad (2019) reviewed the application of deep MARL techniques in fully cooperative games: the survey on this setting is thorough. Hernandez-Leal et al. (2019) summarised how the classic ideas from traditional MAS research, such as emergent behaviour, learning communication, and opponent modelling, were incorporated into deep MARL domains, based on which they proposed a new categorisation for deep MARL methods. Zhang et al. (2019b) performed a selective survey on MARL algorithms that have theoretical convergence guarantees and complexity analysis. To the best of my knowledge, their review is the only one to cover more advanced topics such as decentralised MARL with networked agents, mean-field MARL, and MARL for stochastic potential games.

On the application side, Müller & Fischer (2014) surveyed 152 real-world applications in various sectors powered by MAS techniques. Campos-Rodriguez et al. (2017) reviewed the application of multi-agent techniques for automotive industry applications, such as traffic coordination and route balancing. Derakhshan & Yousefi (2019) focused on real-world applications for wireless sensor networks, Shakshuki & Reid (2015) studied multi-agent applications for the healthcare industry, and Kober et al. (2013) investigated the appli-

cation of robotic control and summarised profitable RL approaches that can be applied to robots in the real world.

# 6 Learning in Identical-Interest Games

The majority of MARL algorithms assume that agents collaborate with each other to achieve shared goals. In this setting, agents are usually considered homogeneous and play an interchangeable role in the environmental dynamics. In a two-player normal-form game or repeated game, for example, this means the payoff matrix is symmetrical.

## 6.1 Stochastic Team Games

One benefit of studying identical interest games is that single-agent RL algorithms with a theoretical guarantee can be safely applied. For example, in the team game[27] setting, since all agents' rewards are always the same, the Q-functions are identical among all agents. As a result, one can simply apply the single-agent RL algorithms over the joint action space $\boldsymbol{a} \in \mathbb{A}$, equivalently, Eq. (14) can be written as

$$\mathbf{eval}^i\Big(\big\{Q^i(s_{t+1}, \cdot)\big\}_{i \in \{1, \ldots, N\}}\Big) = V^i\Big(s_{t+1}, \arg\max_{\boldsymbol{a} \in \mathbb{A}} Q^i\big(s_{t+1}, \boldsymbol{a}\big)\Big). \tag{35}$$

Littman (1994) first studied this approach in SGs. However, one issue with this approach is that when multiple equilibria exist (e.g., a normal-form game with reward $R = \begin{bmatrix} 0,0 & 2,2 \\ 2,2 & 0,0 \end{bmatrix}$), unless the selection process is coordinated among agents, the agents' optimal policy can end up with a worse scenario even though their value functions have reached the optimal values. To address this issue, Claus & Boutilier (1998a) proposed to build belief models about other agents' policies. Similar to fictitious play (Berger, 2007), each agent chooses actions in accordance with its belief about the other agents. Empirical effectiveness, as well as convergence, have been reported for repeated games; however, the convergent equilibrium may not be optimal. In solving this problem, Wang & Sandholm (2003) proposed optimal adaptive learning (OAL) methods that provably converge to the optimal NE almost surely in any team SG. The main novelty of OAL is that it learns the game structure by building so-called *weakly acyclic games* that eliminate all the joint actions with sub-optimal NE values and then applies adaptive play (Young, 1993) to address the equilibrium selection problem for weakly acyclic games specifically. Following this approach, Arslan & Yüksel (2016) proposed decentralised Q-learning algorithms that, under the help of two-timescale analysis (Leslie et al., 2003), converge to an equilibrium policy for weakly acyclic SGs. To avoid sub-optimal equilibria for weakly acyclic SGs, Yongacoglu et al. (2019) further refined the decentralised Q-learners and derived theorems with stronger almost-surely convergence guarantees for optimal policies.

### 6.1.1 Solutions via Q-function Factorisation

Another vital reason that team games have been repeatedly studied is that solving team games is a crucial step in building distributed AI (DAI) (Huhns, 2012; Gasser & Huhns, 2014). The logic is that if each agent only needs to maintain the Q-function of $Q^i(s, a^i)$, which depends on the state and local action $a^i$, rather than joint action $\boldsymbol{a}$, then the combinatorial nature of multi-agent problems can be avoided. Unfortunately, Tan (1993) previously noted that such independent Q-learning methods do not converge in team games. Lauer & Riedmiller (2000) reported similar negative results; however, when the state transition dynamics are deterministic, independent learning through distributed Q-learning can still obtain a convergence guarantee. No additional expense is needed in comparison to the non-distributed case for computing the optimal policies.

Factorised MDPs (Boutilier et al., 1999) are an effective way to avoid exponential blowups. For a coordination task, if the joint-Q function can be naturally written as

$$Q = Q^1(a^1, a^2) + Q^2(a^2, a^4) + Q^3(a^1, a^3) + Q^4(a^3, a^4),$$

---

[27]The terms Markov team games, stochastic team games, and dynamic team games are interchangeably used across different domains of the literature.

then the nested structure can be exploited. For example, $Q^1$ and $Q^3$ are irrelevant in finding the optimal $a^4$; thus, given $a^4$, $Q^1$ becomes irrelevant for optimising $a^3$. Given $a^3, a^4$, one can then optimise $a^1, a^2$. Inspired by this result, Guestrin et al. (2002b;a); Kok & Vlassis (2004) studied the idea of *coordination graphs*, which combine value function approximation with a message-passing scheme by which agents can efficiently find the globally optimal joint action.

However, the coordination graph may not always be available in real-world applications; thus, the ideal approach is to let agents learn the Q-function factorisation from the tasks automatically. Deep neural networks are an effective way to learn such factorisations. Specifically, the scope of the problem is then narrowed to the so-called *decentralisable tasks* in the Dec-POMDP setting, that is, $\exists \{Q_i\}_{i \in \{1,...,N\}} \; \forall o \in \mathbb{O}, a \in \mathbb{A}$, the following condition holds.

$$\arg\max_{a} Q^{\pi}(o, a) = \begin{bmatrix} \arg\max_{a^1} Q^1(o^1, a^1) \\ \vdots \\ \arg\max_{a^N} Q^N(o^N, a^N) \end{bmatrix}. \tag{36}$$

Eq. (36) suggests that a task is decentralisable only if the local maxima on the individual value function per every agent amounts to the global maximum on the joint value function. Different structural constraints, enforced by particular neural architectures, have been proposed to satisfy this condition. For example, VDN (Sunehag et al., 2018) maintains an additivity structure by making $Q^{\pi}(o, a) := \sum_{i=1}^{N} Q^i(o^i, a^i)$. QMIX (Rashid et al., 2018) adopts a monotonic structure by means of a mixing network to ensure $\frac{\partial Q^{\pi}(o,a)}{\partial Q^i(o^i,a^i)} \geq 0, \forall i \in \{1,...,N\}$. QTRAN (Son et al., 2019) introduces a more rigorous learning objective on top of QMIX that proves to be a sufficient condition for Eq. (36). However, these structure constraints heavily depend on specially designed neural architectures, which makes understanding the representational power (i.e., the approximation error) of the above methods almost infeasible. Another drawback is that the structure constraint also damages agents' efficient exploration during training. To mitigate these issues, Yang et al. (2020) proposed Q-DPP, which eradicates the structure constraints by approximating the Q-function through a *determinantal point process (DPP)* (Kulesza et al., 2012). DPP pushes agents to explore and acquire diverse behaviours; consequently, it leads to natural decomposition of the joint Q-function with no need for *a priori* structure constraints. In fact, VDN/QMIX/QTRAN prove to be the exceptional cases of Q-DPP.

### 6.1.2 Solutions via Multi-Agent Soft Learning

In single-agent RL, the process of finding the optimal policy can be equivalently transformed into a probabilistic inference problem on a graphical model (Levine, 2018). The pivotal insight is that by introducing an additional binary random variable $P(\mathcal{O} = 1|s_t, a_t) \propto \exp(R(s_t, a_t))$, which denotes the *optimality* of the state-action pair at time step $t$, one can draw an equal connection between searching the optimal policies by RL methods and computing the marginal probability of $p(\mathcal{O}_t^i = 1)$ by probabilistic inference methods, such as message passing or variational inference (Blei et al., 2017). This equivalence between optimal control and probabilistic inference also holds in the multi-agent setting (Wen et al., 2018; Grau-Moya et al., 2018; Tian et al., 2019; Wen et al., 2019; Shi et al., 2019). In the context of SG (see the red part in Figure 10), the optimality variable for each agent $i$ is defined by $p\left(\mathcal{O}_t^i = 1|\mathcal{O}_t^{-i} = 1, \tau_t^i\right) \propto \exp\left(r^i\left(s_t, a_t^i, a_t^{-i}\right)\right)$, which implies that the optimality of trajectory $\tau_t^i = (s_0, a_0^i, a_0^{-i}, ..., s_t, a_t^i, a_t^{-i})$ depends on whether agent $i$ acts according to its best response against other agents, and $\mathcal{O}_t^{-i} = 1$ indicates that all other agents are perfectly rational and attempt to maximise their rewards. Therefore, from each agent's perspective, its objective becomes maximising $p(\mathcal{O}_{1:T}^i = 1|\mathcal{O}_{1:T}^{-i} = 1)$. As we assume no knowledge of the optimal policies and the model of the environment, we treat states and actions as latent variables and apply variational inference (Blei et al., 2017) to approximate this objective, which leads to

$$\max_{\theta^i} \; J(\pi_\theta) = \log p(\mathcal{O}_{1:T}^i = 1|\mathcal{O}_{1:T}^{-i} = 1)$$

$$\geq \sum_{t=1}^{T} \mathbb{E}_{s \sim P(\cdot|s,a), a \sim \pi_\theta(s)} \left[ r^i\left(s_t, a_t^i, a_t^{-i}\right) + \mathcal{H}\left(\pi_\theta(a_t^i, a_t^{-i}|s_t)\right) \right]. \tag{37}$$

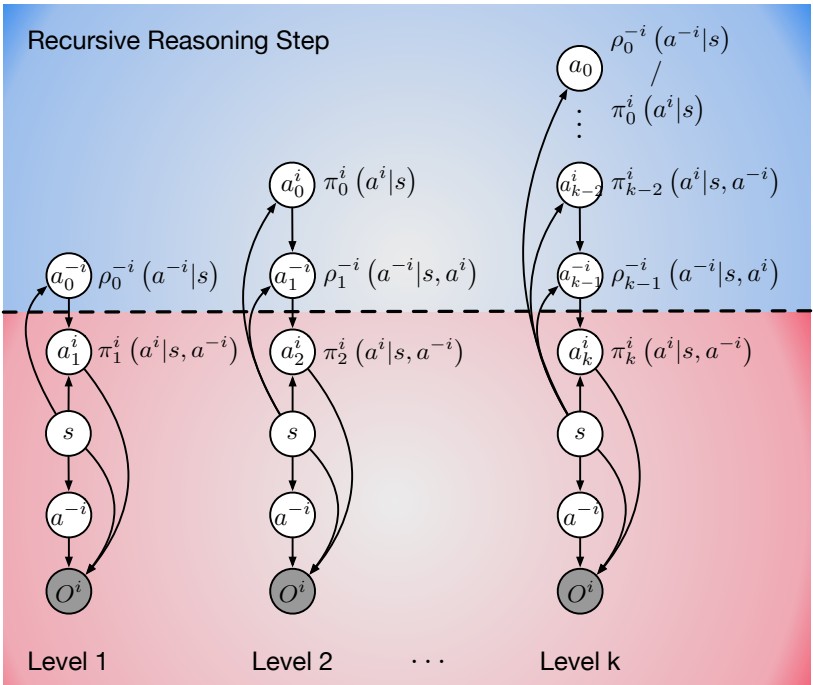

**Figure 10:** Graphical model of the *level-k* reasoning model (Wen et al., 2019). The red part is the equivalent graphical model for the multi-agent learning problem. The blue part corresponds to the recursive reasoning steps. Subscript $a_*$ stands for the level of thinking, not the time step. The opponent policies are approximated by $\rho^{-i}$. The omitted *level-0* model considers opponents that are fully randomised. Agent $i$ rolls out the recursive reasoning about opponents in its mind (blue area). In the recursion, agents with higher-level beliefs take the best response to the lower-level agents. The higher-level models conduct all the computations that the lower-level models have done, e.g., the *level-2* model contains the *level-1* model by integrating out $\pi_0^i(a^i|s)$.

One major difference from traditional RL is the additional entropy term[28] in Eq. (37). Under this new objective, the value function is written as $V^i(s) = \mathbb{E}_{\boldsymbol{\pi}_\theta}\Big[Q^i(s_t, a_t^i, a_t^{-i}) - \log\big(\boldsymbol{\pi}_\theta(a_t^i, a_t^{-i}|s_t)\big)\Big]$, and the corresponding optimal Bellman operator is

$$\big(\mathbf{H}^{\text{soft}}Q^i\big)\big(s, a^i, a^{-i}\big) \triangleq r^i\big(s, a^i, a^{-i}\big) + \gamma \cdot \mathbb{E}_{s' \sim P(\cdot|s,\boldsymbol{a})}\Big[\log \sum_{\boldsymbol{a}} Q^i\big(s', \boldsymbol{a}\big)\Big]. \tag{38}$$

This process is called *soft learning* because $\log \sum_{\boldsymbol{a}} \exp\big(Q(s, \boldsymbol{a})\big) \approx \max_{\boldsymbol{a}} Q\big(s, \boldsymbol{a}\big)$.

One substantial benefit of developing a probabilistic framework for multi-agent learning is that it can help model the *bounded rationality* (Simon, 1972). Instead of assuming perfect rationality and agents reaching NE, bounded rationality accounts for situations in which rationality is compromised; it can be constrained by either the difficulty of the decision problem or the agents' own cognitive limitations. One intuitive example is the psychological experiment of the Keynes beauty contest (Keynes, 1936), in which all players are asked to guess a number between 0 and 100 and the winner is the person whose number is closest to the $1/2$ of the average number of all guesses. Readers are recommended to pause here and think about which number you would guess. Although the NE of this game is 0, the majority of people guess a number between 13 and 25 (Coricelli & Nagel, 2009), which suggests that human beings tend to reason only by 1-2 levels of recursion in strategic games Camerer et al. (2004), i.e., "I believe how you believe how I believe".

Wen et al. (2018) developed the first MARL powered reasoning model that accounts for bounded rationality, which they called *probabilistic recursive reasoning* (PR2). The key idea of PR2 is that a dependency structure

---

[28]Soft learning is also called maximum-entropy RL (Haarnoja et al., 2018).

is assumed when splitting the joint policy $\boldsymbol{\pi}_\theta$, written by

$$\boldsymbol{\pi}_\theta \left( a^i, a^{-i} | s \right) = \pi_{\theta^i}^i \left( a^i | s \right) \rho_{\theta^{-i}}^{-i} \left( a^{-i} | s, a^i \right) \qquad (\textbf{PR2}, \quad \text{Level-1}), \tag{39}$$

that is, the opponent is considering how the learning agent is going to affect its actions, i.e., a Level-1 model. The unobserved opponent model is approximated by a best-fit model $\rho_{\theta^{-i}}$ when optimising Eq. (37). In the team game setting, since agents' objectives are fully aligned, the optimal $\rho_{\phi^{-i}}$ has a closed-form solution $\rho_{\phi^{-i}}^{-i}(a^{-i}|s, a^i) \propto \exp\left( Q^i(s, a^i, a^{-i}) - Q^i(s, a^i) \right)$. Following the direction of recursive reasoning, Tian et al. (2019) proposed an algorithm named ROMMEO that splits the joint policy by

$$\boldsymbol{\pi}_\theta \left( a^i, a^{-i} | s \right) = \pi_{\theta^i}^i \left( a^i | s, a^{-i} \right) \rho_{\theta^{-i}}^{-i} \left( a^{-i} | s \right) \qquad (\textbf{ROMMEO}, \quad \text{Level-1}), \tag{40}$$

in which a Level-1 model is built from the learning agent's perspective. Grau-Moya et al. (2018); Shi et al. (2019) introduced a Level-0 model where no explicit recursive reasoning is considered.

$$\boldsymbol{\pi}_\theta \left( a^i, a^{-i} | s \right) = \pi_{\theta^i}^i \left( a^i | s \right) \rho_{\theta^{-i}}^{-i} \left( a^{-i} | s \right) \qquad (\text{Level-0}). \tag{41}$$

However, they generalised the multi-agent soft learning framework to include the zero-sum setting. Wen et al. (2019) recently proposed a mixture of hierarchy Level-$k$ models in which agents can reason at different recursion levels, and higher-level agents make the best response to lower-level agents (see the blue part in Figure 10). They called this method *generalised recursive reasoning* (GR2).

$$\pi_k^i(a_k^i|s) \propto \int_{a_{k-1}^{-i}} \left\{ \pi_k^i(a_k^i|s, a_{k-1}^{-i}) \cdot \right.$$
$$\underbrace{\int_{a_{k-2}^i} \left[ \rho_{k-1}^{-i}(a_{k-1}^{-i}|s, a_{k-2}^i) \pi_{k-2}^i(a_{k-2}^i|s) \right] \mathrm{d}a_{k-2}^i}_{\text{opponents of level k-1 best responds to agent } i \text{ of level k-2}} \left. \right\} \mathrm{d}a_{k-1}^{-i}. \qquad (\textbf{GR2}, \quad \text{Level-}K). \tag{42}$$

In GR2, practical multi-agent soft actor-critic methods with convergence guarantee were introduced to make large-$K$ reasoning tractable.

## 6.2 Dec-POMDPs

Dec-POMDP is a stochastic team game with partial observability. However, optimally solving Dec-POMDPs is a challenging combinatorial problem that is $NEXP$-complete (Bernstein et al., 2002). As the horizon increases, the doubly exponential growth in the number of possible policies quickly makes solution methods intractable. Most of the solution algorithms for Dec-POMDPs, including the above VDN/QMIX/QTRAN/Q-DPP, are based on the learning paradigm of centralised training with decentralised execution (CTDE) (Oliehoek et al., 2016). CTDE methods assume a centralised controller that can access observations across all agents during training. A typical implementation is through a centralised critic with a decentralised actor (Lowe et al., 2017). In representing agents' local policies, stochastic finite-state controllers and a correlation device are commonly applied (Bernstein et al., 2009). Through this representation, Dec-POMDP can be formulated as non-linear programmes (Amato et al., 2010); this process allows the use of a wide range of off-the-shelf optimisation algorithms. Szer et al. (2005); Dibangoye et al. (2016); Dibangoye & Buffet (2018) introduced the transformation from Dec-POMDP into a continuous-state MDP, named the *occupancy-state MDP (oMDP)*. The occupancy state is essentially a distribution over hidden states and the joint histories of observation-action pairs. In contrast to the standard MDP, where the agent learns an optimal value function that maps histories (or states) to real values, the learner in oMDP learns an optimal value function that maps occupancy states and joint actions to real values (they call the corresponding policy a *plan*). These value functions in oMDP are piece-wise linear and convex. Importantly, the benefit of restricting attention on the occupancy state is that the resulting algorithms are guaranteed to converge to a near-optimal plan for any finite Dec-POMDP with a probability of one, while traditional RL methods, such as REINFORCE, may only converge towards a local optimum.

In addition to CTDE methods, famous approximation solutions to Dec-POMDP include the Monte Carlo policy iteration method (Wu et al., 2010), which enjoys linear-time complexity in terms of the number of

agents, planning by maximum-likelihood methods (Toussaint et al., 2008; Wu et al., 2013), which easily scales up to thousands of agents, and a method that decentralises POMDP by maintaining shared memory among agents (Nayyar et al., 2013).

## 6.3 Networked Multi-Agent MDPs

A rapidly growing area in the optimisation domain for addressing decentralised learning for cooperative tasks is the networked multi-agent MDP (M-MDP). In the context of M-MDP, agents are considered heterogeneous rather than homogeneous; they have different reward functions but still form a team to maximise the team-average reward $\mathsf{R} = \frac{1}{N} \sum_{i=1}^{N} R^i(s, \boldsymbol{a}, s')$. Furthermore, in M-MDP, the centralised controller is assumed to be non-existent; instead, agents can only exchange information with their neighbours in a time-varying communication network defined by $G_t = ([N], E_t)$, where $E_t$ represents the set of all communicative links between any two of the $N$ neighbouring agents at time step $t$. The states and joint actions are assumed to be globally observable, but each agent's reward is only locally observable to itself. Compared to stochastic team games, this setting is believed to be more realistic for real-world applications such as smart grids (Dall'Anese et al., 2013) or transport management (Adler & Blue, 2002).

The cooperative goal of the agents in M-MDP is to maximise the team average cumulative discounted reward obtained by all agents over the network, that is,

$$\max_{\boldsymbol{\pi}} \frac{1}{N} \sum_{i=1}^{N} \mathbb{E} \Big[ \sum_{t \geq 0} \gamma^t R_t^i(s_t, \boldsymbol{a}_t) \Big]. \tag{43}$$

Accordingly, under the joint policy $\boldsymbol{\pi} = \prod_{i \in \{1, \dots, N\}} \pi^i(a^i | s)$, the Q-function is defined as

$$Q^{\boldsymbol{\pi}}(s, \boldsymbol{a}) = \frac{1}{N} \sum_{i=1}^{N} \mathbb{E}_{\boldsymbol{a}_t \sim \boldsymbol{\pi}(\cdot | s_t), s_t \sim P(\cdot | s_t, \boldsymbol{a}_t)} \left[ \sum_{t \geq 0} \gamma^t R_t^i(s_t, \boldsymbol{a}_t) \Big| s_0 = s, a_0 = \boldsymbol{a} \right]. \tag{44}$$

To optimise Eq. (50), the optimal Bellman operator is written as

$$\left( \mathbf{H}^{\text{M-MDP}} Q \right)(s, \boldsymbol{a}) = \frac{1}{N} \sum_{i=1}^{N} R^i(s, \boldsymbol{a}) + \gamma \cdot \mathbb{E}_{s' \sim P(\cdot | s, \boldsymbol{a})} \left[ \max_{\boldsymbol{a}' \sim \mathbb{A}} Q(s', \boldsymbol{a}') \right]. \tag{45}$$

However, since agents can know only their own reward, they do not share the estimation of the Q function but rather maintain their own copy. Therefore, from each agent's perspective, the individual optimal Bellman operator is written as

$$\left( \mathbf{H}^{\text{M-MDP}, i} Q^i \right)(s, \boldsymbol{a}) = R^i(s, \boldsymbol{a}) + \gamma \cdot \mathbb{E}_{s' \sim P(\cdot | s, \boldsymbol{a})} \left[ \max_{\boldsymbol{a}' \sim \mathbb{A}} Q^i(s', \boldsymbol{a}') \right]. \tag{46}$$

To solve the optimal joint policy $\boldsymbol{\pi}^*$, the agents must reach **consensus** over the global optimal policy estimation, that is, if $Q^1 = \dots = Q^N = Q^*$, we know

$$\left( \mathbf{H}^{\text{M-MDP}} Q^* \right)(s, \boldsymbol{a}) = \frac{1}{N} \sum_{i=1}^{N} \left( \mathbf{H}^{\text{M-MDP}, i} Q^i \right). \tag{47}$$

To satisfy Eq. (47), Zhang et al. (2018b) proposed a method based on neural fitted-Q iteration (FQI) (Riedmiller, 2005) in the batch RL setting (Lange et al., 2012). Specifically, let $\mathcal{F}_\theta$ denote the parametric function class of neural networks that approximate Q-functions, let $\mathcal{D} = \{(s_k, \boldsymbol{a}_k^i, s_k')\}$ be the replay buffer that contains all the transition data available to all agents, and let $\{R_k^i\}$ be the local reward known only to each agent. The objective of FQI can be written as

$$\min_{f \in \mathcal{F}_\theta} \frac{1}{N} \sum_{i=1}^{N} \frac{1}{2K} \sum_{j=1}^{K} \left[ y_k^i - f(s_k, \boldsymbol{a}_k; \theta) \right]^2, \quad \text{with } y_k^i = R_k^i + \gamma \cdot \max_{\boldsymbol{a} \in \mathbb{A}} Q_k^i(s_k', \boldsymbol{a}). \tag{48}$$

In each iteration, $K$ samples are drawn from $\mathcal{D}$. Since $y_k^i$ is known only to each agent $i$, Eq. (48) becomes a typical consensus optimisation problem (i.e., consensus must be reached for $\theta$) (Nedic & Ozdaglar, 2009). Multiple effective distributed optimisers can be applied to solve this problem, including the *DIGing* algorithm (Nedic et al., 2017). Let $g^i(\theta^i) = \frac{1}{2K}\sum_{j=1}^K \left[ y_k^i - f(s_k, \boldsymbol{a}_k; \theta) \right]^2$, $\alpha$ be the learning rate, and $G([N], E_l)$ be the topology of the network in the $l$st iteration; the *DIGing* algorithm designs the gradient updates for each agent $i$ as

$$\theta_{l+1}^i = \sum_{j=1}^N E_l(i,j) \cdot \theta_l^j - \alpha \cdot \rho_l^i, \quad \rho_{l+1}^i = \sum_{j=1}^N E_l(i,j) \cdot \rho_l^j + \nabla g^i\left(\theta_{l+1}^i\right) - \nabla g^i\left(\theta_l^i\right). \tag{49}$$

Intuitively, Eq. (49) implies that if all agents aim to reach a consensus on $\theta$, they must incorporate a weighted combination of their neighbours' estimates into their own gradient updates. However, due to the usage of neural networks, the agents may not reach an exact consensus. Zhang et al. (2018b) also studied the finite-sample bound in a high-probability sense that quantifies the generalisation error of the proposed neural FQI algorithm.

The idea of reaching consensus can be directly applied to solving Eq. (43) via policy-gradient methods. Zhang et al. (2018c) proposed an actor-critic algorithm in which the global Q-function is approximated individually by each agent. On the basis of Eq. (15), the critic of $Q^{i,\boldsymbol{\pi}_\theta}(s, \boldsymbol{a})$ is modelled by another neural network parameterised by $\omega^i$, i.e., $Q^i(\cdot, \cdot; \omega^i)$, and the parameter $\omega^i$ is updated as

$$\omega_{t+1}^i = \sum_{j=1}^N E_t(i,j) \cdot \left( \omega_t^j + \alpha \cdot \delta_t^j \cdot \nabla_\omega Q_t^j(\omega_t^j) \right) \tag{50}$$

where $\delta_t^j = R_t^j + \gamma \cdot \max_{\boldsymbol{a} \in \mathbb{A}} Q_t^j(s_t', \boldsymbol{a}; \omega_t^j) - Q_t^j(s_t', \boldsymbol{a}; \omega_t^j)$ is the TD error. Similar to Eq. (49), the update in Eq. (50) is a weighted sum of all the neighbouring gradients. The same group of authors later extended this approach to cover the continuous-action space in which a deterministic policy gradient method of Eq. (16) is applied (Zhang et al., 2018a). Moreover, (Zhang et al., 2018c) and (Zhang et al., 2018a) applied a linear function approximation to achieve an almost sure convergence guarantee. Following this thread, Suttle et al. (2019) and Zhang & Zavlanos (2019) extended the actor-critic method to an off-policy setting, rendering more data-efficient MARL algorithms.

## 6.4 Stochastic Potential Games

The potential game (PG) first appeared in Monderer & Shapley (1996). The physical meaning of Eq. (21) is that if any agent changes its policy unilaterally, the changes in reward will be represented on the potential function shared by all agents. A PG is guaranteed to have a pure-strategy NE – a desirable property that does not generally hold in normal-form games. Many efforts have since been dedicated to finding the NE of (static) PGs (Lã et al., 2016), among which fictitious play (Berger, 2007) and generalised weakened fictitious play (Leslie & Collins, 2006) are probably the most common solutions.

Generally, stochastic PGs (SPGs)[29] can be regarded as the "single-agent component" of a multi-agent stochastic game (Candogan et al., 2011) since all agents' interests in SPGs are described by a single potential function. However, the analysis of SPGs is exceptionally sparse. Zazo et al. (2015) studied an SPG with deterministic transition dynamics in which agents consider only *open-loop policies*[30]. In fact, generalising a PG to the stochastic setting is further complicated because agents must now execute policies that depend on the state and consider the actions of other players. In this setting, González-Sánchez & Hernández-Lerma (2013) investigated a type of SPG in which they derive a sufficient condition for NE, but it requires each agent's reward function to be a concave function of the state and the transition function to be invertible. Macua et al. (2018) studied a general form of SPG where a *closed-loop* NE can be found. Although they demonstrated the equivalence between solving the closed-loop NE and solving a single-agent optimal control

---

[29]As with team games, stochastic PG is also called dynamic PG or Markov PG.

[30]Open loop means that agents' actions are a function of time only. By contrast, close-loop policies take into account the state. In deterministic systems, these policies can be optimal and coincide in value. For a stochastic system, an open-loop strategy is unlikely to be optimal since it cannot adapt to state transitions.

problem, the agents' policies must depend only on disjoint subsets of components of the state. Notably, both González-Sánchez & Hernández-Lerma (2013) and Macua et al. (2018) proposed centralised methods; optimisation over the joint action space surely results in a combinatorial complexity when solving the SPGs. In addition, they do not consider an RL setting in which the system is *a priori* unknown.

The work of Mguni (2020) is probably the most comprehensive treatment of SPGs in a model-free setting. Similar to Macua et al. (2018), the authors revealed that the NE of the PG in pure strategies could be found by solving a dual-form MDP, but they reached the conclusion without the disjoint state assumption: the transition dynamics and potential function must be known. Specifically, they provided an algorithm to estimate the potential function based on the reward samples. To avoid combinatorial explosion, they also proposed a distributed policy-gradient method based on generalised weakened fictitious play (Leslie & Collins, 2006) that has linear-time complexity.

Recently, Mazumdar & Ratliff (2018) studied the dynamics of gradient-based learning on potential games. They found that in a general superclass of potential games named *Morse-Smale games* (Hirsch, 2012), the limit sets of competitive gradient-based learning with stochastic updates are attractors almost surely, and those attractors are either local Nash equilibria or non-Nash locally asymptotically stable equilibria but not saddle points.

## 7 Learning in Zero-Sum Games

Zero-sum games represent a competitive relationship among players in a game. Solving three-player zero-sum games is believed to be $PPAD$-hard (Daskalakis & Papadimitriou, 2005). In the two-player case, the NE $(\pi^{1,*}, \pi^{2,*})$ is essentially a saddle point $\mathbb{E}_{\pi^1,\pi^{2,*}}[R] \leq \mathbb{E}_{\pi^{1,*},\pi^{2,*}}[R] \leq \mathbb{E}_{\pi^{1,*},\pi^2}[R], \forall \pi^1, \pi^2$, and can be formulated as an LP problem in Eq. (51).

$$
\begin{aligned}
\min \ & U_1^* \\
\text{s.t.} \ & \sum_{a^2 \in \mathbb{A}^2} R^1(a^1, a^2) \cdot \pi^2(a^2) \leq U_1^*, \quad \forall a^1 \in \mathbb{A}^1 \\
& \sum_{a^2 \in \mathbb{A}^2} \pi^2(a^2) = 1 \\
& \pi^2(a^2) \geq 0, \quad \forall a^2 \in \mathbb{A}^2
\end{aligned}
\tag{51}
$$

Eq. (51) is considered from the min-player's perspective. One can also derive a dual-form LP from the max-player's perspective. In discrete games, the minimax theorem (Von Neumann & Morgenstern, 1945) is a simple consequence of the strong duality theorem of LP[31] (Matousek & Gärtner, 2007),

$$
\min_{\pi^1} \max_{\pi^2} \mathbb{E}\left[R(\pi^1, \pi^2)\right] = \max_{\pi^2} \min_{\pi^1} \mathbb{E}\left[R(\pi^1, \pi^2)\right]
\tag{52}
$$

which suggests the fact that whether the min player acts first or the max player acts first does not matter. However, the minimax theorem does not hold in general for multi-player zero-sum continuous games in which the reward function is nonconvex-nonconcave. In fact, a barrier to tractability exists for multi-player zero-sum games and two-player zero-sum games with continuous states and actions.

### 7.1 Discrete State-Action Games

Similar to single-agent MDP, value-based methods aim to find an optimal value function, which in the context of zero-sum SGs, corresponds to the minimax NE of the game. In two-player zero-sum SGs with discrete states and actions, we know $V^{1,\pi^1,\pi^2} = -V^{2,\pi^1,\pi^2}$, and by the minimax theorem (Von Neumann & Morgenstern, 1945), the optimal value function is $V^* = \max_{\pi^2} \min_{\pi^1} V^{1,\pi^1,\pi^2} = \min_{\pi^1} \max_{\pi^2} V^{1,\pi^1,\pi^2}$. In each stage game defined by $Q^1 = -Q^2$, the optimal value can be solved by a matrix zero-sum game through a linear program in Eq. (51). Shapley (1953) introduced the first value-iteration method, written as

$$
(\mathbf{H}^{\text{Shapley}} V)(s) = \min_{\pi^1 \in \Delta(\mathbb{A}^1)} \max_{\pi^2 \in \Delta(\mathbb{A}^2)} \mathbb{E}_{a^1 \sim \pi^1, a^2 \sim \pi^2, s' \sim P}\left[R^1(s, a^1, a^2) + \gamma \cdot V(s')\right],
\tag{53}
$$

---

[31]Solving zero-sum games is equivalent to solving a LP; Dantzig (1951) also proved the correctness of the other direction, that is, any LP can be reduced to a zero-sum game, though some degenerate solutions need careful treatments (Adler, 2013).

and proved $\mathbf{H}^{\text{Shapley}}$ is a contraction mapping (in the sense of the infinity norm) in solving two-player zero-sum SGs. In other words, assuming the transitional dynamics and reward function are known, the value-iteration method will generate a sequence of value functions $\{V_t\}_{t \geq 0}$ that asymptotically converges to the fixed point $V^*$, and the corresponding policies will converge to the NE policies $\boldsymbol{\pi}^* = (\pi^{1,*}, \pi^{2,*})$.

In contrast to Shapley's model-based value-iteration method, Littman (1994) proposed a model-free Q-learning method – Minimax-Q – that extends the classic Q-learning algorithm defined in Eq. (13) to solve zero-sum SGs. Specifically, in Minimax-Q, Eq. (14) can be equivalently written as

$$\mathbf{eval}^1\Big(\big\{Q^1(s_{t+1}, \cdot)\big\}\Big) = -\mathbf{eval}^2\Big(\big\{Q^2(s_{t+1}, \cdot)\big\}\Big)$$
$$= \min_{\pi^1 \in \Delta(\mathbb{A}^1)} \max_{\pi^2 \in \Delta(\mathbb{A}^2)} \mathbb{E}_{a^1 \sim \pi^1, a^2 \sim \pi^2}\Big[Q^1(s_{t+1}, a^1, a^2)\Big]. \tag{54}$$

The Q-learning update rule of Minimax-Q is exactly the same as that in Eq. (13). Minimax-Q can be considered an approximation algorithm for computing the fixed point $Q^*$ of the Bellman operator of Eq. (20) through stochastic sampling. Importantly, it assumes no knowledge about the environment. Szepesvári & Littman (1999) showed that under similar assumptions to those for Q-learning (Watkins & Dayan, 1992), the Bellman operator of Minimax-Q is a contraction mapping operator, and the stochastic updates made by Minimax-Q eventually lead to a unique fixed point that corresponds to the NE value. In addition to the tabular-form Q-function in Minimax-Q, various Q-function approximators have been developed. For example, Lagoudakis & Parr (2003) studied the factorised linear architectures for Q-function representation. Yang et al. (2019b) adopted deep neural networks and derived a rigorous finite-sample error bound. Zhang et al. (2018b) also derived a finite-sample bound for linear function approximators in the competitive M-MDPs.

## 7.2 Continuous State-Action Games

Recently, the challenge of training generative adversarial networks (GANs) (Goodfellow et al., 2014a) has ignited tremendous research interest in understanding policy gradient methods in two-player continuous games, specifically, games with a continuous station-action space and nonconvex-nonconcave loss landscape. In GANs, two neural network parameterised models – the generator G and the discriminator D – play a zero-sum game. In this game, the generator attempts to generate data that "look" authentic such that the discriminator cannot tell the difference from the true data; on the other hand, the discriminator tries not to be deceived by the generator. The loss function in this scenario is written as

$$\min_{\theta_{\text{G}} \in \mathbb{R}^d} \max_{\theta_{\text{D}} \in \mathbb{R}^d} f\big(\theta_{\text{G}}, \theta_{\text{D}}\big) = \tag{55}$$
$$\Big[\mathbb{E}_{x \sim p_{\text{data}}}\Big[\log \mathrm{D}_{\theta_{\text{D}}}\big(x\big)\Big] + \mathbb{E}_{z \sim p(z)}\Big[\log\Big(1 - \mathrm{D}_{\theta_{\text{D}}}\big(\mathrm{G}_{\theta_{\text{G}}}(z)\big)\Big)\Big]\Big]$$

where $\theta_{\text{G}}$ and $\theta_{\text{D}}$ represent neural networks parameters and $z$ is a random signal, serving as the input to the generator. In searching for the NE, one naive approach is to update both $\theta_{\text{G}}$ and $\theta_{\text{D}}$ by simultaneously implementing the gradient-descent-ascent (GDA) updates with the same step size in Eq. (55). This approach is equivalent to a MARL algorithm in which both agents are applying policy-gradient methods. With trivial adjustments to the step size (Bowling & Veloso, 2002; Bowling, 2005; Zhang & Lesser, 2010), GDA methods can work effectively in two-player two-action (thus convex-concave) games. However, in the nonconvex-nonconcave case, where the minimax theorem no longer holds, GDA methods are notoriously flawed from three aspects. First, GDA algorithms may not converge at all (Balduzzi et al., 2018a; Mertikopoulos et al., 2018; Daskalakis & Panageas, 2018), resulting in limited cycles[32] in which even the time average[33] does not coincide with NE (Mazumdar et al., 2019a). Second, there exist undesired stable stationary points for the GDA algorithms that are not local optima of the game (Adolphs et al., 2019; Mazumdar et al., 2019a).

---

[32] Limited cycle is a terminology in the study of dynamical systems, which describes oscillatory systems. In game theory, an example of limit cycles in the strategy space can be found in Rock-Paper-Scissor game.

[33] In two-player two-action games, Singh et al. (2000) showed that the time average payoffs would converge to a NE value if their policies do not.

Third, there exist games whose equilibria are not the attractors of GDA methods at all (Mazumdar et al., 2019a). These problems are partly caused by the intransitive dynamics (e.g., a typical intransitive game is rock-paper-scissors game) that are inherent in zero-sum games (Omidshafiei et al., 2020; Balduzzi et al., 2018a) and the fact that each agent may have a non-smooth objective function. In fact, even in simple linear-quadratic games, the reward function cannot satisfy the smoothness condition[34] globally, and the games are surprisingly not convex either (Fazel et al., 2018; Mazumdar et al., 2019a; Zhang et al., 2019c).

Three mainstream approaches have been followed to develop algorithms that have at least a local convergence guarantee. One natural idea is to make the inner loop solvable at a reasonably high level and then focus on a simpler type of game. In other words, the algorithm tries to find a stationary point of the function $\Phi(\cdot) := \max_{\theta_D \in \mathbb{R}^d} f(\cdot, \theta_D)$, instead of Eq. (55). For example, by considering games with a nonconvex and (strongly) concave loss landscape, Lin et al. (2019); Nouiehed et al. (2019); Rafique et al. (2018); Thekumparampil et al. (2019); Lu et al. (2020a); Kong & Monteiro (2019) presented an affirmative answer that GDA methods can converge to a stationary point in the outer loop of optimising $\Phi(\cdot) := \max_{\theta_D \in \mathbb{R}^d} f(\cdot, \theta_D)$. Based on this understanding, they developed various GDA variants that apply the "best response" in the inner loop while maintaining an inexact gradient descent in the outer loop. We refer to Lin et al. (2019) [Table 1] for a detailed summary of the time complexity of the above methods.

The second mainstream idea is to shift the equilibrium of interest from the NE, which is induced by simultaneous gradient updates, to the Stackelberg equilibrium, which is a solution concept in leader-follower (i.e., alternating update) games. Jin et al. (2019) introduced the concept of the local Stackelberg equilibrium, named *local minimax*, based on which he established the connection to GDA methods by showing that all stable limit points of GDA are exactly local minimax points. Fiez et al. (2019) also built connections between the NE and Stackelberg equilibrium by formulating the conditions under which attracting points of GDA dynamics are Stackelberg equilibria in zero-sum games. When the loss function is bilinear, theoretical evidence was found that alternating updates converge faster than simultaneous GDA methods (Zhang & Yu, 2019).

The third mainstream idea is to analyse the loss landscape from a game-theoretic perspective and design corresponding algorithms that mitigate oscillatory behaviour. Compared to the previous two mainstream ideas, which helped generate more theoretical insights than applicable algorithms, works within this stream demonstrate strong empirical improvements in training GANs. Mescheder et al. (2017) investigated the game Hessian and identified that issues on the eigenvalues trigger the limited cycles. As a result, they proposed a new type of update rule based on consensus optimisation, together with a convergence guarantee to a local NE in smooth two-player zero-sum games. Adolphs et al. (2019) leveraged the curvature information of the loss landscape to propose algorithms in which all stable limit points are guaranteed to be local NEs. Similarly, Mazumdar et al. (2019b) took advantage of the differential structure of the game and constructed an algorithm for which the local NEs are the only attracting fixed points. In addition, Daskalakis et al. (2017); Mertikopoulos et al. (2018) addressed the issue of limit cycling behaviour in training GANs by proposing the technique of *optimistic mirror descent (OMD)*. OMD achieves the last-iterate convergent guarantee in bilinear convex-concave games. Specifically, at each time step, OMD adjusts the gradient of that time step by considering the opponent policy at the next time step. Let $M_{t+1}$ be the predictor of the next iteration gradient[35]; we can write OMD as follows.

$$\theta_{G,t+1} = \theta_{G,t} + \alpha \cdot \left( \nabla_{\theta_G,t} f(\theta_G, \theta_D) + M_{\theta_G,t+1} - M_{\theta_G,t} \right)$$
$$\theta_{D,t+1} = \theta_{G,t} - \alpha \cdot \left( \nabla_{\theta_D,t} f(\theta_G, \theta_D) + M_{\theta_D,t+1} - M_{\theta_D,t} \right) \tag{56}$$

In fact, the pivotal idea of opponent prediction in OMD, developed in the optimisation domain, resembles the idea of approximate policy prediction in the MARL domain (Zhang & Lesser, 2010; Foerster et al., 2018a).

Thus far, the most promising results are probably those of Bu et al. (2019) and Zhang et al. (2019c), which reported the first results in solving zero-sum LQ games with a global convergence guarantee. Specifically, Zhang et al. (2019c) developed the solution through projected nested-gradient methods, while Bu et al. (2019) solved the problem through a projection-free Stackelberg leadership model. Both of the models achieve a sublinear rate for convergence.

---

[34]A differentiable function is said to be smooth if the gradients of the function are continuous.
[35]In practice, it is usually set as the last iteration gradient.

### 7.3 Extensive-Form Games

As briefly introduced in Section 3.4, zero-sum EFG with imperfect information can be efficiently solved via LP in sequence form representations (Koller & Megiddo, 1992; 1996). However, these approaches are limited to solving only small-scale problems (e.g., games with $\mathcal{O}(10^7)$ information states). In fact, considerable additional effort is needed to address real-world games (e.g., limit Texas hold'em, which has $\mathcal{O}(10^{18})$ game states); to name a few, Monte Carlo Tree Search (MCTS) techniques[36] (Cowling et al., 2012; Browne et al., 2012; Silver et al., 2016), isomorphic abstraction techniques (Billings et al., 2003; Gilpin & Sandholm, 2006), and iterative (policy) gradient-based approaches (Gordon, 2007; Gilpin et al., 2007; Zinkevich, 2003).

A central idea of iterative policy gradient-based methods is minimising regret[37]. A learning rule achieves no-regret, also called *Hannan consistency* in game theoretical terms (Hannan, 1957), if, intuitively speaking, against any set of opponents it yields a payoff that is no less than the payoff the learning agent could have obtained by playing any one of its pure strategies in hindsight. Recall the reward function under a given policy $\boldsymbol{\pi} = (\pi^i, \pi^{-i})$ in Eq. (27); the (average) regret of player $i$ is defined by:

$$\text{Reg}_T^i = \frac{1}{T} \max_{\pi^i} \sum_{t=1}^{T} \left[ R^i(\pi^i, \pi_t^{-i}) - R^i(\pi_t^i, \pi_t^{-i}) \right]. \tag{57}$$

A no-regret algorithm satisfies $\text{Reg}_T^i \to 0$ as $T \to \infty$ with probability 1. When Eq. (57) equals zero, all agents are acting with their best response to others, which essentially forms a NE. Therefore, one can regard regret as a type of "distance" to NE. As one would expect, the single-agent Q-learning procedure can be shown to be Hannan consistent in a stochastic game against opponents playing stationary policies (Shoham & Leyton-Brown, 2008) [Chapter 7] since the optimal Q-function guarantees the best response. In contrast, the Minimax-Q algorithm in Eq. (54) is not Hannan consistent because if the opponent plays a sub-optimal strategy, Minimax-Q is unable to exploit the opponent due to the over-conservativeness in terms of over-estimating its opponents.

An important result about regret states is that in a zero-sum game at time $T$, if both players' average regret is less than $\epsilon$, then their average strategy constitutes a $2\epsilon$-NE of the game (Zinkevich et al., 2008, Theorem 2). In general-sum games, the average strategy of the $\epsilon$-regret algorithm will reach an $\epsilon$-*coarse correlated equilibrium* of the game (Michael, 2020, Theorem 6.3.1). This result essentially implies that regret-minimising algorithms (or, algorithms with Hannan consistency) applied in a self-play manner can be used as a general technique to approximate the NE of zero-sum games. Building upon this finding, two families of methods are developed, namely, fictitious play types of methods (Berger, 2007) and counterfactual regret minimisation (Zinkevich et al., 2008), which lay the theoretical foundations for modern techniques to solve real-world games.

#### 7.3.1 Variations of Fictitious Play

Fictitious play (FP) (Berger, 2007) is one of the oldest learning procedures in game theory that is provably convergent for zero-sum games, potential games, and two-player n-action games with generic payoffs. In FP, each player maintains a belief about the empirical mean of the opponents' average policy, based on which the player selects the best response. With the best response defined in Eq. (17), we can write the FP updates as

$$a_t^{i,*} \in \mathbf{Br}^i \left( \pi_t^{-i} = \frac{1}{t} \sum_{\tau=0}^{t-1} \mathbb{1} \left\{ a_\tau^{-i} = a, a \in \mathbb{A} \right\} \right), \pi_{t+1}^i = \left( 1 - \frac{1}{t} \right) \pi_t^i + \frac{1}{t} a_t^{i,*}, \forall i. \tag{58}$$

In the FP scheme, each agent is oblivious to the other agents' reward; however, they need full access to their own payoff matrix in the stage game. In the continuous case with an infinitesimal learning rate of $1/t \to 0$,

---

[36]Notably, though MCTS methods such as UCT (Kocsis & Szepesvári, 2006) work remarkably well in turn-based EFGs, such as GO and chess, they cannot converge to a NE trivially in (even perfect-information) simultaneous-move games (Schaeffer et al., 2009). See a rigorous treatment for remedy in Lisy et al. (2013).

[37]One can regard minimising regret as one solution concept for multi-agent learning problems, similar to the reward maximisation in single-agent learning.

Eq. (58) is equivalent to $d\boldsymbol{\pi}_t/dt \in \mathbf{Br}(\boldsymbol{\pi}_t) - \boldsymbol{\pi}_t$ in which $\mathbf{Br}(\boldsymbol{\pi}_t) = \left(\mathbf{Br}(\pi_t^{-1}), ..., \mathbf{Br}(\pi_t^{-N})\right)$. Viossat & Zapechelnyuk (2013) proved that continuous FP leads to no regret and is thus Hannan consistent. If the empirical distribution of each $\pi_t^i$ converges in FP, then it converges to a NE[38].

Although standard discrete-time FP is not Hannan consistent (Cesa-Bianchi & Lugosi, 2006, Exercise 3.8), various extensions have been proposed that guarantee such a property; see a full list summarised in Hart (2013) [Section 10.9]. Smooth FP (Fudenberg & Kreps, 1993; Fudenberg & Levine, 1995) is a stochastic variant of FP (thus also called stochastic FP) that considers a smooth $\epsilon$-best response in which the probability of each action is a softmax function of that action's utility/reward against the historical frequency of the opponents' play. In smooth FP, each player's strategy is a genuine mixed strategy. Let $R^i(a_1^i, \pi_t^{-i})$ be the expected reward of player $i$'s action $a_1^i \in \mathbb{A}^i$ under opponents' strategy $\pi^{-i}$; the probability of playing $a_1^i$ in the best response is written as

$$\mathbf{Br}_\lambda^i(\pi_t^{-i}) := \frac{\exp\left(\frac{1}{\lambda} R^i\left(a_1^i, \pi_t^{-i}\right)\right)}{\sum_{k=1}^{|\mathbb{A}^i|} \exp\left(\frac{1}{\lambda} R^i\left(a_k^i, \pi_t^{-i}\right)\right)}. \tag{59}$$

Benaïm & Faure (2013) verified the Hannan consistency of the smooth best response with the smoothing parameter $\lambda$ being time dependent and vanishing asymptotically. In potential games, smooth FP is known to converge to a neighbourhood of the set of NE (Hofbauer & Sandholm, 2002). Recently, Swenson & Poor (2019) showed a generic result that in almost all $N \times 2$ potential games, smooth FP converges to the neighbourhood of a pure-strategy NE with a probability of one.

In fact, "smoothing" the cumulative payoffs before computing the best response is crucial to designing learning procedures that achieve Hannan consistency (Kaniovski & Young, 1995). One way to achieve such smoothness is through stochastic smoothing or adding perturbations[39]. For example, the smooth best response in Eq. (59) is a closed-form solution if one perturbs the cumulative reward by an additional entropy function, that is,

$$\pi^{i,*} \in \mathbf{Br}(\pi^{-i}) = \left\{ \arg\max_{\hat{\pi} \in \Delta(\mathbb{A}^i)} \mathbb{E}_{\hat{\pi}^i, \pi^{-i}}\left[R^i + \lambda \cdot \log(\hat{\pi})\right] \right\}. \tag{60}$$

Apart from smooth FP, another way to add perturbation is the *sampled FP* in which during each round, the player samples historical time points using a randomised sampling scheme, and plays the best response to the other players' moves, restricted to the set of sampled time points. Sampled FP is shown to be Hannan consistent when used with Bernoulli sampling (Li & Tewari, 2018).

Among the many extensions of FP, the most important is probably *generalised weakened FP (GWFP)* (Leslie & Collins, 2006), which releases the standard FP by allowing both approximate best response and perturbed average strategy updates. Specifically, if we write the $\epsilon$-best response of player $i$ as

$$R^i\left(\mathbf{Br}_\epsilon(\pi^{-i}), \pi^{-i}\right) \geq \sup_{\pi \in \Delta(\mathbb{A}^i)} R^i\left(\pi, \pi^{-i}\right) - \epsilon. \tag{61}$$

then the GWFP updating steps change from Eq. (58) to

$$\pi_{t+1}^i = \left(1 - \alpha^{t+1}\right)\pi_t^i + \alpha_{t+1}\left(\mathbf{Br}_\epsilon^i(\pi^{-i}) + M_{t+1}^i\right), \quad \forall i. \tag{62}$$

GWFP is Hannan consistent if $\alpha_t \to 0, \epsilon_t \to 0, \sum_{\alpha_t} = \infty$ when $t \to \infty$, and $\{M_t\}$ meets $\lim_{t \to \infty} \sup_k \left\{ \left\| \sum_{i=t}^{k-1} \alpha^{i+1} M^{i+1} \right\| \text{ s.t. } \sum_{i=t}^{k-1} \alpha^{i+1} \leq T \right\} = 0$. It is trivial to see that GWFP recovers FP when $\alpha_t = 1/t, \epsilon_t = 0, M_t = 0$. GWFP is an important extension of FP in that it provides two key components for bridging game theoretic ideas with RL techniques. With the approximate best response (highlighted in blue, also named as the "weakened" term), this approach allows one to adopt a model-free RL algorithm, such as deep Q-learning, to compute the best response. Moreover, the perturbation term (highlighted in

---

[38]Note that the convergence in Nash strategy does not necessarily mean the agents will receive the expected payoff value at NE. In the example of Rock-Paper-Scissor games, agents' actions are still miscorrelated after convergence, flipping between one of the three strategies, though their average policies do converge to $(1/3, 1/3, 1/3)$.

[39]The physical meaning of perturbing the cumulative payoff is to consider the incomplete information about what the opponent has been playing, variability in their payoffs, and unexplained trembles.

red, also named as the "generalised" term) enables one to incorporate policy exploration; if one applies an entropy term as the perturbation in addition to the best response (in which the smooth FP in Eq. (60) is also recovered), the scheme of maximum-entropy RL methods (Haarnoja et al., 2018) is recovered. In fact, the generalised term also accounts for the perturbation that comes from the fact the beliefs are not updated towards the exact mixed strategy $\pi^{-i}$ but instead towards the observed actions (Benaïm & Hirsch, 1999). As a direct application, Perolat et al. (2018) implemented the GWFP process through an actor-critic framework (Konda & Tsitsiklis, 2000) in the MARL setting.

Brown's original version of FP (Berger, 2007) describes alternating updates by players; yet, the modern usage of FP involves players updating their beliefs simultaneously (Berger, 2007). In fact, Heinrich et al. (2015) only recently proposed the first FP algorithm for EFG using the sequence-form representation. The extensive-form FP is essentially an adaptation of GWFP from NFG to EFG based on the insight that a mixture of normal-form strategies can be implemented by a weighted combination of behavioural strategies that have the same realisation plan (recall Section 3.3.2). Specifically, let $\pi$ and $\beta$ be two behavioural strategies, $\Pi$ and $B$ be the two realisation-equivalent mixed strategies[40], and $\alpha \in \mathbb{R}^+$; then, for each information state $S$, we have

$$\tilde{\pi}(S) = \pi(S) + \frac{\alpha\mu^\beta(\sigma_S)}{(1-\alpha)\mu^\pi(\sigma_S) + \alpha\mu^\beta(\sigma_S)}\Big(\beta(S) - \pi(S)\Big), \quad \forall S \in \mathbb{S}, \tag{63}$$

where $\sigma_S$ is the sequence leading to $S$, $\mu^{\pi/\beta}(\sigma_S)$ is the realisation probability of $\sigma_S$ under a given policy, and $\tilde{\pi}(S)$ defines a new behaviour that is realisation equivalent to the mixed strategy $(1-\alpha)\Pi + \alpha B$. The extensive-form FP essentially iterates between Eq. (61), which computes the $\epsilon$-best response, and Eq. (63), which updates the old behavioural strategy with a step size of $\alpha$. Note that these two steps must iterate over all information states of the game in each iteration. Similar to the normal-form FP in Eq. (58), extensive-form FP generates a sequence of $\{\pi_t\}_{t\geq 1}$ that provably converges to the NE of a zero-sum game under self-play if the step size $\alpha$ goes to zero asymptotically. As a further enhancement, Heinrich & Silver (2016) implemented neural fictitious self-play (NFSP), in which the best response step is computed by deep Q-learning (Mnih et al., 2015) and the policy mixture step is computed through supervised learning. NFSP requires the storage of large replay buffers of past experiences; Lockhart et al. (2019) removes this requirement by obtaining the policy mixture for each player through an independent policy-gradient step against the respective best-responding opponent. All these amendments help make extensive-form FP applicable to real-world games with large-scale information states.

### 7.3.2 Counterfactual Regret Minimisation

Another family of methods achieve Hannan consistency by directly minimising the regret, in particular, a special kind of regret named counterfactual regret (CFR) (Zinkevich et al., 2008). Unlike FP methods, which are developed from the stochastic approximation perspective and generally have asymptotic convergence guarantees, CFR methods are established on the framework of online learning and online convex optimisation (Shalev-Shwartz et al., 2011), which makes analysing the speed of convergence, i.e., the regret bound, to the NE possible.

The key insight from CFR methods is that in order to minimise the total regret in Eq. (57) to approximate the NE, it suffices to minimise the *immediate counterfactual regret* at the level of each information state. Mathematically, Zinkevich et al. (2008) [Theorem 3] shows that the sum of the immediate counterfactual regret over all encountered information states provides an upper bound for the total regret in Eq. (57), i.e.,

$$\mathsf{Reg}_T^i \leq \sum_{S \in \mathbb{S}^i} \max\Big\{\mathsf{Reg}_{T,imm}^i(S), 0\Big\}, \quad \forall i. \tag{64}$$

To fully describe $\mathsf{Reg}_{T,imm}^i(S)$, we need two additional notations. Let $\mu^{\boldsymbol{\pi}}(\boldsymbol{\sigma}_S \to \boldsymbol{\sigma}_T)$ denote, given agents' behavioural policies $\boldsymbol{\pi}$, the realisation probability of going from the sequence $\boldsymbol{\sigma}_S$[41], which leads to the

---

[40]Recall that in games with perfect recall, Kuhn's theorem (Kuhn, 1950a) suggests that the behavioural strategy and mixed strategies are equivalent in terms of the realisation probability of different outcomes.

[41]Recall that for games of perfect recall, the sequence that leads to the information state, including all the choice nodes within that information state, is unique.

information state $S \in \mathbb{S}^i$ to its extended sequence $\boldsymbol{\sigma}_T$, which continues from $S$ and reaches the terminal state $T$. Let $\hat{v}^i(\boldsymbol{\pi}, S)$ be the *counterfactual value function*, i.e., the expected reward of agent $i$ in non-terminal information state $S$, which is written as

$$\hat{v}^i(\boldsymbol{\pi}, S) = \sum_{s \in S, T \in \mathbb{T}} \mu^{\pi^{-i}}(\boldsymbol{\sigma}_s) \mu^{\boldsymbol{\pi}}(\boldsymbol{\sigma}_s \to \boldsymbol{\sigma}_T) R^i(T). \tag{65}$$

Note that in Eq. (65), the contribution from player $i$ in realising $\boldsymbol{\sigma}_s$ is excluded; we treat whatever action current player $i$ needs to reach state $s$ as having a probability of one, that is, $\mu^{\pi^i}(\boldsymbol{\sigma}_s) = 1$. The motivation is that now one can make the value function $\hat{v}^i(\boldsymbol{\pi}, S)$ "counterfactual" simply by writing the consequence of player $i$ not playing action $a$ in the information state $S$ as $\left(\hat{v}^i(\boldsymbol{\pi}|_{S \to a}, S) - \hat{v}^i(\boldsymbol{\pi}, S)\right)$, in which $\boldsymbol{\pi}|_{S \to a}$ is a joint strategy profile identical to $\boldsymbol{\pi}$, except player $i$ always chooses action $a$ when information state $S$ is encountered. Finally, based on Eq. (65), the immediate counterfactual regret can be expressed as

$$\text{Reg}^i_{T,imm}(S) = \max_{a \in \chi(S)} \text{Reg}^i_T(S, a), \quad \text{Reg}^i_T(S, a) = \frac{1}{T} \sum_{t=1}^{T} \left(\hat{v}^i(\boldsymbol{\pi}_t|_{S \to a}, S) - \hat{v}^i(\boldsymbol{\pi}_t, S)\right). \tag{66}$$

Note that the $T$ in Eq. (65) is different from that in Eq. (66).

Since minimising the immediate counterfactual regret minimises the overall regret, we can find an approximate NE by choosing a specific behavioural policy $\pi^i(S)$ that minimises Eq. (66). To this end, one can apply Blackwell's approachability theorem (Blackwell et al., 1956) to minimise the regret independently on each information set, also known as *regret matching* (Hart & Mas-Colell, 2001). As we are most concerned with positive regret, denoted by $\lfloor \cdot \rfloor_+$, we have $\forall S \in \mathbb{S}^i, \forall a \in \chi(S)$, the strategy of player $i$ at time $T + 1$ as Eq. (67).

$$\pi^i_{T+1}(S, a) = \begin{cases} \dfrac{\lfloor \text{Reg}^i_T(S, a) \rfloor_+}{\sum_{a \in \chi(S)} \lfloor \text{Reg}^i_T(S, a) \rfloor_+} & \text{if } \sum_{a \in \chi(S)} \lfloor \text{Reg}^i_T(S, a) \rfloor_+ > 0 \\[3mm] \dfrac{1}{|\chi(S)|} & \text{otherwise} \end{cases}. \tag{67}$$

In the standard CFR algorithm, for each information set, Eq. (67) is used to compute action probabilities in proportion to the positive cumulative regrets. In addition to regret matching, another online learning tool that minimises regret is *Hedge* (Freund & Schapire, 1997; Littlestone & Warmuth, 1994), in which an exponentially weighted function is used to derive a new strategy, which is

$$\pi_{t+1}(a_k) = \frac{\pi_t(a_k) e^{-\eta R_t(a_k)}}{\sum_{j=1}^{K} \pi_t(a_j) e^{-\eta R_t(a_j)}}, \quad \pi_1(\cdot) = \frac{1}{K}. \tag{68}$$

In computing Eq. (68), Hedge needs access to the full information of the reward values for all actions, including those that are not selected. *EXP3* (Auer et al., 1995) extended the Hedge algorithm for a *partial information game* in which the player knows only the reward of the the chosen action (i.e., a bandit version) and has to estimate the loss of the actions that it does not select. Brown et al. (2017) augmented the Hedge algorithm with a tree-pruning technique based on dynamic thresholding. Gordon (2007) developed *Lagrangian hedging*, which unifies no-regret algorithms, including both regret matching and Hedge, through a class of potential functions. We recommend Cesa-Bianchi & Lugosi (2006) for a comprehensive overview of no-regret algorithms.

No-regret algorithms, under the framework of online learning, offer a natural way to study the regret bound (i.e., how fast the regret decays with time). For example, CFR and its variants ensure a counterfactual regret bound of $\mathcal{O}(\sqrt{T})$[42], as a result of Eq. (64), the convergence rate for the total regret is upper bounded by $\mathcal{O}(\sqrt{T} \cdot |\mathbb{S}|)$, which is linear in the number of information states. In other words, the average policy of

---

[42]According to Zinkevich (2003), any online convex optimisation problem can be made to incur $\text{Reg}_T = \Omega(\sqrt{T})$.

applying CFR-type methods in a two-player zero-sum EFG generates an $\mathcal{O}(|\mathbb{S}|/\sqrt{T})$-approximate NE after $T$ steps through self-play[43].

Compared with the LP approach (recall Eq. (33)), which is applicable only for small-scale EFGs, the standard CFR method can be applied to limit Texas hold'em with as many as $10^{12}$ states. CFR$^{+}$, the fastest implementation of CFR, can solve games with up to $10^{14}$ states (Tammelin et al., 2015). However, CFR methods still have a bottleneck in that computing Eq. (65) requires a traversal of the entire game tree to the terminal nodes in each iteration. Pruning the sub-optimal paths in the game tree is a natural solution (Brown & Sandholm, 2015; Brown et al., 2017; Brown & Sandholm, 2017). Many CFR variants have been developed to improve computational efficiency further. Lanctot et al. (2009) integrated Monte Carlo sampling with CFR (MCCFR) to significantly reduce the per iteration time cost of CFR by traversing a smaller sampled portion of the tree. Burch et al. (2012) improved MCCFR by sampling only a subset of a player's actions, which provides even faster convergence rate in games that contain many player actions. Gibson et al. (2012); Schmid et al. (2019) investigated the sampling variance and proposed MCCFR variants with a variance reduction module. Johanson et al. (2012b) introduced a more accurate MCCFR sampler by considering the set of outcomes from the chance node, rather than sampling only one outcome, as in all previous methods. Apart from Monte Carlo methods, function approximation methods have also been introduced (Waugh et al., 2014; Jin et al., 2018). The idea of these methods is to predict regret directly, and the no-regret algorithm then uses these predictions in place of the true regret to define a sequence of policies. To this end, the application of deep neural networks has led to great success (Brown et al., 2019).

Interestingly, there exists a hidden equivalence between model-free policy-based/actor-critic MARL methods and the CFR algorithm (Jin et al., 2018; Srinivasan et al., 2018). In particular, if we consider the counterfactual value function in Eq. (65) to be explicitly dependent on the action $a$ that player $i$ chooses at state $S$, in which we have $\hat{v}^i(\boldsymbol{\pi}, S) = \sum_{a \in \chi(S)} \pi^i(S, a)\hat{q}^i(\boldsymbol{\pi}, S, a)$, then it is shown in Srinivasan et al. (2018) [Section 3.2] that the Q-function in standard MARL $Q^{i,\boldsymbol{\pi}}(s, \mathbf{a}) = \mathbb{E}_{s' \sim P, \mathbf{a} \sim \boldsymbol{\pi}}\left[\sum_t \gamma^t R^i(s, \mathbf{a}, s')|s, \mathbf{a}\right]$ differs from $\hat{q}^i(\boldsymbol{\pi}, S, a)$ in CFR only by a constant of the probability of reaching $S$, that is,

$$Q^{i,\boldsymbol{\pi}}(s, \mathbf{a}) = \frac{\hat{q}^i(\boldsymbol{\pi}, S, a)}{\sum_{s \in S} \mu^{\pi^{-i}}(\boldsymbol{\sigma}_s)}. \tag{69}$$

Subtracting a value function on both sides of Eq. (69) leads to the fact that the counterfactual regret of $\mathsf{Reg}_T^i(S, a)$ in Eq. (66) differs from the advantage function in MARL, i.e., $Q^{i,\boldsymbol{\pi}}(s, a^i, a^{-i}) - V^{i,\boldsymbol{\pi}}(s, a^{-i})$, only by a constant of the realisation probability. As a result, the multi-agent actor-critic algorithm (Foerster et al., 2018b) can be formulated as a special type of CFR method, thus sharing a similar convergence guarantee and regret bound in two-player zero-sum games. The equivalence has also been found by (Hennes et al., 2019), where the CFR method with Hedge can be written as a particular actor-critic method that computes the policy gradient through replicator dynamics.

## 7.4 Online Markov Decision Processes

A common situation in which online learning techniques are applied is in stateless games, where the learning agent faces an identical decision problem in each trial (e.g., playing a multi-arm bandit in the casino). However, real-world decision problems often occur in a dynamic and changing environment. Such an environment is commonly captured by a state variable which, when incorporated into online learning, leads to an online MDP. Online MDP (Even-Dar et al., 2009; Yu et al., 2009; Auer et al., 2009), also called adversarial MDP[44], focuses on the problem in which the reward and transition dynamics can change over time, i.e., they are non-stationary and time-dependent.

---

[43]The self-play assumption can in fact be released. Johanson et al. (2012a) shows that in two-player zero-sum games, as long as both agents minimise their regret, not necessarily through the same algorithm, their time-average policies will converge to NE with the same regret bound $\mathcal{O}(\sqrt{T})$. An example is to let a CFR player play against a best-response opponent.

[44]The word "adversarial" is inherited from the online learning literature, i.e., *stochastic bandit* vs *adversarial bandit* (Auer et al., 2002). Adversary means there exists a virtual adversary (or, nature) who has complete control over the reward function and transition dynamics, and the adversary does not necessarily maintain a fully competitive relationship with the learning agent.

In contrast to an ordinary stochastic game, the opponent/adversary in an online MDP is not necessarily rational or even self-optimising. The aim of studying online MDP is to provide the agent with policies that perform well against every possible opponent (including but not limited to adversarial opponents), and the objective of the learning agent is to minimise its average loss during the learning process. Quantitatively, the loss is measured by how worse off the agent is compared to the best stationary policy in retrospect. The *expected regret* is thus different from Eq. (57) (unless in repeated games) and is written as

$$\mathsf{Reg}_T = \frac{1}{T} \sup_{\pi \in \Pi} \mathbb{E}_\pi \Big[ \sum_{t=1}^T R_t\big(s_t^*, a_t^*\big) - R_t\big(s_t, a_t\big) \Big] \tag{70}$$

where $\mathbb{E}_\pi$ denotes the expectation over the sequence of $(s_t^*, a_t^*)$ induced by the stationary policy $\pi$. Note that the reward function sequence and the transition kernel sequence are given by the adversary, and they are not influenced by the retrospective sequence $(s_t^*, a_t^*)$.

The goal is to find a no-regret algorithm that can satisfy $\mathsf{Reg}_T \to 0$ as $T \to \infty$ with probability 1. A sufficient condition that ensures the existence of no-regret algorithms for online MDPs is the *oblivious* assumption – both the reward functions and transition kernels are fixed in advance, although they are unknown to the learning agent. This scenario is in contrast to the stateless setting in which no-regret is achievable, even if the opponent is allowed to be *adaptive/non-oblivious*: they can choose the reward function and transition kernels in accordance to $(s_0, a_0, ..., s_t)$ from the learning agent. In short, Yu et al. (2009); Mannor & Shimkin (2003) demonstrated that in order to achieve sub-linear regret, it is essential that the changing rewards are chosen obliviously. Furthermore, Yadkori et al. (2013) showed with the example of an online shortest path problem that there does not exist a polynomial-time solution (in terms of the size of the state-action space) where both the reward functions and transition dynamics are adversarially chosen, even if the adversary is *oblivious* (i.e., it cannot adapt to the other agent's historical actions). Most recently, Ortner et al. (2020); Cheung et al. (2020) investigated online MDPs where the transitional dynamics are allowed to change slowly (i.e., the total variation does not exceed a specific budget). Therefore, the majority of existing no-regret algorithms for online MDP focus on an oblivious adversary for the reward function only. The nuances of different algorithms lie in whether the transitional kernel is assumed to be known to the learning agent and whether the feedback reward that the agent receives is in the full-information setting or in the bandit setting (i.e., one can only observe the reward of a taken action).

Two design principles can lead to no-regret algorithms that solve online MDPs with an oblivious adversary controlling the reward function. One is to leverage the local-global regret decomposition result (Even-Dar et al., 2005; 2009) [Lemma 5.4], which demonstrates that one can in fact achieve no regret globally by running a local regret-minimisation algorithm at each state; a similar result is observed for the CFR algorithm described in Eq. (66). Let $\mu^*(\cdot)$ denote the state occupancy induced by policy $\pi^*$; we then obtain the decomposition result by

$$\mathsf{Reg}_T = \sum_{s \in \mathbb{S}} \mu^*(s) \sum_{t=1}^T \underbrace{\sum_{a \in \mathbb{A}} \Big( \pi^*(a \mid s) - \pi_t(a \mid s) \Big) Q_t\big(s, a\big)}_{\text{local regret in state } s \text{ with reward function } Q_t(s, \cdot)} . \tag{71}$$

Under full knowledge of the transition function and full-information feedback about the reward, Even-Dar et al. (2009) proposed the famous *MDP-Expert (MDP-E)* algorithm, which adopts *Hedge* (Freund & Schapire, 1997) as the regret minimiser and achieves $\mathcal{O}(\sqrt{\tau^3 T \ln |\mathbb{A}|})$ regret, where $\tau$ is the bound on the mixing time of MDP [45]. For comparison, the theoretical lower bound for regret in a fixed MDP (i.e., no adversary perturbs the reward function) is $\Omega(\sqrt{|\mathbb{S}||\mathbb{A}|T})$[46] (Auer et al., 2009). Interestingly, Neu et al. (2017) showed that there in fact exists an equivalence between TRPO methods (Schulman et al., 2015) and MDP-E methods. Under bandit feedback, Neu et al. (2010) analysed *MDP-EXP3*, which achieves a regret bound of $\mathcal{O}(\sqrt{\tau^3 T |\mathbb{A}| \log |\mathbb{A}|/\beta})$, where $\beta$ is a lower bound on the probability of reaching a certain state under a

---

[45] Roughly, it can be considered as the time that a policy needs to reach the stationary status in MDPs. See a precise definition in Even-Dar et al. (2009) [Assumption 3.1].

[46] This lower bound has recently been achieved by Azar et al. (2017) up to a logarithmic factor.

given policy. Later, Neu et al. (2014) removed the dependency on $\beta$ and achieved $\mathcal{O}(\sqrt{T}\log T)$ regret. One major advantage of local-global design principle is that it can work seamlessly with function approximation methods (Bertsekas & Tsitsiklis, 1996). For example, Yu et al. (2009) eliminated the requirement of knowing the transition kernel by incorporating Q-learning methods; their proposed *Q-follow the perturbed leader (Q-FPL)* method achieved $\mathcal{O}(T^{2/3})$ regret. Abbasi-Yadkori et al. (2019) proposed *POLITEX*, which adopted a least square policy evaluation (LSPE) with linear function approximation and achieved $\mathcal{O}(T^{3/4} + \epsilon_0 T)$ regret, in which $\epsilon_0$ is the worst-case approximation error, and Cai et al. (2019a) used the same LSPE method. However, the proposed *OPPO* algorithm achieves $\mathcal{O}(\sqrt{T})$ regret.

Apart from the local-global decomposition principle, another design principle is to formulate the regret minimisation problem as an online linear optimisation (OLO) problem and then apply gradient-descent type methods. Specifically, since the regret in Eq. (71) can be further written as the inner product of $\mathsf{Reg}_T = \sum_{t=1}^{T} \langle \mu^* - \mu_t, R_t \rangle$, one can run the gradient descent method by

$$\mu_{t+1} = \arg\max_{\mu \in \mathcal{U}} \left\{ \langle \mu, R_t \rangle - \frac{1}{\eta} \mathcal{D}(\mu|\mu_t) \right\}, \tag{72}$$

where $\mathcal{U} = \left\{ \mu \in \Delta_{\mathbb{S} \times \mathbb{A}} : \sum_a \mu(s,a) = \sum_{s',a'} P(s|s',a')\mu(s',a') \right\}$ is the set of all valid stationary distributions[47], where $\mathcal{D}$ denotes a certain form of divergence and the policy can be extracted by $\pi_{t+1}(a|s) = \mu_{t+1}(s,a)/\mu(s)$. One significant advantage of this type of method is that it can flexibly handle different model constraints and extensions. If one uses Bregman divergence as $\mathcal{D}$, then online mirror descent is recovered (Nemirovsky & Yudin, 1983) and is guaranteed to achieve a nearly optimal regret for OLO problems (Srebro et al., 2011). Zimin & Neu (2013) and Dick et al. (2014) adopted a relative entropy for $\mathcal{D}$; the subsequent *online relative entropy policy search (O-REPS)* algorithm achieves an $\mathcal{O}(\sqrt{\tau T \log(|\mathbb{S}||\mathbb{A}|)})$ regret in the full-information setting and an $\mathcal{O}(\sqrt{T|\mathbb{S}||\mathbb{A}|\log(|\mathbb{S}||\mathbb{A}|)})$ regret in the bandit setting. For comparison, the aforementioned MDP-E algorithm achieves $\mathcal{O}(\sqrt{\tau^3 T \ln|\mathbb{A}|})$ and $\mathcal{O}(\sqrt{\tau^3 T |\mathbb{A}| \log|\mathbb{A}|/\beta})$, respectively. When the transition dynamics are unknown to the agent, Rosenberg & Mansour (2019) extended O-REPS by incorporating the classic idea of *optimism in the face of uncertainty* in Auer et al. (2009), and the induced *UC-O-REPS* algorithm achieved $\mathcal{O}(|\mathbb{S}|\sqrt{|\mathbb{A}|T})$ regret.

### 7.5 Turn-Based Stochastic Games

An important class of games that lie in the middle of SG and EFG is the two-player zero-sum turn-based SG (2-TBSG). In TBSG, the state space is split between two agents, $\mathbb{S} = \mathbb{S}^1 \cup \mathbb{S}^2, \mathbb{S}^1 \cap \mathbb{S}^2 = \emptyset$, and in every time step, the game is in exactly one of the states, either $\mathbb{S}^1$ or $\mathbb{S}^2$. Two players alternate taking turns to make decisions, and each state is controlled[48] by only one of the players $\pi^i : \mathbb{S}^i \to \mathbb{A}^i, i = 1, 2$. The state then transitions into the next state with probability $P : \mathbb{S}^i \times \mathbb{A}^i \to \mathbb{S}^j, i, j = 1, 2$. Given a joint policy $\boldsymbol{\pi} = (\pi^1, \pi^2)$, the first player seeks to maximise the value function $V^{\boldsymbol{\pi}(s)} = \mathbb{E}\left[ \sum_{t=0}^{\infty} \gamma^t R(s_t, \pi(s_t)) | s_0 = s \right]$, while the second player seeks to minimise it, and the saddle point is the NE of the game.

Research on 2-TBSG leads to many important finite-sample bounds, i.e., how many samples one would need before reaching the NE at a given precision, for understanding multi-agent learning algorithms. Hansen et al. (2013) extended Ye (2005; 2010)'s result from single-agent MDP to 2-TBSG and proved that the strongly polynomial time complexity of policy iteration algorithms also holds in the context of 2-TBSG if the payoff matrix is fully accessible. In the RL setting, in which the transition model is unknown, Sidford et al. (2018; 2020) provided a near-optimal Q-learning algorithm that computes an $\epsilon$-optimal strategy with high-probability given $\mathcal{O}((1-\gamma)^{-3}\epsilon^{-2})$ samples from the transition function for each state-action pair. This result of polynomial-time sample complexity is remarkable since it was believed to hold for only single-agent MDPs. Recently, Jia et al. (2019) showed that if the transition model can be embedded in some state-action feature space, i.e., $\exists \psi_k(s')$ such that $P(s'|s,a) = \sum_{k=1}^{K} \phi_k(s,a)\psi_k(s'), \forall s' \in \mathbb{S}, (s,a) \in \mathbb{S} \times \mathbb{A}$, then the sample complexity of the two-player Q-learning algorithm towards finding an $\epsilon$-NE is only linear to the number of features $\mathcal{O}(K/(\epsilon^2(1-\gamma)^4))$.

---

[47]In the online MDP literature, it is generally assumed that every policy reaches its stationary distribution immediately; see the policy mixing time assumption in Yu et al. (2009) [Assumption 2.1].

[48]Note that since the game is turned based, the Nash policies are deterministic.

---

**Algorithm 1** A General Solver for Open-Ended Meta-Games

---

1: **Initialise:** the "high-level" policy set $\mathbb{S} = \prod_{i \in \mathcal{N}} \mathbb{S}^i$, the meta-game payoff $\mathbf{M}, \forall S \in \mathbb{S}$, and meta-policy $\boldsymbol{\pi}^i = \text{UNIFORM}(\mathbb{S}^i)$.
2: **for** iteration $t \in \{1, 2, ...\}$ **do**:
3:     **for** each player $i \in \mathcal{N}$ **do**:
4:         Compute the meta-policy $\boldsymbol{\pi}_t$ by meta-game solver $\mathcal{S}(\mathbf{M}_t)$.
5:         Find a new policy against others by Oracle: $S_t^i = \mathcal{O}^i(\boldsymbol{\pi}_t^{-i})$.
6:         Expand $\mathbb{S}_{t+1}^i \leftarrow \mathbb{S}_t^i \cup \{S_t^i\}$ and update meta-payoff $\mathbf{M}_{t+1}$.
7:     **terminate if:** $\mathbb{S}_{t+1}^i = \mathbb{S}_t^i, \forall i \in \mathcal{N}$.
8: **Return:** $\boldsymbol{\pi}$ and $\mathbb{S}$.

---

**Table 3:** Variations of Different Meta-Game Solvers

| Method | $\mathcal{S}$ | $\mathcal{O}$ | Game type |
|---|---|---|---|
| Self-play (Fudenberg et al., 1998) | $[0, ..., 0, 1]^N$ | $\mathbf{Br}(\cdot)$ | multi-player potential |
| GWFP (Leslie & Collins, 2006) | UNIFORM | $\mathbf{Br}_\epsilon(\cdot)$ | two-player zero-sum / potential |
| Double Oracle (McMahan et al., 2003) | NE | $\mathbf{Br}(\cdot)$ | two-player zero-sum |
| PSRO$_N$ (Lanctot et al., 2017) | NE | $\mathbf{Br}_\epsilon(\cdot)$ | two-player zero-sum |
| PSRO$_{rN}$ (Balduzzi et al., 2019) | NE | **Rectified $\mathbf{Br}_\epsilon(\cdot)$** | symmetric zero-sum |
| $\alpha$-PSRO (Muller et al., 2019) | $\alpha$-Rank | $\mathbf{PBr}(\cdot)$ | multi-player general-sum |

All the above works focus on the offline domain, where they assume that there exists an *oracle* that can unconditionally provide state-action transition samples. Wei et al. (2017) studied an online setting in an averaged-reward two-player SG. They achieved a polynomial sample-complexity bound if the opponent plays an optimistic best response, and a sublinear regret round against an arbitrary opponent.

### 7.6 Open-Ended Meta-Games

In solving real-world zero-sum games, such as GO or StarCraft, since the number of atomic pure strategy can be prohibitively large, one feasible approach instead is to focus on *meta-games*. A meta-game is constructed by simulating games that cover combinations of "high-level" policies in the policy space (e.g., "bluff" in Poker or "rushing" in StarCraft), with entries corresponding to the players' empirical payoffs under a certain joint "high-level" policy profile; therefore, meta-game analysis is often called as *empirical game-theoretic analysis (EGTA)* (Wellman, 2006; Tuyls et al., 2018). Analysing meta-games is a practical approach to tackling games that have huge pure-strategy space, since the number of "high-level" policies is usually far smaller than the number of pure strategies. For example, the number of tactics in StarCraft is at hundreds, compared to the vast raw action space of approximately $10^8$ possibilities (Vinyals et al., 2017). Traditional game-theoretical concepts such as NE can still be computed on meta-games, but in a much more scalable manner; this is because the number of "higher-level" strategies in the meta-game is usually far smaller than the number of atomic actions of the underlying game. Furthermore, it has been shown that an $\epsilon$-NE of the meta-game is in fact a $2\epsilon$-NE of the underlying game (Tuyls et al., 2018). Meta-games are often **open-ended** because in general there exists an infinite number of policies to play a real-world game, and, as new strategies will be discovered and added to agents' strategy sets during training, the dimension of the meta-game payoff table

will also be expanded. If one writes the game evaluation engine as $\phi : \mathbb{S}^1 \times \mathbb{S}^2 \to \mathbb{R}$ such that if $S^1 \in \mathbb{S}^1$ beats $S^2 \in \mathbb{S}^2$, we have $\phi(S^1, S^2) > 0$, and $\phi < 0, \phi = 0$ refers to losses and ties, then the meta-game payoff can be represented by $\mathbf{M} = \left\{ \phi(S^1, S^2) : (S^1, S^2) \in \mathbb{S}^1 \times \mathbb{S}^2 \right\}$. The sets of $\mathbb{S}^1$ and $\mathbb{S}^2$ can be regarded as, for example, two populations of deep neural networks (DNNs) and each $S^1, S^2$ is a DNN with independent weights. In such a context, the goal of learning in meta-games is to find $\mathbb{S}^i$ and policy $\boldsymbol{\pi}^i \in \Delta(\mathbb{S}^i)$ such that the *exploitability* can be minimised, which is,

$$\text{Exploitability}\,(\boldsymbol{\pi}) = \sum_{i \in \{1,2\}} \left[ \mathbf{M}^i \big( \mathbf{Br}^i(\boldsymbol{\pi}^{-i}), \boldsymbol{\pi}^{-i} \big) - \mathbf{M}^i(\boldsymbol{\pi}) \right]. \tag{73}$$

It is easy to see Eq. (73) reaches zero when $\boldsymbol{\pi}$ is a NE.

A general solver for open-ended meta-games is the *policy space response oracle (PSRO)* (Lanctot et al., 2017). Inspired by the *double oracle* algorithm (McMahan et al., 2003), which leverages the *Benders' decomposition* (Benders, 1962) on solving large-scale linear programming for two-player zero-sum games, PSRO is a direct extension of double oracle (McMahan et al., 2003) by incorporating an RL subroutine as an approximate best response. Specifically, one can write PSRO and its variations in Algorithm 1, which essentially involves an iterative two-step process of solving for the meta-policy first (e.g., Nash over the meta-game), and then based on the meta-policy, finding a new better-performing policy, against the opponent's current meta-policy, to augment the existing population. The meta-policy solver, denoted as $\mathcal{S}(\cdot)$, computes a joint meta-policy profile $\boldsymbol{\pi}$ based on the current payoff $\mathbf{M}$ where different solution concepts can be adopted (e.g., NE). Finding a new policy is equivalent to solving a single-player optimisation problem given opponents' policy sets $\mathbb{S}^{-i}$ and meta-policies $\boldsymbol{\pi}^{-i}$, which are fixed and known. One can regard a new policy as given by an *Oracle*, denoted by $\mathcal{O}$. In two-player zero-sum cases, an oracle represents $\mathcal{O}^1(\boldsymbol{\pi}^2) = \{S^1 : \sum_{S^2 \in \mathbb{S}^2} \boldsymbol{\pi}^2(S^2) \cdot \phi(S^1, S^2) > 0\}$. Generally, Oracles can be implemented through optimisation subroutines such as RL algorithms. Finally, after a new policy is found, the payoff table $\mathbf{M}$ is expanded, and the missing entries are filled by running new game simulations. The above two-step process loops over each player at each iteration, and it terminates if no new policies can be found for any players.

Algorithm 1 is a general framework, with appropriate choices of meta-game solver $\mathcal{S}$ and Oracle $\mathcal{O}$, it can represent solvers for different types of meta-games. We summarise variations of meta-game solvers in Table 3. For example, it is trivial to see that FP/GWFP is recovered when $\mathcal{S} = \text{UNIFORM}(\cdot)$ and $\mathcal{O}^i = \mathbf{Br}^i(\cdot)/\mathbf{Br}^i_\epsilon(\cdot)$. The double oracle (McMahan et al., 2003) and PSRO methods (Lanctot et al., 2017) refer to the cases when the meta-solver computes NE. On solving symmetric zero-sum games (i.e., $\mathbb{S}^1 = \mathbb{S}^2$, and $\phi(S^1, S^2) = -\phi(S^2, S^1), \forall S^1, S^2 \in \mathbb{S}^1$), Balduzzi et al. (2019) proposed the *rectified best response* to promote behavioural diversity, written as

$$\textbf{Rectified } \mathbf{Br}_\epsilon\big(\boldsymbol{\pi}^2\big) \subseteq \arg\max_{S^1} \sum_{S^2 \in \mathbb{S}^2} \boldsymbol{\pi}^{2,*}(S^2) \cdot \lfloor \phi(S^1, S^2) \rfloor_+. \tag{74}$$

Through rectifying only the positive values on $\phi(S^1, S^2)$ in Eq. (74), player 1 is encouraged to amplify its strengths and ignore its weaknesses in finding a new policy when it plays with the NE of player 2 during training; this turns out to be a critical component to tackle zero-sum games with strong non-transitive dynamics[49].

Double oracle and PSRO methods can only solve zero-sum games. When it comes to multi-player general-sum games, a new solution concept named $\alpha$-Rank (Omidshafiei et al., 2019) can be used to replace the intractable NE. The idea of $\alpha$-Rank is built on the *response graph* of a game. On the response graph, each joint pure-strategy profile is a node, and a directed edge points from node $\sigma \in \mathbb{S}$ to node $S \in \mathbb{S}$ if 1) $\sigma$ and $S$ differ in only one single player's strategy, and 2) that deviating player, denoted by $i$, benefits from deviating from $S$ to $\sigma$ such that $\mathbf{M}^i(\sigma) > \mathbf{M}^i(S)$. The *sink strongly-connected components (SSCC)* nodes on the response graph that have only incoming edges but no outgoing edges are of great interest. To find those

---

[49]Any symmetric zero-sum games consist of both transitive and non-transitive components (Balduzzi et al., 2019). A game is transitive if the $\phi$ can be represented by a monotonic rating function $f$ such that performance on the game is the difference in ratings: $\phi(S^1, S^2) = f(S^1) - f(S^2)$, and it is non-transitive if $\phi$ satisfies $\int_{S^2 \in \mathbb{S}} \phi(S^1, S^2) \cdot dS^2 = 0$, meaning that winning against some strategies will be counterbalanced with losses against other strategies in the population.

SSCC nodes, $\alpha$-Rank constructs a random walk along the directed response graph, which can be equivalently described by a Markov chain, with the transition probability matrix $\boldsymbol{C}$ being:

$$\boldsymbol{C}_{S,\sigma} = \begin{cases} \eta \dfrac{1-\exp\left(-\alpha\left(\mathbf{M}^k(\sigma)-\mathbf{M}^k(s)\right)\right)}{1-\exp\left(-\alpha m\left(\mathbf{M}^k(\sigma)-\mathbf{M}^k(s)\right)\right)} & \text{if } \mathbf{M}^k(\sigma) \neq \mathbf{M}^k(S) \\ \dfrac{\eta}{m} & \text{otherwise} \end{cases} ,$$

$$\boldsymbol{C}_{S,S} = 1 - \sum_{i\in\mathcal{N}} \boldsymbol{C}_{S,\sigma} \tag{75}$$

$\eta = (\sum_{i\in\mathcal{N}}(|S^i|-1))^{-1}, m \in \mathbb{N}, \alpha > 0$ are three constants. Large $\alpha$ ensures the Markov chain is irreducible, and thus guarantees the existence and uniqueness of the $\alpha$-Rank solution, which is the resulting unique stationary distribution $\boldsymbol{\pi}$ of the Markov chain, $\boldsymbol{C}^\top \boldsymbol{\pi} = \boldsymbol{\pi}$. The probability mass of each joint strategy in $\boldsymbol{\pi}$ can be interpreted as the longevity of that strategy during an evolution process (Omidshafiei et al., 2019). The main advantage of $\alpha$-Rank is that it is unique and its solution is $P$-complete even on multi-player general-sum games. $\alpha^\alpha$-Rank developed by Yang et al. (2019a) computes $\alpha$-Rank based on stochastic gradient methods such that there is no need to store the whole transition matrix in Eq. (75) before getting the final output of $\boldsymbol{\pi}$, this is particularly important when meta-games are prohibitively large in real-world domains.

When PSRO adopts $\alpha$-Rank as the meta-solver, it is found that a simple best response fails to converge to the SSCC of a response graph before termination (Muller et al., 2019). To suit $\alpha$-Rank, Muller et al. (2019) later proposed *preference-based best response oracle*, written as

$$\mathbf{PBr}^i\left(\boldsymbol{\pi}^{-i}\right) \subseteq \arg\max_{\sigma\in\mathbb{S}^i} \mathbb{E}_{S^{-i}\sim\boldsymbol{\pi}^{-i}}\left[\mathbb{1}\left[\mathbf{M}^i\left(\sigma,S^{-i}\right) > \mathbf{M}^i\left(S^i,S^{-i}\right)\right]\right], \tag{76}$$

and the combination of $\alpha$-Rank with $\mathbf{PBr}(\cdot)$ in Eq. (76) is called $\alpha$-PSRO. Due to the tractability of $\alpha$-Rank on general-sum games, the $\alpha$-PSRO is credited as a generalised training approach for multi-agent learning.

## 8 Learning in General-Sum Games

Solving general-sum SGs entails an entirely different level of difficulty than solving team games or zero-sum games. In a static two-player normal-form game, finding the NE is known to be $PPAD$-complete (Chen & Deng, 2006).

### 8.1 Solutions by Mathematical Programming

To solve a two-player general-sum discounted stochastic game with discrete states and discrete actions, Filar & Vrieze (2012) [Chapter 3.8] formulated the problem as a nonlinear programme; the matrix form is written as follows:

$$\min_{\mathbf{V},\boldsymbol{\pi}} f(\mathbf{V},\boldsymbol{\pi}) = \sum_{i=1}^2 \mathbf{1}_{|\mathbb{S}|}^T\left[V^i - \left(\mathbf{R}^i(\boldsymbol{\pi}) + \gamma \cdot \mathbf{P}(\boldsymbol{\pi})V^i\right)\right]$$

$$\text{s.t.} \quad \begin{aligned} &\text{(a) } \pi^2(s)^T\left[\mathbf{R}^1(s) + \gamma \cdot \textstyle\sum_{s'} \mathbf{P}(s'|s)V^1(s')\right] \leq V^1(s)\mathbf{1}_{|\mathbb{A}^1|}^T, \quad \forall s \in \mathbb{S} \\ &\text{(b) } \left[\mathbf{R}^2(s) + \gamma \cdot \textstyle\sum_{s'} \mathbf{P}(s'|s)V^2(s')\right]\pi^1(s) \leq V^2(s)\mathbf{1}_{|\mathbb{A}^2|}, \quad \forall s \in \mathbb{S} \\ &\text{(c) } \pi^1(s) \geq \mathbf{0}, \quad \pi^1(s)^T\mathbf{1}_{|\mathbb{A}^1|} = 1, \quad \forall s \in \mathbb{S} \\ &\text{(d) } \pi^2(s) \geq \mathbf{0}, \quad \pi^2(s)^T\mathbf{1}_{|\mathbb{A}^2|} = 1, \quad \forall s \in \mathbb{S} \end{aligned} \tag{77}$$

where

- $\mathbf{V} = \langle V^i : i = 1, 2 \rangle$ is the vector of agents' values over all states, $V^i = \langle V^i(s) : s \in \mathbb{S} \rangle$ is the value vector for the $i$-th agent.

- $\boldsymbol{\pi} = \langle \pi^i : i = 1, 2 \rangle$ and $\pi^i = \langle \pi^i(s) : s \in \mathbb{S} \rangle$, where $\pi^i(s) = \langle \pi^i(a|s) : a \in \mathbb{A}^i \rangle$ is the vector representing the stochastic policy in state $s \in \mathbb{S}$ for the $i$-th agent.

- $\mathbf{R}^i(s) = \left[ R^i \left( s, a^1, a^2 \right) : a^1 \in \mathbb{A}^1, a^2 \in \mathbb{A}^2 \right]$ is the reward matrix for the $i^{\text{th}}$ agent in state $s \in \mathbb{S}$. The rows correspond to the actions of the second agent, and the columns correspond to those of the first agent. With a slight abuse of notation, we use $\mathbf{R}^i(\boldsymbol{\pi}) = \mathbf{R}^i \left( \langle \pi^1, \pi^2 \rangle \right) = \left\langle \pi^2(s)^T \mathbf{R}^i(s) \pi^1(s) : s \in \mathbb{S} \right\rangle$ to represent the expected reward vector over all states under joint policy $\boldsymbol{\pi}$.

- $\mathbf{P}(s'|s) = \left[ P(s'|s, \boldsymbol{a}) : \boldsymbol{a} = \langle a^1, a^2 \rangle, a^1 \in \mathbb{A}^1, a^2 \in \mathbb{A}^2 \right]$ is a matrix representing the probability of transitioning from the current state $s \in \mathbb{S}$ to the next state $s' \in \mathbb{S}$. The rows represent the actions of the second agent, and the columns represent those of the first agent. With a slight abuse of notation, we use $\mathbf{P}(\boldsymbol{\pi}) = \mathbf{P} \left( \langle \pi^1, \pi^2 \rangle \right) = \left[ \pi^2(s)^T \mathbf{P}(s'|s) \pi^1(s) : s \in \mathbb{S}, s' \in \mathbb{S} \right]$ to represent the expected transition probability over all state pairs under joint policy $\boldsymbol{\pi}$.

This is a nonlinear programme because the inequality constraints in the optimisation problem are quadratic in $\mathbf{V}$ and $\boldsymbol{\pi}$. The objective function in Eq. (77) aims to minimise the TD error for a given policy $\boldsymbol{\pi}$ over all states, similar to the policy evaluation step in the traditional policy iteration method, and the constraints of ($a$) and ($b$) in Eq. (77) act as the policy improvement step, which satisfies the equation when the optimal value function is achieved. Finally, constraints ($c$) and ($d$) ensure the policy is properly defined.

Although the NE is proved to exist in general-sum SGs in the form of stationary strategies, solving Eq. (77) in the two-player case is notoriously challenging. First, Eq. (77) has a non-convex feasible region; second, only the global optimum[50] of Eq. (77) corresponds to the NE of SGs, while the common gradient-descent type of methods can only guarantee convergence to a local minimum. Apart from the efforts by Filar & Vrieze (2012), Breton et al. (1986) [Chapter 4] developed a formulation that has nonlinear objectives but linear constraints. Furthermore, Dermed & Isbell (2009) formulated the NE solution as multi-objective linear program. Herings et al. (2004); Herings & Peeters (2010) proposed an algorithm in which a *homotopic path* between the equilibrium points of $N$ independent MDPs and the $N$-player SG is traced numerically. This approach yields a Nash equilibrium point of the stochastic game of interest. However, all these methods are tractable only in small-size SGs with at most tens of states and only two players.

## 8.2 Solutions by Value-Based Methods

A series of value-based methods have been proposed to address general-sum SGs. A majority of these methods adopt classic Q-learning (Watkins & Dayan, 1992) as a centralised controller, with the differences being what solution concept the central Q-learner should apply to guide the agents to converge in each iteration. For example, the Nash-Q learner in Eqs. (19 & 20) applies NE as the solution concept, the correlated-Q learner adopts correlated equilibrium (Greenwald et al., 2003), and the friend-or-foe learner considers both cooperative (see Eq. (35)) and competitive equilibrium (see Eq. (54)) (Littman, 2001a). Although many algorithms come with convergence guarantees, the corresponding assumptions are often overly restrictive to be applicable in general. When Nash-Q learning was first proposed (Hu et al., 1998), it required the NE of the SG be unique such that the convergence property could hold. Though strong, this assumption was still noted by Bowling (2000) to be insufficient to justify the convergence of the Nash-Q algorithm. Later, Hu & Wellman (2003) corrected her convergence proof by tightening the assumption even further; the uniqueness of the NE must hold for every single stage game encountered during state transitions. Years later, a strikingly negative result by Zinkevich et al. (2006) concluded that the entire class of value-iteration methods could be excluded from consideration for computing stationary equilibria, including both NE and correlated equilibrium, in general-sum SGs. Unlike those in single-agent RL, the Q values in the multi-agent case are inherently defective for reconstructing the equilibrium policy.

## 8.3 Solutions by Two-Timescale Analysis

In addition to the centralised Q-learning approach, decentralised Q-learning algorithms have recently received considerable attention because of their potential for scalability. Although independent learners have been accused of having convergence issues (Tan, 1993), decentralised methods have made substantial progress with the help of two-timescale stochastic analysis (Borkar, 1997) and its application in RL (Borkar, 2002).

---

[50]Note that in the zero-sum case, every local optimum is global.

Two-timescale stochastic analysis is a set of tools certifying that, in a system with two coupled stochastic processes that evolve at different speeds, if the fast process converges to a unique limit point for any particular fixed value of the slow process, we can, quantitatively, analyse the asymptotic behaviour of the algorithm as if the fast process is always fully calibrated to the current value of the slow process (Borkar, 1997). As a direct application, Leslie et al. (2003); Leslie & Collins (2005) noted that independent Q-learners with agent-dependent learning rates could break the symmetry that leads to the non-convergent limited cycles; as a result, they can converge almost surely to the NE in two-player collaboration games, two-player zero-sum games, and multi-player matching pennies. Similarly, Prasad et al. (2015) introduced a two-timescale update rule that ensures the training dynamics reach a stationary local NE in general-sum SGs if the critic learns faster than the actor. Later, Perkins et al. (2015) proposed a distributed actor-critic algorithm that enjoys provable convergence in solving static potential games with continuous actions. Similarly, Arslan & Yüksel (2016) developed a two-timescale variant of Q-learning that is guaranteed to converge to an equilibrium in SGs with weakly acyclic characteristics, which generalises potential games. Other applications include developing two-timescale update rules for training GANs (Heusel et al., 2017) and developing a two-timescale algorithm with guaranteed asymptotic convergence to the Stackelberg equilibrium in general-sum Stackelberg games.

## 8.4 Solutions by Policy-Based Methods

Convergence to NE via direct policy search has been extensively studied; however, early results were limited mainly by stateless two-player two-action games (Singh et al., 2000; Bowling & Veloso, 2002; Bowling, 2005; Conitzer & Sandholm, 2007; Abdallah & Lesser, 2008; Zhang & Lesser, 2010). Recently, GAN training has posed a new challenge, thereby rekindling interest in understanding the policy gradient dynamics of continuous games (Nagarajan & Kolter, 2017; Heusel et al., 2017; Mescheder et al., 2018; 2017).

Analysing gradient-based algorithms through dynamic systems (Shub, 2013) is a natural approach to yield more significant insights into convergence behaviour. However, a fundamental difference is observed when one attempts to apply the same analysis from the single-agent case to the multi-agent case because the combined dynamics of gradient-based learning schemes in multi-agent games do not necessarily correspond to a proper *gradient flow* – a critical premise for almost sure convergence to a local minimum. In fact, the difficulty of solving general-sum continuous games is exacerbated by the usage of deep networks with stochastic gradient descent. In this context, a key equilibrium concept of interest is the *local NE* (Ratliff et al., 2013) or *differential NE* (Ratliff et al., 2014), defined as follows.

**Definition 10 (Local Nash Equilibrium)** *For an $N$-player continuous game denoted by $\{\ell_i : \mathbb{R}^d \to R\}_{i \in \{1,...,N\}}$ with each agent's loss $\ell_i$ being twice continuously differentiable, the parameters are $\boldsymbol{w} = (\boldsymbol{w}_1, ..., \boldsymbol{w}_n) \in \mathbb{R}^d$, and each player controls $\boldsymbol{w}_i \in \mathbb{R}^{d_i}$, $\sum_i d_i = d$. Let $\boldsymbol{\xi}(\mathbf{w}) = (\nabla_{\mathbf{w}_1}\ell_1, \ldots, \nabla_{\mathbf{w}_n}\ell_n) \in \mathbb{R}^d$ be the simultaneous gradient of the losses w.r.t. the parameters of the respective players, and let $\mathbf{H}(\mathbf{w}) := \nabla_{\mathbf{w}} \cdot \boldsymbol{\xi}(\mathbf{w})^\top$ be the $(d \times d)$ Hessian matrix of the gradient, written as*

$$\mathbf{H}(\mathbf{w}) = \begin{pmatrix} \nabla^2_{\mathbf{w}_1}\ell_1 & \nabla^2_{\mathbf{w}_1,\mathbf{w}_2}\ell_1 & \cdots & \nabla^2_{\mathbf{w}_1,\mathbf{w}_n}\ell_1 \\ \nabla^2_{\mathbf{w}_2,\mathbf{w}_1}\ell_2 & \nabla^2_{\mathbf{w}_2}\ell_2 & \cdots & \nabla^2_{\mathbf{w}_2,\mathbf{w}_n}\ell_2 \\ \vdots & & & \vdots \\ \nabla^2_{\mathbf{w}_n,\mathbf{w}_1}\ell_n & \nabla^2_{\mathbf{w}_n,\mathbf{w}_2}\ell_n & \cdots & \nabla^2_{\mathbf{w}_n}\ell_n \end{pmatrix}$$

*where $\nabla^2_{\mathbf{w}_i,\mathbf{w}_j}\ell_k$ is the $(d_i \times d_j)$ block of 2nd-order derivatives. A differentiable NE for the game is $\boldsymbol{w}^*$ if $\boldsymbol{\xi}(\mathbf{w}^*) = 0$ and $\nabla^2_{\mathbf{w}_i}\ell_i \succ 0$, $\forall i \in \{1, ..., N\}$; furthermore, this result is a local NE if $\det \mathbf{H}(\mathbf{w}^*) \neq 0$.*

A recent result by Mazumdar & Ratliff (2018) suggested that gradient-based algorithms can almost surely avoid a subset of local NE in general-sum games; even worse, there exist non-Nash stationary points. As a tentative treatment, Balduzzi et al. (2018a) applied *Helmholtz decomposition*[51] to decompose the game Hessian $\mathbf{H}(\boldsymbol{w})$ into a potential part plus a Hamiltonian part. Based on the decomposition, they designed a

---

[51]This approach is similar in ideology to the work by Candogan et al. (2011), where they leverage the combinatorial Hodge decomposition to decompose any multi-player normal-form game into a potential game plus a harmonic game. However, their equivalence is an open question.

gradient-based method to address each part and combined them into *symplectic gradient adjustment (GDA)*, which is able to find all local NE for zero-sum games and a subset of local NE for general-sum games. More recently, Chasnov et al. (2019) separately considered the cases of 1) agents with oracle access to the exact gradient $\boldsymbol{\xi}(\mathbf{w})$ and 2) agents with only an unbiased estimator for $\boldsymbol{\xi}(\mathbf{w})$. In the first case, they provided asymptotic and finite-time convergence rates for the gradient-based learning process to reach the differential NE. In the second case, they derived concentration bounds guaranteeing with high probability that agents will converge to a neighbourhood of a stable local NE in finite time. In the same framework, Fiez et al. (2019) studied Stackelberg games in which agents take turns to conduct the gradient update rather than acting simultaneously and established the connection under which the equilibrium points of simultaneous gradient descent are Stackelberg equilibria in zero-sum games. Mertikopoulos & Zhou (2019) investigated the local convergence of no-regret learning and found local NE is attracting under gradient play if and only if a NE satisfies a property known as *variational stability*. This idea is inspired by the seminal notion of *evolutionary stability* observed in animal populations (Smith & Price, 1973).

Finally, it is worth highlighting that the above theoretical analysis of the performance of gradient-based methods on stateless continuous games cannot be taken for granted in SGs. The main reason is that the assumption on the differentiability of the loss function required in continuous games may not hold in general-sum SGs. As clearly noted by Mazumdar et al. (2019a); Fazel et al. (2018); Zhang et al. (2019c), even in the extreme setting of linear-quadratic games, the value functions are not guaranteed to be globally smooth (w.r.t. each agent's policy parameter).

# 9 Learning in Games when $N \to +\infty$

As detailed in Section 4, designing learning algorithms in a multi-agent system with $N \gg 2$ is a challenging task. One major reason is that the solution concept, such as Nash equilibrium, is difficult to compute in general due to the curse of dimensionality of the multi-agent problem itself. However, if one considers a continuum of agents with $N \to +\infty$, then the learning problem becomes surprisingly tractable. The intuition is that one can effectively transform a many-body interaction problem into a two-body interaction problem (i.e., agent vs the population mean) via mean-field approximation.

The idea of mean-field approximation, which considers the behaviour of large numbers of particles where individual particles have a negligible impact on the system, originated from physics. Important applications include solving Ising models[52] (Weiss, 1907; Kadanoff, 2009), or more recently, understanding the learning dynamics of over-parameterised deep neural networks (Lu et al., 2020b; Song et al., 2018; Sirignano & Spiliopoulos, 2020; Hu et al., 2019). In the game theory and MARL context, mean-field approximation essentially enables one to think of the interactions between every possible permutation of agents as an interaction between each agent itself and the aggregated mean effect of the population of the other agents, such that the $N$-player game ($N \to +\infty$) turns into a "two"-player game. Moreover, under *the law of large numbers* and *the theory of propagation of chaos* (Gärtner, 1988; McKean, 1967; Sznitman, 1991), the aggregated version of the optimisation problem in Eq. (80) asymptotically approximates the original $N$-player game.

The assumption in the mean-field regime that each agent responds only to the mean effect of the population may appear rather limited initially; however, for many real-world applications, agents often cannot access the information of all other agents but can instead know the global information about the population. For example, in high-frequency trading in finance (Cardaliaguet & Lehalle, 2018; Lehalle & Mouzouni, 2019), each trader cannot know every other trader's position in the market, although they have access to the aggregated order book from the exchange. Another example is real-time bidding for online advertisements

---

[52]An Ising model is a model used to study magnetic phase transitions under different system temperatures. In a 2D Ising model, one can imagine the magnetic spins are laid out on a lattice, and each spin can have one of two directions, either up or down. When the system temperature is high, the direction of the spins is chaotic, and when the temperature is low, the directions of the spins tend to be aligned. Without the mean-field approximation, computing the probability of the spin direction is a combinatorial hard problem; for example, in a $5 \times 5$ 2D lattice, there are $2^{25}$ possible spin configurations. A successful approach to solving the Ising model is to observe the phase change under different temperatures and compare it against the ground truth.

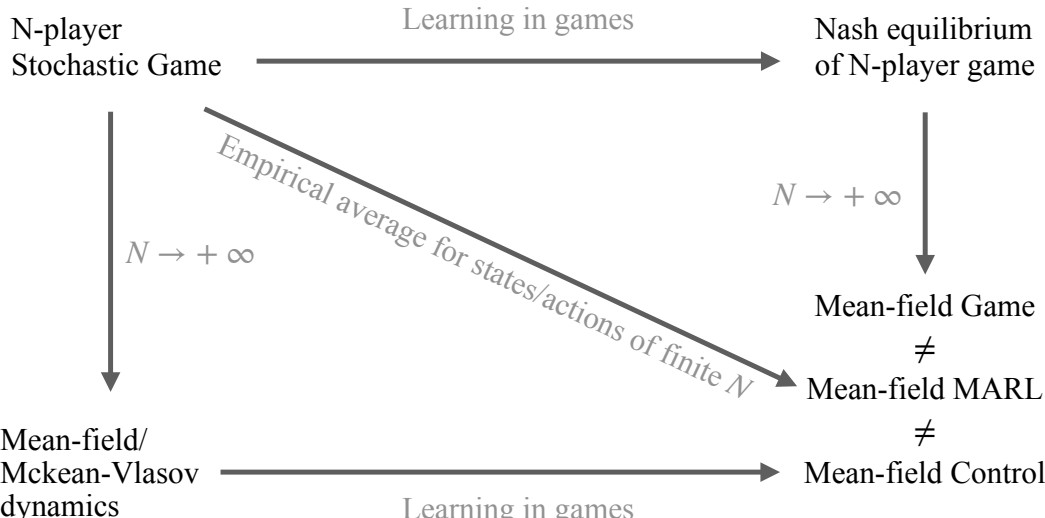

**Figure 11:** Relations of mean-field learning algorithms in games with large $N$.

(Guo et al., 2019; Iyer et al., 2014), in which participants can only observe, for example, the second-best prize that wins the auction but not the individual bids from other participants.

There is a subtlety associated with types of games in which one applies the mean-field theory. If one applies the mean-field type theory in non-cooperative[53] games, in which agents act independently to maximise their own individual reward, and the solution concept is NE, then the scenario is usually referred to as a *mean-field game (MFG)* (Jovanovic & Rosenthal, 1988; Huang et al., 2006; Lasry & Lions, 2007; Guéant et al., 2011). If one applies mean-field theory in cooperative games in which there exists a central controller to control all agents cooperatively to reach some Pareto optima, then the situation is usually referred to as *mean-field control (MFC)* (Bensoussan et al., 2013; Andersson & Djehiche, 2011), or *McKean-Vlasov dynamics (MKV)* control. If one applies the mean-field approximation to solve a standard SG through MARL, specifically, to factorise each agent's reward function or the joint-Q function, such that they depend only on the agent's local state and the mean action of others, then it is called *mean-field MARL (MF-MARL)* (Yang et al., 2018b; Subramanian et al., 2020; Zhou et al., 2019).

Despite the difference in the applicable game types, technically, the differences among MFG/MFC/MF-MARL can be elaborated from the perspective of the order in which the equilibrium is learned (optimised) and the limit as $N \to +\infty$ is taken (Carmona et al., 2013). MFG learns the equilibrium of the game first and then takes the limit as $N \to +\infty$, while MFC takes the limit first and optimises the equilibrium later. MF-MARL is somewhat in between. The mean-field in MF-MARL refers to the empirical average of the states and/or actions of a *finite* population; $N$ does not have to reach infinity, though the approximation converges asymptotically to the original game when $N$ is large. This result is in contrast to the mean-field in MFG and MFC, which is essentially a probability distribution of states and/or actions of an *infinite* population (i.e., the Mckean-Vlasov dynamics). Before providing more details, we summarise the relationships of MFG, MFC, and MF-MARL in Figure 11. Readers are recommended to revisit their differences after finishing reading the below subsections.

## 9.1 Non-cooperative Mean-Field Game

MFGs have been widely studied in different domains, including physics, economics, and stochastic control (Carmona et al., 2018; Guéant et al., 2011). An intuitive example to quickly illustrate the idea of MFG is the problem of *when does the meeting start* (Guéant et al., 2011). For a meeting in the real world, people often schedule a calendar time $t$ in advance, and the actual start time $T$ depends on when the

---

[53]Note that the word "non-cooperative" does not mean agents cannot collaborate to complete a task, it means agents cannot collude to form a coalition: they have to behave independently.

majority of participants (e.g., 90%) arrive. Each participant plans to arrive at $\tau^i$, and the actual arrival time, $\tilde{\tau}^i = \tau^i + \sigma^i \epsilon^i$, is often influenced by some uncontrolled factors $\sigma^i \epsilon^i, \epsilon^i \sim \mathcal{N}(0, 1)$, such as weather or traffic. Assuming all players are rational, they do not want to be later than either $t$ or $T$; moreover, they do not want to arrive too early and have to wait. The cost function of each individual can be written as $c^i(t, T, \tilde{\tau}^i) = \mathbb{E}\big[ \alpha \lfloor \tilde{\tau}^i - t \rfloor_+ + \beta \lfloor \tilde{\tau}^i - T \rfloor_+ + \gamma \lfloor T - \tilde{\tau}^i \rfloor_+ \big]$, where $\alpha, \beta, \gamma$ are constants. The key question to ask is when is the best time for an agent to arrive, as a result, when will the meeting actually start, i.e., what is $T$?

The challenge of the above problem lies in the coupled relationship between $T$ and $\tau^i$; that is, in order to compute $T$, we need to know $\tau^i$, which is based on $T$ itself. Therefore, solving the time $T$ is essentially equivalent to finding the fixed point, if it exists, of the stochastic process that generates $T$. In fact, $T$ can be effectively computed through a two-step iterative process, and we denote as $\Gamma^1$ and $\Gamma^2$. At $\Gamma^1$, given the current[54] value of $T$, each agent solves their optimal arrival time $\tau^i$ by minimising their cost $R^i(t, T, \tilde{\tau}^i)$. At $\Gamma^2$, agents calibrate the new estimate of $T$ based on all $\tau^i$ values that were computed in $\Gamma^1$. $\Gamma^1$ and $\Gamma^2$ continue iterating until $T$ converges to a fixed point, i.e., $\Gamma^2 \circ \Gamma^1(T^*) = T^*$. The key insight is that the interaction with other agents is captured simply by the mean-field quantity. Since the meeting starts only when 90% of the people arrive, if one considers a continuum of players with $N \to +\infty$, $T$ becomes the $90th$ quantile of a distribution, and each agent can easily find the best response. This result contrasts to the cases of a finite number of players, in which the ordered statistic is intractable, especially when $N$ is large (but still finite).

Approximating an $N$-player SG by letting $N \to +\infty$ and letting each player choose an optimal strategy in response to the population's macroscopic information (i.e., the mean field), though analytically friendly, is not cost-free. In fact, MFG makes two major assumptions: 1) the impact of each player's action on the outcome is infinitesimal, resulting in all agents being identical, interchangeable, and indistinguishable; 2) each player maintains *weak interactions* with others only through a mean field, denoted by $L^i \in \Delta^{|\mathbb{S}||\mathbb{A}|}$, which is essentially a population state-action joint distribution

$$L^i = \left( \mu^{-i}(\cdot), \alpha^{-i}(\cdot) \right) = \lim_{N \to +\infty} \left( \frac{\sum_{j \neq i} \mathbb{1}(s^j = \cdot)}{N - 1}, \frac{\sum_{j \neq i} \mathbb{1}(a^j = \cdot)}{N - 1} \right) \tag{78}$$

where $s^j$ and $a^j$ player $j$'s local state[55] and local action. Therefore, for SGs that do not share the homogeneity assumption[56] and weak interaction assumption, MFG is not an effective approximation. Furthermore, since agents have no identity in MFG, one can choose a representative agent (the agent index is thus omitted) and write the formulation[57] of the MFG as

$$V\left( s, \pi, \{L_t\}_{t=0}^{\infty} \right) := \mathbb{E}\left[ \sum_{t=0}^{\infty} \gamma^t R\left( s_t, a_t, L_t \right) \Big| s_0 = s \right]$$

$$\text{subject to } s_{t+1} \sim P\left( s_t, a_t, L_t \right), a_t \sim \pi_t\left( s_t \right). \tag{79}$$

Each agent applies a local policy[58] $\pi_t : \mathbb{S} \to \Delta(\mathbb{A})$, which assumes the population state is not observable. Note that both the reward function and the transition dynamics depend on the sequence of the mean-field terms $\{L_t\}_{t=0}^{\infty}$. From each agent's perspective, the MDP is time-varying and is determined by all other agents.

The solution concept in MFG is a variant of the (Markov perfect) NE named the mean-field equilibrium, which is a pair of $\{\pi_t^*, L_t^*\}_{t \geq 0}$ that satisfies two conditions: 1) for fixed $L^* = \{L_t^*\}$, $\pi^* = \{\pi_t^*\}$ is the optimal

---

[54] At time step 0, it can be a random guess. Since the fixed point exists, the final convergence result is irrelevant to the initial guess.

[55] Note that in mean-field learning in games, the state is not assumed to be global. This is different from Dec-POMDP, in which there exists an observation function that maps the global state to the local observation for each agent.

[56] In fact, the homogeneity in MFG can be relaxed to allow agents to have (finite) different types (Lacker & Zariphopoulou, 2019), though within each type, agents must be homogeneous.

[57] MFG is more commonly formulated in a continuous-time setting in the domain of optimal control, where it is typically composed by a backward *Hamilton-Jacobi-Bellman equation* (e.g., the Bellman equation in RL is its discrete-time counterpart) that describes the optimal control problem of an individual agent and a forward *Fokker-Planck equation* that describes the dynamics of the aggregate distribution (i.e., the mean field) of the population.

[58] A general non-local policy $\pi(s, L) : \mathbb{S} \times \Delta^{|\mathbb{S}||\mathbb{A}|} \to \Delta(\mathbb{A})$ is also valid for MFG, and it makes the learning easier by assuming $L$ is fully observable.

policy, that is, $V(s, \pi^*, L^*) \geq V(s, \pi, L^*), \forall \pi, s$; 2) $L^*$ matches with the generated mean field when agents follow $\pi^*$. The two-step iteration process in the meeting start-time example applied in MFG is then expressed as $\Gamma^1(L_t) = \pi_t^*$ and $\Gamma^2(L_t, \pi_t^*) = L_{t+1}$, and it terminates when $\Gamma^2 \circ \Gamma^1(L) = L = L^*$. Mean-field equilibrium is essentially a fixed point of MFG, its existence for discrete-time[59] discounted MFGs has been verified by Saldi et al. (2018) in the infinite-population limit $N \to +\infty$ and also in the partially observable setting (Saldi et al., 2019). However, these works consider the case where the mean field in MFG includes only the population state. Recently, Guo et al. (2019) demonstrated the existence of NE in MFG, taking into account both the population states and actions distributions. In addition, they proved that if $\Gamma^1$ and $\Gamma^2$ meet *small parameter conditions* (Huang et al., 2006), then the NE is unique in the sense of $L^*$. In terms of uniqueness, a common result is based on assuming monotonic cost functions (Lasry & Lions, 2007). In general, MFGs admit multiple equilibria (Nutz et al., 2020); the reachability of multiple equilibria is studied when the cost functions are anti-monotonic (Cecchin et al., 2019) or quadratic (Delarue & Tchuendom, 2020).

Based on the two-step fixed-point iteration in MFGs, various model-free RL algorithms have been proposed for learning the NE. The idea is that in the step $\Gamma^1$, one can approximate the optimal $\pi_t$ given $L_t$ through single-agent RL algorithms[60] such as (deep) Q-learning (Guo et al., 2019; Anahtarcı et al., 2019; Anahtarcı et al., 2020), (deep) policy-gradient methods (Guo et al., 2020; Elie et al., 2020; Subramanian & Mahajan, 2019; uz Zaman et al., 2020), and actor-critic methods (Yang et al., 2019c; Fu et al., 2019). Then, in step $\Gamma^2$, one can compute the forward $L_{t+1}$ by sampling the new $\pi_t$ directly or via fictitious play (Cardaliaguet & Hadikhanloo, 2017; Hadikhanloo & Silva, 2019; Elie et al., 2019). A surprisingly good result is that the sample complexity of both value-based and policy-based learning methods for MFG in fact shares the same order of magnitude as those of single-agent RL algorithms (Guo et al., 2020). However, one major subtlety of these learning algorithms for MFGs is how to obtain stable samples for $L_{t+1}$. For example, Guo et al. (2020) discovered that applying a softmax policy for each agent and projecting the mean-field quantity on an $\epsilon$-net with finite cover help to significantly stabilise the forward propagation of $L_{t+1}$.

## 9.2 Cooperative Mean-Field Control

MFC maintains the same homogeneity assumption and weak interaction assumption as MFG. However, unlike MFG, in which each agent behaves independently, there is a central controller that coordinates all agents' behaviours in the context of MFC. In cooperative multi-agent learning, assuming each agent observes only a local state, the central controller maximises the aggregated accumulative reward:

$$\sup_{\boldsymbol{\pi}} \frac{1}{N} \sum_{i=1}^N \mathbb{E}_{\mathbf{s}_{t+1} \sim P, \mathbf{a}_t \sim \boldsymbol{\pi}} \left[ \sum_t \gamma^t R^i(\mathbf{s}_t, \boldsymbol{a}_t) \Big| \mathbf{s}_0 = \mathbf{s} \right]. \tag{80}$$

Solving Eq. (80) is a combinatorial problem. Clearly, the sample complexity of applying the Q-learning algorithm grows exponentially in $N$ (Even-Dar & Mansour, 2003). To avoid the curse of dimensionality in $N$, MFC (Carmona et al., 2018; Gu et al., 2019) pushes $N \to +\infty$, and under the law of large numbers and the theory of propagation of chaos (Gärtner, 1988; McKean, 1967; Sznitman, 1991), the optimisation problem in Eq. (80), in the view of a representative agent, can be equivalently written as

$$\sup_{\pi} \mathbb{E} \left[ \sum_t \gamma^t \tilde{R}(s_t, a_t, \mu_t, \alpha_t) \Big| s_0 \sim \mu \right]$$
$$\text{subject to } s_{t+1} \sim P(s_t, a_t, \mu_t, \alpha_t), a_t \sim \pi_t(s_t, \mu_t). \tag{81}$$

in which $(\mu_t, \alpha_t)$ is the respective state and action marginal distribution of the mean-field quantity, $\mu_t(\cdot) = \lim_{N \to +\infty} \sum_{i=1}^N \mathbb{1}(s_t^i = \cdot)/N$, $\alpha_t(\cdot) = \sum_{s \in \mathbb{S}} \mu_t(s) \cdot \pi_t(s, \mu_t)(\cdot)$, and $\tilde{R} = \lim_{N \to +\infty} \sum_i R^i/N$. The MFC approach is attractive not only because the dimension of MFC is independent of $N$, but also because MFC

---

[59] The existence of equilibrium in continuous-time MFGs is widely studied in the area of stochastic control (Huang et al., 2006; Lasry & Lions, 2007; Carmona & Delarue, 2013; Carmona et al., 2015b; 2016; Cardaliaguet et al., 2015; Fischer et al., 2017; Lacker, 2018; 2015), though it may be of less interest to RL researchers.

[60] Since agents in MFG are homogeneous, if the representative agent reaches convergence, then the joint policy is the NE. Additionally, given $L_t$, the MDP to the representative agent is stationary.

has shown to approximate the original cooperative game in terms of both game values and optimal strategies (Lacker, 2017; Motte & Pham, 2019).

Although the MFC formulation in Eq. (81) appears similar to the MFG formulation in Eq. (79), their underlying physical meaning is fundamentally different. As is illustrated in Figure 11, the difference is which operation is performed first: learning the equilibrium of the $N$-player game or taking the limit as $N \to +\infty$. In the fixed-point iteration of MFG, one first assumes $L_t$ is given and then lets the (infinite) number of agents find the best response to $L_t$, while in MFC, one assumes an infinite number of agents to avoid the curse of dimensionality in cooperative MARL and then finds the optimal policy for each agent from a central controller perspective. In addition, compared to mean-field NE in MFG, the solution concept of the central controller in MFC is the Pareto optimum[61], an equilibrium point where no individual can be better off without making others worse off. Finally, other differences between MFG and MFC can be found in Carmona et al. (2013).

In MFC, since the marginal distribution of states serves as an input in the agent's policy and is no longer assumed to be known in each iteration (in contrast to MFG), the dynamic programming principle no longer holds in MFC due to its non-Markovian nature (Andersson & Djehiche, 2011; Buckdahn et al., 2011; Carmona et al., 2015a). That is, MFC problems are inherently time-inconsistent. A counter-example of the failure of standard Q-learning in MFC can be found in Gu et al. (2019). One solution is to learn MFC by adding common noise to the underlying dynamics such that all existing theory on learning MDP with stochastic dynamics can be applied, such as Q-learning (Carmona et al., 2019b). In the special class of linear-quadratic MFCs, Carmona et al. (2019a) studied the policy-gradient method and its convergence, and Luo et al. (2019) explored an actor-critic algorithm. However, this approach of adding common noise still suffers from high sample complexity and weak empirical performance (Gu et al., 2019). Importantly, applying dynamic programming in this setting lacks rigorous verifications, leaving aside the measurability issues and the existence of a stationary optimal policy.

Another way to address the time inconsistency in MFCs is to consider an **enlarged** state-action space (Laurière & Pironneau, 2014; Pham & Wei, 2016; 2018; 2017; Djete et al., 2019; Gu et al., 2019). This technique is also called "lift up", which essentially means to lift up the state space and the action space into their corresponding probability measure spaces in which dynamic programming principles hold. For example, Gu et al. (2019); Motte & Pham (2019) proposed to lift the finite state-action space $\mathbb{S}$ and $\mathbb{A}$ to a compact state-action space embedded in Euclidean space denoted by $\mathcal{C} := \Delta(\mathbb{S}) \times \mathcal{H}$ and $\mathcal{H} := \{h : \mathbb{S} \to \Delta(\mathbb{A})\}$, and the optimal Q-function associated with the MFC problem in Eq. (81) is

$$Q_{\mathcal{C}}(\mu, h) = \sup_{\pi} \mathbb{E}\left[\sum_{t=0}^{\infty} \gamma^t \tilde{R}(s_t, a_t, \mu_t, \alpha_t)\Big| s_0 \sim \mu, u_0 \sim \alpha, a_t \sim \pi_t\right], \forall(\mu, h) \sim \mathcal{C}. \tag{82}$$

The physical meaning of $\mathcal{H}$ is the set of all possible local policies $h : \mathbb{S} \to \Delta(\mathbb{A})$ over all different states. Note that after lift up, the mean-field term $\mu_t$ in $\pi_t$ of Eq. (81) no longer exists as an input to $h$. Although the support of each $h$ is $|\Delta(\mathbb{A})|^{|\mathbb{S}|}$, it proves to be the minimum space under which the Bellman equation can hold. The Bellman equation for $Q_{\mathcal{C}} : C \to \mathbb{R}$ is

$$Q_{\mathcal{C}}(\mu, h) = R(\mu, h) + \gamma \sup_{\tilde{h} \in \mathcal{H}} Q_{\mathcal{C}}\left(\Phi(\mu, h), \tilde{h}\right) \tag{83}$$

where $R$ and $\Phi$ are the reward function and transition dynamics written as

$$R(\mu, h) = \sum_{s \in \mathbb{S}} \sum_{a \in \mathcal{A}} \tilde{R}(s, a, \mu, \alpha(\mu, h)) \cdot \mu(s) \cdot h(s)(a) \tag{84}$$

$$\Phi(\mu, h) = \sum_{s \in \mathbb{S}} \sum_{a \in \mathbb{A}} P(s, a, \mu, \alpha(\mu, h)) \cdot \mu(s) \cdot h(s)(a) \tag{85}$$

with $\alpha(\mu, h)(\cdot) := \sum_{s \in \mathbb{S}} \mu(s) \cdot h(s)(\cdot)$ representing the marginal distribution of the mean-field quantity in action. The optimal value function is $V^*(\mu) = \max_{h \in \mathcal{H}} Q_{\mathcal{C}}(\mu, h)$. Since both $\mu$ and $h$ are probability

---

[61]The Pareto optimum is a subset of NE.

distributions, the difficulty of learning MFC then changes to how to deal with continuous state and continuous action inputs to $Q_{\mathcal{C}}(\mu, h)$, which is still an open research question. Gu et al. (2020) tried to discretise the lifted space $\mathcal{C}$ through $\epsilon$-net and then adopted the kernel regression on top of the discretisation; impressively, the sample complexity of the induced Q-learning algorithm is independent of the number of agents $N$.

## 9.3 Mean-Field MARL

The scalability issue of multi-agent learning in non-cooperative general-sum games can also be alleviated by applying the mean-field approximation directly to each agent's Q-function (Zhou et al., 2019; Yang et al., 2018b; Subramanian et al., 2020). In fact, Yang et al. (2018b) was the first to combine mean-field theory with the MARL algorithm. The idea is to first factorise the Q-function using only the local pairwise interactions between agents (see Eq. (86)) and then apply the mean-field approximation; specifically, one can write the neighbouring agent's action $a^k$ as the sum of the mean action $\bar{a}^j$ and a fluctuation term $\delta a^{j,k}$, i.e., $a^k = \bar{a}^j + \delta a^{j,k}$, $\bar{a}^j = \frac{1}{N^j} \sum_k a^k$, in which $\mathcal{N}(j)$ is the set of neighbouring agents of the learning agent $j$ with its size being $N^j = |\mathcal{N}^j|$. With the above two processes, we can reach the mean-field Q-function $Q^j(s, a^j, \bar{a}^j)$ that approximates $Q^j(s, \mathbf{a})$ as follows

$$Q^j(s, \mathbf{a}) = \frac{1}{N^j} \sum_k Q^j(s, a^j, a^k) \tag{86}$$

$$= \frac{1}{N^j} \sum_k \left[ Q^j(s, a^j, \bar{a}^j) + \nabla_{\bar{a}^j} Q^j(s, a^j, \bar{a}^j) \cdot \delta a^{j,k} \right.$$

$$\left. + \frac{1}{2} \delta a^{j,k} \cdot \nabla^2_{\tilde{a}^{j,k}} Q^j(s, a^j, \tilde{a}^{j,k}) \cdot \delta a^{j,k} \right] \tag{87}$$

$$= Q^j(s, a^j, \bar{a}^j) + \nabla_{\bar{a}^j} Q^j(s, a^j, \bar{a}^j) \cdot \left[ \frac{1}{N^j} \sum_k \delta a^{j,k} \right]$$

$$+ \frac{1}{2N^j} \sum_k \left[ \delta a^{j,k} \cdot \nabla^2_{\tilde{a}^{j,k}} Q^j(s, a^j, \tilde{a}^{j,k}) \cdot \delta a^{j,k} \right] \tag{88}$$

$$= Q^j(s, a^j, \bar{a}^j) + \frac{1}{2N^j} \sum_k R^j_{s,a^j}(a^k) $$

$$\approx Q^j(s, a^j, \bar{a}^j) . \tag{89}$$

The second term in Eq. (88) is zero by definition, and the third term can be bounded if the Q-function is smooth, and it is neglected on purpose. The mean-field action $\bar{a}^j$ can be interpreted as the empirical distribution of the actions taken by agent $j$'s neighbours. However, unlike the mean-field quantity in MFG or MFC, this quantity does not have to assume an infinite population of agents, which is more friendly for many real-world tasks, although a large $N$ can reduce the approximation error between $a^k$ and $\bar{a}^j$ due to the law of large numbers. In addition, the mean-field term in MF-MARL does not include the state distribution, unlike MFG or MFC.

Based on the mean-field Q-function, one can write the Q-learning update as

$$Q^j_{t+1}(s, a^j, \bar{a}^j) = (1 - \alpha) Q^j_t(s, a^j, \bar{a}^j) + \alpha \left[ R^j + \gamma v^{j,\mathrm{MF}}_t(s') \right]$$

$$v^{j,\mathrm{MF}}_t(s') = \sum_{a^j} \pi^j_t(a^j \mid s', \bar{a}^j) \cdot \mathbb{E}_{\bar{a}^j(a^{-j}) \sim \pi^{-j}_t} \left[ Q^j_t(s', a^j, \bar{a}^j) \right]. \tag{90}$$

The mean action $\bar{a}^j$ depends on $a^j, j \in \mathcal{N}(j)$, which itself depends on the mean action. The chicken-and-egg problem is essentially the time inconsistency that also occurs in MFC. To avoid coupling between $a^j$ and $\bar{a}^j$, Yang et al. (2018b) proposed a filtration such that in each stage game $\{Q_t\}$, the mean action $\bar{a}^j$ is computed first using each agents' current policies, i.e., $\bar{a}^j = \frac{1}{N^j} \sum_k a^k, a^k \sim \pi^k_t$, and then given $\bar{a}^j$, each agent finds the best response by

$$\pi^j_t(a^j \mid s, \bar{a}^j) = \frac{\exp\left(\beta Q^j_t(s, a^j, \bar{a}^j)\right)}{\sum_{a^j \in \mathbb{A}^j} \exp\left(\beta Q^j_t(s, a^{j'}, \bar{a}^j)\right)}. \tag{91}$$

For large $\beta$, the Boltzmann policy in Eq. (91) proves to be a contraction mapping, which means the optimal action $a^j$ is unique given $\bar{a}^{j}\_{\_\_}$; therefore, the chicken-and-egg problem is resolved[62].

MF-Q can be regarded as a modification of the Nash-Q learning algorithm (Hu & Wellman, 2003), with the solution concept changed from NE to mean-field NE (see the definition in MFG). As a result, under the same conditions, which include the strong assumption that there exists a unique NE at every stage game encountered, $\mathbf{H}^{\mathrm{MF}}\boldsymbol{Q}(s, \boldsymbol{a}) = \mathbb{E}_{s' \sim p}\left[\boldsymbol{R}(s, \boldsymbol{a}) + \gamma \boldsymbol{v}^{\mathrm{MF}}(s')\right]$ proves to be a contraction operator. Furthermore, the asymptotic convergence of the MF-Q learning update in Eq. (90) has also been established.

Considering only pairwise interactions in MF-Q may appear rather limited. However, it has been noted that the pairwise approximation of the agent and its neighbours, while significantly reducing the complexity of the interactions among agents, can still preserve global interactions between any pair of agents (Blume, 1993). In fact, such an approach is widely adopted in other machine learning domains, for example, factorisation machines (Rendle, 2010) and learning to rank (Cao et al., 2007). Based on MF-Q, Li et al. (2019a) solved the real-world taxi order dispatching task for Uber China and demonstrated strong empirical performance against humans. Subramanian & Mahajan (2019) extended MF-Q to include multiple types of agents and applied the method to a large-scale predator-prey simulation scenario. Ganapathi Subramanian et al. (2020) further relaxed the assumption that agents have access to exact cumulative metrics regarding the mean-field behaviour of the system, and proposed partially observable MF-Q that maintains a distribution to model the uncertainty regarding the mean field of the system.

## 10 Future Directions of Interest

**MARL Theory.**   In contrast to the remarkable empirical success of MARL methods, developing theoretical understandings of MARL techniques are very much under-explored in the literature. Although many early works have been conducted on understanding the convergence property and the finite-sample bound of single-agent RL algorithms (Bertsekas & Tsitsiklis, 1996), extending those results into multi-agent, even many-agent, settings seem to be non-trivial. Furthermore, it has become a common practice nowadays to use DNNs to represent value functions in RL and multi-agent RL. In fact, many recent remarkable successes of multi-agent RL benefit from the success of deep learning techniques (Vinyals et al., 2019a; Baker et al., 2019b; Pachocki et al., 2018). Therefore, there are pressing needs to develop theories that could explain and offer insights into the effectiveness of deep MARL methods. Overall, I believe there is an approximate ten-year gap between the theoretical developments of single-agent RL and multi-agent RL algorithms. Learning the lessons from single-agent RL theories and extending them into multi-agent settings, especially understanding the incurred difficulty due to involving multiple agents, and then generalising the theoretical results to include DNNs could probably act as a practical road map in developing MARL theories. Along this thread, I recommend the work of Zhang et al. (2019b) for a comprehensive summary of existing MARL algorithms that come with theoretical convergence guarantee.

**Safe and Robust MARL.**   Although RL provides a general framework for optimal decision making, it has to incorporate certain types of constraints when RL models are truly to be deployed in the real-world environment. I believe it is critical to firstly account for MARL with robustness and safety constraints; one direct example is on autonomous driving. At a very high level, robustness refers to the property that an algorithm can generalise and maintain robust performance in settings that are different from the training environment (Morimoto & Doya, 2005; Abdullah et al., 2019). And safety refers to the property that an algorithm can only act in a pre-defined safety zone with minimum times of violations even during training time (García & Fernández, 2015). In fact, the community is still at the early stage of developing theoretical frameworks to encompass either robust or safe constraint in single-agent settings. In the multi-agent setting, the problem could only become more challenging because the solution now requires to take into account the coupling effect between agents, especially those agents that have conflict interests (Li et al., 2019b). In addition to opponents, one should also consider robustness towards the uncertainty of environmental

---

[62]Coincidentally, the techniques of fixing the mean-field term first and adopting the Boltzmann policy for each agent were discovered by Guo et al. (2019) in learning MFGs at the same time.

dynamics (Zhang et al., 2020), which in turn will change the behaviours of opponents and pose a more significant challenge.

**Model-Based MARL.** Most of the algorithms I have introduced in this monograph are *model-free*, in the sense that the RL agent does not need to know how the environment works and it can learn how to behave optimally through purely interacting with the environment. In the classic control domain, *model-based* approaches have been extensively studied in which the learning agent will first build an explicit state-space "model" to understand how the environment works in terms of state-transition dynamics and reward function, and then learn from the "model". The benefit of model-based algorithms lies in the fact that they often require much fewer data samples from the environment (Deisenroth & Rasmussen, 2011). The MARL community has initially come up with model-based approaches, for example the famous R-MAX algorithm (Brafman & Tennenholtz, 2002), nearly two decades ago. Surprisingly, the developments along the model-based thread halted ever since. Given the impressive results that model-based approaches have demonstrated on single-agent RL tasks (Schrittwieser et al., 2020; Hafner et al., 2019a;b), model-based MARL approaches deserves more attention from the community.

**Multi-Agent Meta-RL.** Throughout this monograph, I have introduced many MARL applications; each task needs a bespoke MARL model to solve. A natural question to ask is whether we can use one model that can generalise across multiple tasks. For example, Terry et al. (2020) has put together almost one hundred MARL tasks, including Atari, robotics, and various kinds of board games and pokers into a Gym API. An ambitious goal is to develop algorithms that can solve all of the tasks in one or a few shots. This requires multi-agent meta-RL techniques. Meta-learning aims to train a generalised model on a variety of learning tasks, such that it can solve new learning tasks with few or without additional training samples. Fortunately, Finn et al. (2017) has proposed a general meta-learning framework – MAML – that is compatible with any model trained with gradient-descent based methods. Although MAML works well on supervised learning tasks, developing meta-RL algorithms seems to be highly non-trivial (Rothfuss et al., 2018), and introducing the meta-learning framework on top of MARL is even an uncharted territory. I expect multi-agent meta-RL to be a challenging yet fruitful research topic, since making a group of agents master multiple games necessarily requires agents to automatically discover their identities and roles when playing different games; this itself is a hot research idea. Besides, the meta-learner in the outer loop would need to figure out how to compute the gradients with respect to the entire inner-loop subroutine, which must be a MARL algorithm such as multi-agent policy gradient method or mean-field Q-learning, and, this would probably lead to exciting enhancements to the existing meta-learning framework.

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
