# OpenReview forum: "A Survey of Multi-agent Reinforcement Learning from Game Theoretical Perspective"
_TMLR — Withdrawn by Authors_

### Review · Reviewer_ZKQA · 2022-05-31

**Summary Of Contributions:**

The authors present a review of (mostly) recent multi-agent reinforcement learning (MARL) literature, and game theoretical approches to such multi-agent problems, attempting at providing a fresh & game-theory centric view of MARL.

**Broader Impact Concerns:**

The manuscript did not include a particular statement about societal impact. I believe it could use one that could focus on what the authors might perceive as weaknesses wrt. the particular set of materials they chose to review (particularly with the intention to figure out what effect this review might have on work that didn't get reviewed). The manuscript does attempt to list some of these at the end, but IMO it can do better.

**Requested Changes:**

Overall, I don't think this manuscript is ready for publication at TMLR, as its targeting is a bit confusing and could skew (particularly younger) memebers of the community towards misunderstanding rather than enlightnment over the field of MARL.

Here's a list of improvements I would like to see to be able to improve my eventual score:

1. Improve some of the handwavy statements made when talking about MARL / deep learning (and combinations of the two).

2. Attempt to reduce the dependency on the original material where possible, especially on some of the more complex game-theoretical algorithmic details;

3. Focus and declare the targeting of a specific audience (IMO either incoming researchers, or existing members of game-theory / MARL folks), and generally work out the required level of details to use when discussing straightforward as well as complex work.,


More specifically, these are notes I took while reading the manuscript that loosely match the above points:

### Abstract

- Seems to imply that MARL boomed up due to AphaGo, but that's unclear to me. Most theoretical MARL work happened before AlphaGo, and more recent applied work has been primarily driven due to the emergence of better testbeds and learning systems.

### Section 1

- Nit: a computer program that can perform certain task(s) -> Grammar: Either a certain task or certain tasks.

- special kind of -> Why special?

- which allows the software to train itself to perform new tasks rather than merely relying on the programmer for designing hand-crafted rules -> This is _very_ handwavey, and really applies to all of machine learning.

### 1.1

- Nit: sub-field -> subfield.

- an image labelled with cats -> poor phrasing.

- RL is goal-oriented: it constructs a learning model that learns to achieve the optimal long-term goal by improvement through trial and error, with the learner having no labelled data to obtain knowledge from. -> this is an imprecise characterisation of RL. RL technically is neither goal-orientied nor generally model-learning focused, furthermore a reward function is a type of label over a particular task (just a particularly poorly informative and indirect kind of label)

- from DeepMind -> poor form to mention the company in question. Focus on the authors / manuscript unless making a meta-point.

- As a result, RL methods can naturally be used to train a computer program (an agent) to a performance level comparable to that of a human on certain tasks. -> this does not follow from the previous sentence; ie the fact that some RL tasks can be successfully solved today is not due to RL being (eg) nature-inspired.

- Nit: GO -> Go / go / baduk.

### 1.2

- It is possible that the AlphaGo series (Silver et al., 2016; 2017; 2018) has largely fulfilled people’s expectations for the effectiveness of RL methods, such that there is a lack of interest in further advancements in the field. -> seems to be a counterargument to a previously made point about the surging popularity of RL. Which is what?

- One popular test-bed of MARL is StarCraft II (Vinyals et al., 2017) -> Pysc2 is not a multi-agent testbed. Did you mean to talk about SMAC here?

- However, a breakthrough was accomplished by AlphaStar in 2019 (Vinyals et al., 2019b), which has exhibited grandmaster-level skills by ranking above 99.8% of human players. -> Arguably very much not a MARL breakthrough.

- Another prominent video game-based test-bed for MARL is Dota2 -> Is it? Only handful of papers on DOTA in the litearature (and IIRC not many at all on MARL!), due to the interface being private.

- Nit: test-bed -> testbed

### 2.1

- Define delta / simplex function in 3rd bullet point instead of later paragraph.

- Generally worth stating whether you are talking about episodic vs non episodic MDPs.

### 2.3

- vice-versa -> not sure what this is referring to. Worth being clearer.

- The word “learning" essentially means that the agent turns its experience gained during the interaction into knowledge about the model of the environment. -> not really? Even given a fairly generous interpretation of what RL does, this seems factually wrong.

- What's the "solution target"?

- contraction mapping -> Undefined, like some of the other perhaps more unusual mathematical terms across the manuscript. IMO too specific of a detail for beginners of the field, and too imprecise for experts.

- replay buffer -> undefined; worth explaining more here.

### 2.3.2

- update the parameter θ in the direction that maximises the cumulative reward [equation] ->  the equation of the weights update here is doing a lot; worth unpacking it...

- When the policy is deterministic and the action set is continuous, one obtains the deterministic policy gradient (DPG) theorem (Silver et al., 2014) -> why was DPG singled out here in particular?

- Important variants -> Why are they important?

### 3

- reward function that each agent receives -> Best to specify here that these could be different between different agents, think eg cooperative vs non-coop settings.

### 3.1

- The rewards have absolute values uniformly bounded by $$R_{max}$$ -> Why? Can you elaborate more here?

- We use the superscript of (·i,·−i) -> Where does this notation come from? Perhaps Foerster 2017b? Helpful to mention why it helps.

### 3.2.1

- The single-agent Q-learning update in Eq. (7) still holds in the multi-agent case. -> where is this shown?

### 3.2.2

- This characteristic necessitates the development of policy-based algorithms with function approximations. -> Does it mean that PG methods don't suffer from the curse of dimensionality? Surely they do (just in different manners)?

- Considering a continuous action set with a deterministic policy -> Again, why singleing out MADDPG amongst all of possible PG settings?

### 3.2.5

- However, the benefits of studying games in the partially observable setting come from the algorithmic advantages -> What does this mean?

- I conclude Section 3 by presenting the relationships between the many different types of POSGs through a Venn diagram in Figure 6. -> Should be we? Also the section is not concluded...

### 3.3

- An important feature of EFGs is that they can handle imperfect information for multi-player decision making. -> Should probably be mentioned earlier in the section.

### 4

- MARL methods pose more theoretical challenges -> Do you mean MARL problems?

- all result in the majority of MARL algorithms being capable of solving games with ⃝4 only two players -> I'm skeptical this is generally true. Very often MARL is not even concerned with the # of players, and scale issues come more from task complexity as well as # of agents (and they are not the same thing?).

- I will -> we will.

### 4.1

- Q-function decomposition -> Needs definition.

- Reaching both rationality and convergence gives rise to reaching the NE. -> Odd phrasing.

### 4.3

- The stationarity is thus maintained, so single-agent RL algorithms can still be applied. ->  In principle many RL algorithms are commonly applied in these settings?

- which provide the learning algorithms expressive function approximators. -> unsure what this means?

### 5

- I provide -> We provide.

- (Again) I introduce -> We introduce.

- I review -> ...


### 5.1

- unanimously defined -> Unsure about this. Very often it is hard to define the right MDP in a single agent problem.

- learning objectives -> Are rationality and convergence here "learning objectives" in the same way maximising the reward function would be? Is that fair?

... and so on.


**Strengths And Weaknesses:**

Let me first start by mentioning that I applaud the authors for attempting what few researches seem to be willimg to do in our field: writing detailed, overarching surveys about fast evolving, large, fields of research. We need more of these, and we need them to be of very high quality to help getting past the arxiv noise.

Overal, I found the survey to be extremely helpful to brush back up on fundamental and recent proofs in theoretical game theory applied to agent learning, and overall found it quite decent to read compared to the foundational literature in the field. However, I believe the manuscript requires some good improvements before it can become the work the authors wish it to be. To summarise:

### Strengths

1. Really strong coverage of game-theoretical work. While it is not my primary area of research, I attempted a similar work as an exercise to review this manuscript, and overall found it to be quite reasonably covering of the most relevant parts of the field. Obviously this is not to say say that it is suffiecient, but it provides enough pointers and background to the willing reader to be a strong entry point for game theory applied to multi-atent deicison making problems.

2. A lot of complex work is explained in simple terms, with ; the paper is stated to be written (also) for a generalist researchers looking to work on the field, and it does so excellently;

3. It does a good job of summarising recent important (Deep)MARL work, which have not been meta-surveyed as of today;

4. Tackles the intersection of two popular and important (and perhaps growing) fields, so I believe this to be of great interestest to the TMLR community.

### Weaknesses

1. The writing is extremely handwavey at times. My expecatation is that a survey should be extremely precise when making statements about existing literature, and speculativge when declared so (to provide some kind of narrative and meta-take over a particular set of research work).

2. It is not clear to me the authors targeted a useful subset of researchers as the target audience. Very often in the manuscript will go over many important proofs and concepts in a short amount of lines, in such a way that forces the reader to essentially revert back to the original work when looking for specific mathematical details (and jargon!). At the same time, many perhaps simpler concepts are explained in perhaps unnecessary detail (again, for a semi-expert reader).

3. The review feels like it was primarily written as a survey of game theoretical work, and then MARL was retrofitted to be included as a strengthening point. Whilst there are indeed many references and overall a good overview of recent MARL work and the general state of the field, it does seem like for the vast majority of cited MARL algorithmic work, most of the details are left to the reader to go pick up from the original material, rather than providing the necessary info in text.

4. Following up on (2) and (3), the applied game-theoretical "taxonomy" makes it perhaps hard to compare cited MARL work appropriately between each other; it's not clear to me whether a new reader would find it a particularly useful view of the field.

---

### Review · Reviewer_bvGr · 2022-06-01

**Summary Of Contributions:**

This work presents an extensive survey of the field of multi-agent reinforcement learning from a game theoretic perspective. Firstly, it incorporates theoretical background on both single- and multi-agent RL, problem formulation, main solution concepts and challenges. Secondly, it also puts in perspective the previous surveys of the field and proceeds to offer an overview of complementary taxonomies and present algorithmic solutions devised on the considered problem setting (i.e., game formulation).

**Requested Changes:**

My main concerns are with respect to Sections 5 and 6 of the paper. In Section 5.1, the work presents 4 perspectives from which a taxonomy of the field can be constructed: stage game types, knowledge assumptions, learning paradigms, 5 AI Agendas, however it is not clear that this is actually the overarching taxonomy proposed by the work. It is also not clear how Section 6 then maps to this proposed taxonomy. A bit of extra text on explaining and consolidating these elements would help here.

Secondly, it would be useful to also add a small discussion on the interactions between the different taxonomy perspectives. For example, there is a strong connection between the knowledge assumptions and the learning paradigms, or between the learning paradigms and the game settings (e.g., CTDE in cooperative settings).

Minor remarks:
- In the MDP and SG definitions, $\gamma$ is defined to represent the value of time. I argue to also add that this is with respect to the reward.
- Page 7, sentence 'When the state-action space of an MDP is continuous, LP formulation cannot help solve either.' is unclear, or seems incomplete. Please verify.
- Notation overload R_t as immediate reward (Section 2.3.1), versus return (Section 2.3.2)
- Abbreviation overload for PG, Section 2.3.2 policy gradient versus Section 6.4 potential game
- Page 13, Definition 4, inconsistent notation of the state space in the NFG bullet
- Equation 38, missing 'exp'?
- check quotation marks, they appear inconsistent on the right sides
- Missing reference to Table 2 in text (e.g., this can be added in the Based on Five AI Agendas paragraph)

Optional suggestions:
- using the 'we' versus 'I' personal pronoun usually gives a more objective tone to the work, however it depends on the number of authors, so I leave this to the authors to decide upon
- another potential emerging direction is multi-objective multi-agent decision-making: Rădulescu, R., Mannion, P., Roijers, D. M., & Nowé, A. (2020). Multi-objective multi-agent decision making: a utility-based analysis and survey. Autonomous Agents and Multi-Agent Systems, 34(1), 1-52.

Other references:
- Section 6.1.1 -  Böhmer, W., Kurin, V., & Whiteson, S. (2020, November). Deep coordination graphs. In International Conference on Machine Learning (pp. 980-991). PMLR.


**Strengths And Weaknesses:**

I believe this work is highly valuable to the rapidly growing MARL community as it presents an extensive overview of the field, with a great emphasis on the problem formulation side.

Strengths:
- comprehensive background, problem formulation and solution concepts
- good summary of challenges of the field
- extensive survey at both theoretical and algorithmic levels
- great support for researchers of all levels

Weaknesses:
- Taxonomy formulation can stand to be improved and better highlighted
- A discussion on the interactions between the complementary taxonomies is warranted

---

### Review · Reviewer_t3MK · 2022-06-01

**Summary Of Contributions:**

The author has produced a very comprehensive up-to-date well written survey of multi-agent reinforcement learning, a field that has seen very strong activity in recent years and that I believe will just continue to be growing in importance within AI/ML. Much of the on-going work have strong ties to game theory, which suitable has been chosen as the perspective and language to approach to the topic from. Otherwise the author remains very even handed in both selection and presentation, and the result is a very useful reference work as well as a guide to the field as it stands in 2022.

**Broader Impact Concerns:**

No such concerns with providing a survey of the field.
If applications were brought up the authors could have also discussed what impact the further development of MARL might have on society.

**Requested Changes:**

I do not propose any changes, though I would not mind seeing 1-3 pages about applications.

**Strengths And Weaknesses:**

The author has chosen to try to just represent the field as it stands and is not adding too much opinion or aiming to reveal new connections or ways to see things, even if they do utilize the game theory to outline how the field sits together and they have carried this out more comprehensively than previous authors. This is both the strength and the weakness, while from the criteria of this journal that are focused on clarity and correctness it ends up squarely in the strength category.

One area that is not really featured is applications, even if illustrative examples that relates to applications are used.

---

### Note · Authors · 2022-07-07

**Comment:**

Revised and resubmit

**Withdrawal Confirmation:**

I have read and agree with the venue's withdrawal policy on behalf of myself and my co-authors.